# Analysis and source identification of dust event in Khuzestan province using MODIS and SENTINEL satellite data and HYSPLIT model (2018–2022)

Alireza Yousefi Kebriya[1], Mehdi Nadi[1]*, Ebadat Ghanbari Parmehr[2], Zhongchang Sun[3]*

**1** Department of Water Engineering, Sari Agricultural Sciences and Natural Resources University, sari, Iran, **2** Department of Remote Sensing, Faculty of Surveying Engineering, Noshirvani Babol University of Technology, Babol, Iran, **3** International Research Center of Big Data for Sustainable Development Goals, Beijing, China

* mehdi.nadi@gmail.com (MN); sunzc@aircas.ac.cn (ZS)

## Abstract

Dust events present significant environmental and public health challenges in Khuzestan Province, Iran, severely affecting air quality, human health, and ecosystems. This study analyzed dust dynamics from 2018 to 2022 using satellite-derived Aerosol Optical Depth (AOD), Aerosol Absorption Index (AAI), ground-based $PM_{2.5}$ and $PM_{10}$ data, HYSPLIT trajectory modeling, wind rose analysis, and wetland assessments. Strong correlations were found between AOD and $PM_{2.5}$ ($R^2 = 0.79$) and $PM_{10}$ ($R^2 = 0.91$), confirming the reliability of satellite monitoring. $PM_{2.5}$ ranged 12–750 µg/m³ and $PM_{10}$ 10–800 µg/m³, with peaks linked to drought, wetland shrinkage, and transboundary inflows. AOD-Max reached 3.2–3.9, with temporal variability reflecting flooding and drought events. Annual trajectory clustering identified six major dust pathways to Ahvaz; the primary route (43.48%) crossed eastern Iraq and Hoor al-Azim, with 65.2% of pathways traversing Iraqi deserts. Weighted Potential Source Contribution Function (WPSCF) and Weighted Concentration Weighted Trajectory (WCWT) mapping revealed a northwest–southeast dust belt across Iraq (AOD 0.6–0.8, locally >1.0) and secondary zones in southern Iraq, the Khuzestan border, eastern Syria, and Ninawa (AOD 0.4–0.6, rising to 0.6–0.9). Dust reached Ahvaz within 24 hours, with strong AOD correlations ($R^2 = 0.85$–0.75; 2018–2022). The Hoor al-Azim Wetland contracted dramatically from ~2,000 km² (>5 m water) in 2019 to ~73 km² (1.2–2.9 m) in July 2022, with low MNDWI values (0.33 to −0.29). Wetland area correlated strongly with AOD ($R^2 = 0.78$) and water level ($R^2 = 0.71$). Three hydrological–air quality states emerged: healthy (~1,100–2,000 km², AOD < 0.4), transitional (400–1,000 km², AOD 0.4–1.0), and critical (<250 km², AOD > 1.0, up to ~1.7), with a ~500 km² threshold marking sharp reduction in dust suppression. These findings highlight the urgent need for wetland restoration, desertification control, and transboundary cooperation to mitigate dust risks in the region.

**Data availability statement:** The data underlying the results presented in this study were obtained from multiple sources. Meteorological and environmental variables were processed and extracted using Google Earth Engine. Some datasets were provided by the Department of Environment of Iran, while high-resolution HYSPLIT input data were obtained from the NOAA website. Additionally, MODIS true-color satellite imagery was sourced from NASA. All relevant processed data generated and analyzed during this study are included within the paper and its Supporting Information files.

**Funding:** This work was supported by the National Natural Science Foundation of China (grant numbers 42361144884 and 42171291) and the Joint HKU-CAS Laboratory for iEarth (313GJHZ2022074MI, E4F3050300). The authors appreciate the support of Sari Agricultural Sciences and Natural Resources University.

**Competing interests:** The authors have declared that no competing interests exist.

# 1. Introduction

## 1.1. Background of the study

Dust events, a severe form of atmospheric pollution originating in arid and semi-arid regions, have emerged as a global environmental challenge with wide-ranging consequences for ecosystems, human health, agriculture, transportation, and power generation [1], as well as significant radiative and climatic impacts [2]. In several regions of the world, the increasing frequency and intensity of dust events have been linked to climate change and human activities, while in other areas, their occurrence has shown a declining trend in recent years, reflecting strong regional variability in dust dynamics. The Middle East, due to its specific climatic conditions such as droughts and strong winds, has a long history of experiencing dust events [3]. These storms typically originate from arid regions like deserts and transport dust particles to other areas [4].

Wetlands are disappearing globally at alarming rates, with 71% of wetlands converted to other land-cover types since 1900 [5]. In southwestern Iran, the Shadegan Wetland has been increasingly exposed to drought and land-cover changes, causing previously inundated areas to dry out and become active dust sources. Remote sensing and supervised classification analyses of Landsat TM, ETM, and OLI images for 1989, 2003, and 2017 indicate that while agricultural land, wetlands, water bodies, and built-up areas increased between 1989 and 2003, bare lands and wetland vegetation decreased, enhancing soil erosion. Between 2003 and 2017, bare lands, water bodies, and built-up areas further increased, whereas agricultural lands, wetlands, and wetland vegetation decreased, with soil erosion rates rising from 45.56% in 1989 to 52.24% in 2017. These results indicate that land-cover changes, particularly wetland degradation, have made the Shadegan Wetland a major internal source of dust in Khuzestan Province [6].

The increasing frequency and intensity of dust events have been attributed to a combination of factors, including droughts, land degradation, and changes in land use, vegetation cover, and soil erosion [7]. This highlights the diverse origins of dust events, which are no longer confined to remote desert landscapes but are also occurring in a wide range of warm, semi-arid, and temperate regions [8]. The geographical location and meteorological parameters of these regions play a significant role in dust event formation and transport, leading to cross-border pollution issues and necessitating a comprehensive analysis of their sources and impacts [9]. The complex nature of dust events demands a multidisciplinary approach to investigate their environmental drivers and quantify their effects at regional and global scales [10].

The use of the HYSPLIT model has emerged as a powerful tool for tracking and predicting the movement of dust events, providing valuable insights into their origins and trajectories [11]. This, coupled with advanced remote sensing techniques, has significantly transformed the capacity of studies in this field to monitor and analyze the environmental parameters that govern dust event formation [12]. Variables such as soil moisture levels, atmospheric temperatures, precipitation patterns, vegetation cover, and atmospheric pressures can now be observed with unprecedented

accuracy, creating favorable conditions for detailed dust event analysis. This information is crucial for understanding dust sources, predicting dust distribution over time, and assessing the spatial impacts of dust event [13].

Khuzestan province, located in southwestern Iran along the border with Iraq, encompasses extensive desert areas and a diverse range of land uses, including agricultural lands, urban areas, and natural landscapes. The province also hosts two major wetlands, Shadegan and Hoor al-Azim, which are crucial for local hydrology, biodiversity, and ecosystem services. Over recent decades, these wetlands have experienced significant shrinkage due to prolonged drought and changes in water management, exposing large areas of dry land that act as major dust sources. Despite the ecological and economic importance of Khuzestan, detailed studies on the dynamics of dust events originating from these local sources remain limited. Consequently, the province has become one of the most affected regions in Iran by dust event, with notable impacts on air quality and public health [14].

## 1.2. Review of previous studies

A study investigated aerosol pollution sources and their optical properties during a 2018 dust event in northwestern China using various data sources and models, including MODIS, OMI, CALIPSO, and HYSPLIT. The research identified the Taklimakan Desert as the primary source of dust events in this region and highlighted the ecological sensitivity of desertification [15]. White et al. [16] used HYSPLIT modeling to analyze dust event trajectories in central Arizona, revealing that dust events often originate from southern regions with high particle concentrations from the southwest. Yousefi Kebriya et al. [17] analyzed dust indices in Sistan and Baluchestan (2018–2023) using MODIS data in Google Earth Engine and HYSPLIT backward trajectories. TDI rose from 0.0055 (2018–2019) to 0.0058 (2021), then fell to 0.0056 (2022) and 0.0057 (2023; min. threshold 0.018), identifying Zabol, Iranshahr, Khash, and Chabahar as key internal sources. AOD-Max peaked at 3.5 (2022, hazardous) and 3.1 (2023, very hazardous) in eastern/southern borders; AOD-Sum indicated >100 dusty days/year in east/northeast and >120 in south, averaging 1.9 (2023, hazardous). HYSPLIT revealed dust origins in Turkmenistan/Afghanistan deserts, reaching Zabol in 24 h (>1500 m to <500 m altitude) and Zahedan in 48 h.

Filonchyk et al. [18] investigated aerosol properties in Antarctica using AERONET data from seven stations and MERRA-2 reanalysis. The study analyzed aerosol optical depth (AOD), volume size distribution (0.09–0.76 μm), and direct aerosol radiative forcing (DARF). Results showed that aerosols are mainly natural (sea salt and marine sulfates), with AOD values ranging from $0.027 \pm 0.019$ to $0.082 \pm 0.042$, lowest at inland and highest at coastal sites. Surface radiative forcing indicated cooling effects ($-14.3 \pm 4$ to $-3 \pm 0.5$ W/m²), while atmospheric heating rates varied from $0.03 \pm 0.005$ to $0.25 \pm 0.03$ K.day$^{-1}$. The study underscores Antarctica as a key region for understanding aerosol impacts on climate and the importance of further research on aerosol sources and radiative effects. Sai Krishnaveni et al. [19] applied Fuzzy C-Means clustering to classify aerosols over Gadanki, India, using 26 optical and microphysical parameters from Sky Radiometer data (2008–2020). Three distinct clusters were identified mixed-mode moderately absorbing, mixed-mode slightly absorbing, and coarse-mode slightly absorbing aerosols with robust classification accuracy of ~98.4%, retaining ~94.5% of records under 10% random perturbation. Seasonal patterns showed biomass burning aerosols dominating pre-monsoon, polluted marine aerosols during the monsoon, and polluted continental aerosols in winter and post-monsoon. The clusters corresponded closely to global aerosol types, confirming the effectiveness of FCM for reliable aerosol classification.

Filonchyk et al. [20] analyzed sand and dust storms in northern China during March–April 2023, originating from the Gobi and Taklamakan deserts. Using satellite, ground-based, and reanalysis data, they found daily $PM_{10}$ concentrations exceeding 1000 μg/m³ and AOD values above 1 across most northern regions. Aerosol properties indicated dominance of coarse-mode particles, with radiative forcing ranging from $-48.5$ to $+2.7$ W/m² at the top of the atmosphere and $-180.8$ to $-66.6$ W/m² at the surface, producing atmospheric heating rates of 1.8–3.7 K day$^{-1}$. The storms were mainly driven by cold fronts associated with low-pressure systems, lifting dust into the atmosphere and transporting it downwind. Chen et al. [21]

quantified the contributions of major Asian dust sources to spring 2023 dust events in China using the WRF-Chem model. Results showed that Mongolia was the dominant contributor, accounting for ~56% (82.7 µg m$^{-3}$) of dust in North China and ~51% (15.9 µg m$^{-3}$) in the Korean Peninsula. In southwest China, ~46% of dust originated from Inner Mongolia, while dust over the Tibetan Plateau mainly came from Xinjiang due to topographic blocking. The study highlights the role of surface soil conditions in driving frequent dust events and emphasizes the need for regional cooperation in dust mitigation. Jain et al. [22] analyzed a pre-monsoon dust storm in May 2018 originating from the Thar Desert using satellite, reanalysis, and ground-based data. Dust column mass density increased 3–5 times during the event, reaching ~3.9 g m$^{-2}$ compared to background levels of ~0.2–0.6 g m$^{-2}$, with AOD > 2 indicating prolonged dust residence. HYSPLIT trajectories confirmed transport toward the Indo-Gangetic Plains, and strong associations with $PM_{10}/PM_{2.5}$ and meteorological variables highlighted severe air quality impacts.

Zhang et al. [23] investigated urban dust event in Lanzhou, China, using field observations and two theoretical models: the dust passing model and the point-source diffusion model. The results classified urban dust into external-source and local-source dust. External dust originates from the Badain Jaran and Tengger Deserts and is transported to Lanzhou through the Qilian and Helan Mountains, consistent with the HYSPLIT model results. Local dust originates from four bare ground areas upstream of the observation site and propagates along the Yellow River from northwest to southeast. Forecasting methods include HYSPLIT forward trajectory analysis for external dust and monitoring three-dimensional electric field anomalies over upstream bare ground for local dust. Shirgholami et al. [24] examined dust event trends in Yazd province, Iran (2003–2022), using synoptic station data (~0.5 dusty days/year increase, peaking March–May) and MODIS AOD (rising annual load, May maximum). Central/northern regions were most affected, driven by drought, desertification, wetland drying, land use changes, and transboundary dust from Iraq/Syria/Saudi Arabia. Soleimani Sardo et al. [25] investigated a December 16–20, 2016 dust event in Iran using MODIS AOD, WRF-Chem, and HYSPLIT models. HYSPLIT forward trajectories showed particles from Jazmurian wetland moving southward, with AOD peaking at 0.8 in the region. Significant AOD increases occurred in southern Sistan-Baluchestan, Afghanistan-Pakistan border, west Oman Sea, and east Persian Gulf, highlighting Jazmurian's emerging role as a major fine-dust source due to reduced wetland inflow from climatic and anthropogenic factors. Rangzan and Balouei [26] studied dust events in Khuzestan Province, Iran (2013–2022) using MODIS AOD as the main dust indicator. Vegetation cover (NDVI), annual rainfall, and topography (elevation, slope, slope direction from 90 m SRTM DEM) were included as environmental factors. Vegetation indices were averaged annually from ~24 satellite images per year via Google Earth Engine. Dust analysis focused on June–September, the peak dust season. Results showed highest dust levels in 2015–2018, with southwestern Khuzestan reaching a maximum AOD of 2.82 in 2013–2014. Elevation (r = 0.83) and slope direction (r = 0.72) were the strongest predictors, with higher elevations reducing dust and south-facing flat areas increasing it. The study emphasizes MODIS AOD and remote sensing as effective tools for monitoring and managing dust events in the region. Vatanparast Ghaleh Juq et al. [27] analyzed dust variations in western Iran using MODIS (2012–2021) and Sentinel-5 (2018–2021) data. Aerosol levels (AOD, AI) showed lowest values in January 2021 and highest in July–September 2021, concentrated in Khuzestan and Ilam. Aerosols were mostly over bare soils and low-vegetation areas, emphasizing the role of vegetation and soil management in dust mitigation. Arami et al. [28] assessed dust event risk in Khuzestan Province (64,057 km²) from 1995–2022 using hourly station data and GIS-based interpolation. Abadan had the highest frequency: 18 days/year. Most events occurred 9:30 am–6:30 pm (61%), and 79% originated from extra-regional sources. Over 88% of the province was classified as high or very high risk. Farzanehpey et al. [29] examined dust trends in Khuzestan Province using monthly AOD, LST, and air temperature (13 stations). Dust trends were analyzed with the Mann-Kendall test, and correlations between AOD and temperature parameters were assessed in 5–20 km buffers. AOD showed >90% increasing trend in Aug–Sep–Nov at most stations. Strong correlations with LST occurred mainly in May–July, while air temperature had mostly very weak correlations. Results indicate LST is a better predictor of dust intensity than air temperature.

Salmabadi et al. [30] investigated the major dust sources affecting Ahvaz, Iran, by analyzing HYSPLIT model data from winter 2015 to fall 2017. Their findings indicated that deserts in Iraq, Iran, Saudi Arabia, and Kuwait were significant contributors to PM concentrations in Ahvaz. This study highlights the transboundary nature of dust transport in western Iran and emphasizes the role of regional dust sources in air pollution. Ebrahimi-Khusfi et al. [31] examined how fluctuations in wetland water area (WWA) affect air pollution in Iranian cities. They found that shrinking wetlands, especially Hamoun and Parishan, significantly contribute to dust event in nearby cities. Strong negative correlations between dust event and WWA were observed in winter and spring (e.g., Shiraz-Parishan, r = −0.33; Zabul-Hamoun, r = −0.32). Annually, reductions in WWA accounted for 25% of the dust event changes around these wetlands, highlighting the need for effective wetland management to reduce air pollution. Gammoudi et al. [32] examined the transport and deposition of Saharan dust in Central Europe by analyzing atmospheric dust dispersion patterns using the HYSPLIT model and GDAS meteorological data. Their findings revealed a seasonal variation in dust transport, with peak dust events occurring in spring and a secondary peak in summer. The study highlighted that the most dust-laden air masses originate from North Africa and frequently reach Central Europe, following different pathways depending on seasonal and climatic conditions. Additionally, the frequency and intensity of Saharan dust events (SDEs) have increased in recent decades, particularly in winter, due to variations in Mediterranean cyclone activity and climate change-induced lee-side cyclogenesis near the Atlas Mountains. Endale et al. [33] conducted a study to assess particulate matter ($PM_1$, $PM_{2.5}$, and $PM_{10}$) over Dire Dawa in May 2021 using Purple Sensor (PS) for real-time measurements and the Gravimetric Method (GM) for mass-based analysis. The results indicated that $PM_{10}$ levels were within acceptable limits, whereas $PM_{2.5}$ levels suggested moderate pollution. The PS recorded higher $PM_{10}$ concentrations compared to GM. Analysis of $PM_1/PM_{10}$ and $PM_{2.5}/PM_{10}$ ratios suggested the dominance of coarse particles, with fine and coarse particles being equally present in some cases. Spatial variations were observed based on filter placement. HYSPLIT backward trajectory analysis identified multiple air mass transport pathways, highlighting both urban and desert dust contributions. The study by Alainejad et al. [34] investigates dust event emissions, trajectory tracking, and synoptic analysis in Ilam city, located in western Iran. Using HYSPLIT model simulations, MODIS imagery, and MERRA-2 data, the research identifies key sources of dust emissions and tracks the movement of dust episodes. The study recorded 165 dust events between 2012 and 2018 and found a strong negative correlation (Spearman's coefficient of −0.66) between inter-annual visibility and $PM_{10}$ concentration. The study determined that dust emission sources in Ilam originated primarily from Iraq (38%), Syria (22%), Saudi Arabia (17%), and local sources in Khuzestan Province (4%). Additionally, the synoptic analysis revealed that low-pressure cyclones and Shamal winds, originating from the northwest, play a significant role in dust event occurrence. Alzaid, et al. [35] Dust event simulation and source apportionment using the HYSPLIT model in Saudi Arabia's eastern region. The study applied the HYSPLIT model to simulate dust events in Saudi Arabia, achieving a high level of accuracy ($R^2 = 0.9965$) by calibrating and validating the model with optimized parameters. The study identified Iraq and Syria as the primary sources of severe dust events in the region. Noroozi and Shoaei [36] investigated dust generation potential in Khuzestan, Iran using statistical methods, remote sensing, and modeling. Results indicated that the most frequent dust events occurred between 2008–2009, primarily in late spring and early summer (May–July). Satellite AOD and BTD analyses showed that 80% of dust events were concentrated in the west and southwest of the province.

## 1.3. Objectives and innovations of the study

This study presents an integrated and multidisciplinary framework for analyzing dust events and identifying their environmental drivers in Khuzestan Province, Iran. The approach leverages a novel combination of multi-satellite data, advanced statistical modeling, and atmospheric trajectory analysis to simultaneously identify local and transboundary dust sources. In contrast to previous studies, which largely relied on a single satellite dataset or descriptive analyses, the present research utilizes data from MODIS, Sentinel-5, Sentinel-2, and Jason-2 to provide a comprehensive and multi-sensory representation of environmental conditions. AOD and AAI were employed to assess the intensity and spatial distribution of

suspended particles, while MNDWI and Water Level were used to monitor wetland shrinkage and its relationship with dust events. To enhance the temporal and spatial characterization of aerosols, two new indices were introduced: AOD-Max and AOD-Sum, which capture peak intensity and cumulative dust load over time, respectively. By integrating these indices with HYSPLIT back-trajectory simulations, MODIS true-color imagery, trajectory clustering (Clusters), and WPSCF/WCWT analyses, the study provides a precise identification of dust transport pathways and quantification of transboundary contributions. Additionally, a 24-hour lag correlation analysis between AOD values in neighboring countries and Ahvaz city quantifies the impact of transboundary dust transport—a dimension largely overlooked in regional studies. The research further applies a comprehensive suite of quantitative and statistical analyses, including R², linear regression, Pearson and Spearman correlation, mutual information, and K-Means clustering. This rigorous analytical framework addresses a critical gap in previous studies, which often lacked robust quantitative validation. By integrating multi-sensory satellite data, atmospheric trajectory modeling, and extensive statistical analyses, this study provides novel and actionable insights into the complex interactions between dust events and wetland shrinkage. The proposed approach not only improves the accuracy of dust source identification but also offers a replicable and practical methodology for arid and transboundary regions worldwide. The methodology is summarized in the flowchart presented in Fig 1.

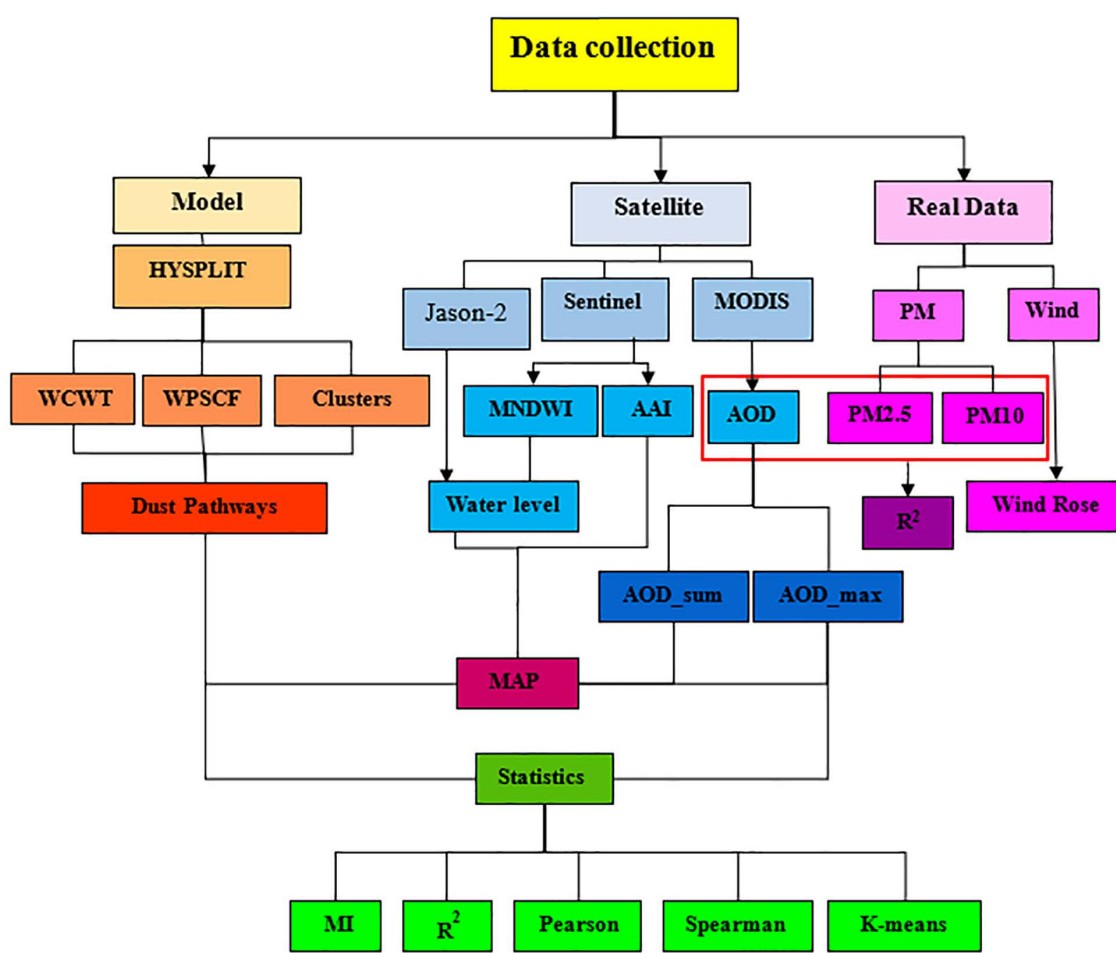

**Fig 1. Step-by-Step Process of Dust Source Analysis Using Satellite Data and HYSPLIT Model.**

## 2. Materials and methods

### 2.1. Study area

Khuzestan Province is located between 29.95° to 33.55° N and 47.66° to 50.55° E, covering 9.3% of Iran, as shown in Fig 2. The province includes diverse landscapes, from plains to mountainous areas, and contains major wetlands such as Shadegan and Hoor al-Azim, which are significant sources of dust in the region. Hoor al-Azim Wetland, situated in the central-southern part of the province, strongly influences local environmental and hydrological conditions. The climate in this province is predominantly hot desert, although the northern and northeastern regions, including the foothills of the northern Zagros Mountains, experience more climatic diversity, with warm and dry to semi-humid oceanic climates observed in these areas [37].

### 2.2. Climate and satellite data

This research utilized data on particulate matter (PM$_{2.5}$ and PM$_{10}$) for a 6-year period (2018–2023) from 22 air pollution monitoring stations in Khuzestan Province. To assess the impact of dust events originating from Iraq on Khuzestan Province particularly on the city of Ahvaz 25 virtual stations were designed and positioned outside Iran's western borders, in

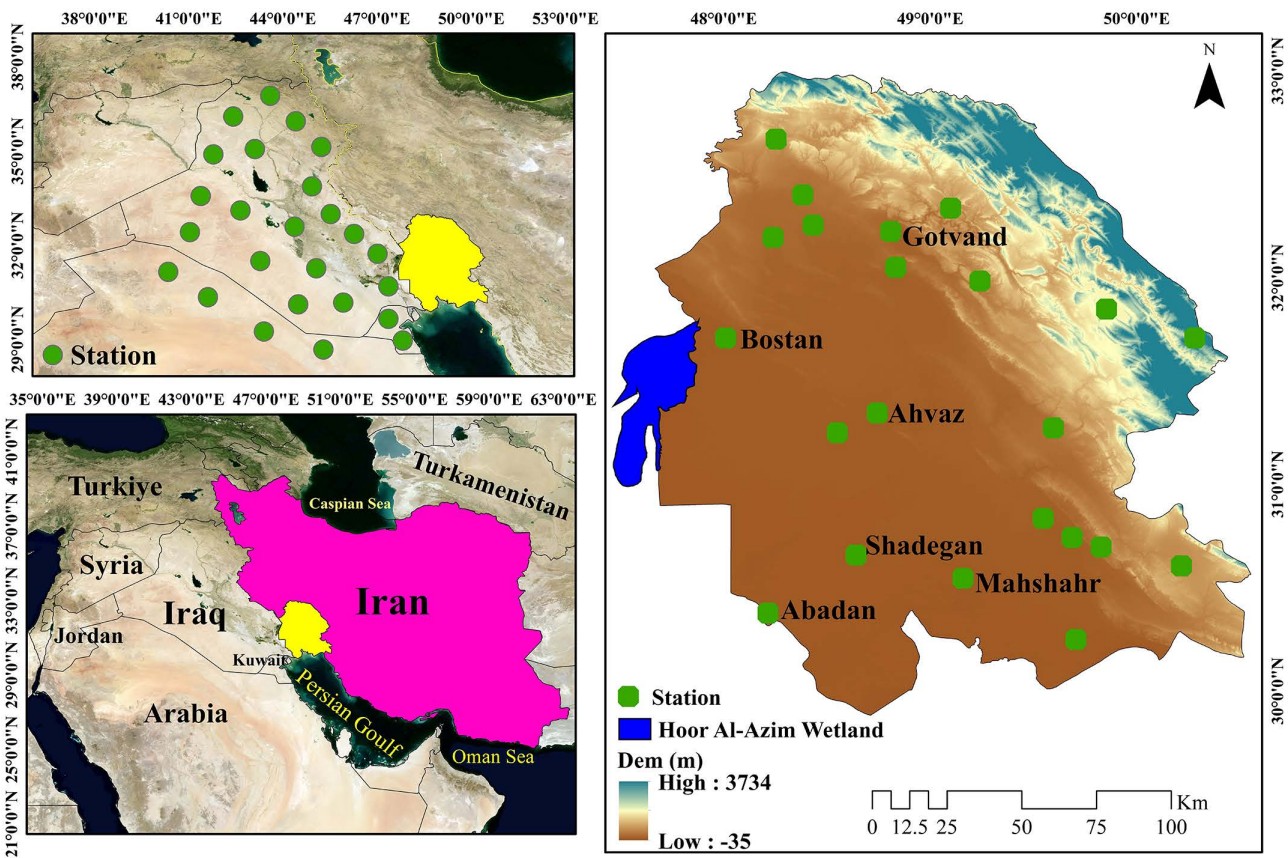

Source: NASA EOSDIS Worldview (MODIS Terra/Aqua true color imagery, 2024). Processed and visualized by the authors.

**Fig 2. Location and topographic map of Khuzestan Province and the Shadegan Wetland.** Data were obtained from NASA EOSDIS Worldview (https://worldview.earthdata.nasa.gov/) using MODIS Terra/Aqua true color imagery (2024). The visualization and processing were performed by the authors.

Iraq and its surrounding areas, to examine the 24-hour lag effect and its influence on dust transport toward Khuzestan. Among the real stations, data from Shadegan, Ahvaz, Abadan, Bostan, Gotvand, and Mahshahr were used to generate wind roses and analyze wind direction and speed patterns to better understand pollution dispersion, as shown in Fig 2.

Dust event data were obtained from the Iranian Air Quality Control website, while the satellite datasets included AOD, AOD-Max, and AOD-Sum from MODIS, MNDWI from Sentinel-2, AAI from Sentinel-5, water level (wetland water depth) data from Jason-2, and MODIS Corrected Reflectance (True Color) imagery to investigate the spatial and temporal behavior of dust events. All images and datasets were extracted using coding in the Google Earth Engine platform, a process that involved scripting to compile image collections and extract the required information both regionally and at specific points.

The HYSPLIT model was then applied along with data from days exhibiting varying dust indices to track dust transport trajectories and evaluate their impacts on air quality at the provincial scale. In this study, a total of 220 dust transport trajectories were analyzed for the year 2022, selected as a representative year characterized by high dust pollution levels. These trajectories were generated using HYSPLIT driven by GLDAS meteorological data at a 1° spatial resolution and a 3P configuration, and were computed as backward trajectories. Subsequently, a trajectory clustering procedure was applied to identify the dominant dust transport patterns.

Overall, 17 AOD-related images were analyzed, including 7 AOD-Max, 5 AOD-Sum, and 5 AOD images. Additionally, 5 AAI images from Sentinel-5 and 12 MNDWI images from Sentinel-2 were utilized. Monthly MODIS AOD data, Sentinel-2 MNDWI datasets, and water level data from Jason-2 were incorporated into the analysis over a five-year period. Furthermore, 2 MODIS True Color images and 4 trajectory maps derived from HYSPLIT were employed to support the interpretation of spatial patterns of dust events.

### 2.3.. AOD max and AOD sum

In this study, two indices derived from MODIS (Moderate Resolution Imaging Spectroradiometer) satellite data are employed to analyze AOD over Khuzestan Province. MODIS is a multispectral sensor installed on NASA's Terra (launched in 1999) and Aqua (launched in 2002) satellites, operating under the Earth Observing System (EOS) program. It provides near-daily global coverage with spatial resolutions of 250 m, 500 m, and 1 km, depending on the spectral band. MODIS data are widely used for monitoring aerosols, land surface changes, and atmospheric properties due to their high temporal frequency and broad spectral range (36 spectral bands ranging from 0.4 to 14.4 μm). The sensor's AOD products typically carry an uncertainty of $\pm (0.05 + 0.15 \times AOD)$ over land surfaces, which is considered acceptable for regional and global aerosol monitoring. For this research, the MODIS MCD19A2 (Collection 6) dataset was used, which provides daily AOD observations derived from the Multi-Angle Implementation of Atmospheric Correction (MAIAC) algorithm. The time series spans from 2018 to 2022, providing consistent data for trend and variability analysis. In this methodology, the band Optical_Depth_055 was selected. The images were clipped to the boundaries of the specified geographic area [28].

To analyze AOD variations, two statistical indices were computed: AOD Max and AOD Sum. AOD Max represents the maximum aerosol optical depth observed for each pixel over a specific period, such as annually, allowing the identification of areas experiencing the highest peak dust concentrations and greatest environmental risk. To reduce noise and remove isolated outlier pixels in MODIS AOD before calculating AOD Max, spatial filtering was applied using 3×3 median, mean, and Gaussian filters. All images (Optical_Depth_055 band, MCD19A2_GRANULES) were clipped to the study area to ensure robust and reliable maximum values. In contrast, AOD Sum represents the cumulative aerosol loading over the defined period, providing insights into the total burden of dust on ecosystems and populations. These indices are particularly suitable for capturing extreme dust events, which may not be reflected in conventional statistics like mean or median, as these measures tend to smooth out short-term but highly impactful pollution episodes. Together, these indices enable a comprehensive understanding of both peak and aggregated dust impacts, making them more appropriate than average-based metrics for assessing regions most affected by dust, including those near border areas, and for identifying

transboundary dust transport corridors. By utilizing both indices, the study not only captures the most severe dust events but also evaluates the overall exposure in Khuzestan Province, thereby offering a detailed perspective on both short-term health risks and long-term environmental consequences.

## 2.4. Air quality index

The Air Quality Index (AQI) is measured daily or even hourly, assessing the concentration of each air pollutant. The concentration of each pollutant is measured and converted to a numerical value from zero to 500 using a standardized scale or index. The AQI is a relative measure, where lower values indicate better air quality and less concern for health, and higher values indicate poorer air quality and increased health risks [38]. The AQI is determined using the AOD index, which measures the absorption and scattering of light by suspended particles (aerosols) in the atmosphere. AOD is a dimensionless index that quantifies the extinction of solar radiation due to aerosols in the atmosphere. Additionally, the AQI incorporates the AAI, a qualitative index indicating the presence of elevated layers of aerosols that absorb UV radiation. Higher AAI values correspond to more polluted air. For both AOD and AAI data, spatial filtering including median (3×3 pixels), mean (3×3 pixels), and Gaussian smoothing was applied to reduce noise, remove outliers, and correct isolated artifacts across the study area. Table 1 shows the relationship between AOD, AAI, and their impact on AQI. In this research, the Air Quality Index (AQI) based on particulate matter (PM) was employed to investigate the correlation between real-time data and satellite data [39]. Table 1 further details the classification of AQI values and their corresponding health risks.

## 2.5. Dust transport indices and analysis framework

To identify the dominant atmospheric transport pathways and potential source regions influencing the study area, a combination of Lagrangian air-mass trajectory analysis and grid-based statistical metrics—including trajectory clustering, the WPSCF, and the WCWT was applied. These approaches enable the simultaneous characterization of prevailing airflow regimes, the spatial probability of source regions, and the relative intensity of their contributions to the observed dust load.

   2.5.1. HYSPLIT.  Air-mass trajectories were simulated using the Hybrid Single-Particle Lagrangian Integrated Trajectory (HYSPLIT) model developed by the National Oceanic and Atmospheric Administration (NOAA, USA) and implemented through the NOAA HYSPLIT READY Archive platform. The model applies a Lagrangian particle dispersion framework, solving the equations of motion for individual particles to represent advection, turbulent diffusion, and deposition processes, thereby enabling the simulation of both dust transport pathways and dispersion behavior. Meteorological input fields were obtained from the Global Data Assimilation System (GDAS) with a spatial resolution of 0.25°×0.25°, incorporating wind speed and direction, temperature, pressure, and humidity at regional and local scales. Forward and backward three-dimensional trajectories with a 72-hour integration time were generated to identify potential dust source regions, dominant transport corridors, and receptor areas affected by dust-laden air masses [40].

**Table 1. Air quality index Basics Pollution [38,39].**

| Levels of Concern | Values of Index AAI | Values of Index AOD | Values of Index AQI |
|---|---|---|---|
| Air quality is satisfactory, and air pollution poses little or no risk. | <0 | <0.1 | 0-50 |
| Air quality is acceptable. However, there may be a risk for some people, particularly those who are unusually sensitive to air pollution. | 0-0.5 | 0.1-0.49 | 51-100 |
| Some members of the general public may experience health effects; members of sensitive groups may experience more serious health effects. | 0.5-2 | 0.5-1 | 101-200 |
| Health alert: The risk of health effects is increased for everyone. | 2-5 | >1 | 201-300 |
| Health warning of emergency conditions: everyone is more likely to be affected. | >5 | – | >300 |

**2.5.2. Trajectory clustering.** Trajectory clustering was employed to reduce the complexity of the large ensemble of individual air-mass pathways and to extract dominant transport patterns. In this approach, trajectories were grouped based on spatial similarity and geometric convergence, such that within-cluster spatial variance was minimized while between-cluster separation was maximized. Each resulting cluster represents a characteristic synoptic-scale airflow regime and is interpreted as a dominant dust transport pathway. The relative frequency of each cluster was used to quantify the contribution of different flow directions to dust occurrence under varying pollution levels [20].

**2.5.3. WPSCF.** The WPSCF method was applied to estimate the spatial probability of different grid cells acting as potential source regions during high dust loading events. The study domain was divided into a regular grid, and for each cell, the ratio of the number of trajectory endpoints associated with dust concentrations exceeding a predefined threshold to the total number of trajectory endpoints passing through that cell was calculated. To reduce statistical uncertainty in cells with limited trajectory counts, a weighting function was applied, thereby enhancing the robustness and reliability of the spatial patterns. Higher WPSCF values indicate regions with a greater likelihood of contributing to elevated dust levels at the receptor site [18].

**2.5.4. WCWT.** The WCWT approach was used to quantify the relative intensity of contributions from different source regions. In this method, each grid cell is assigned a concentration-weighted value based on the mean observed dust concentration or dust-related index at the receptor site at the times when air-mass trajectories traversed that cell. Unlike WPSCF, which emphasizes source probability, WCWT ranks regions according to their relative impact on the magnitude of dust loading. Cells with higher WCWT values are interpreted as stronger contributors to high-concentration dust events [18].

All simulated trajectories and outputs derived from trajectory clustering, WPSCF, and WCWT analyses were imported into the MapInfo geographic information system for spatial processing, grid generation, and cartographic visualization. This integrated analytical framework enables the simultaneous assessment of both the frequency and intensity of dust transport mechanisms at the annual scale under varying pollution conditions.

## 2.6. Wind

Wind is a critical meteorological factor influencing the emission and dispersion of dust. It plays a significant role in transporting dust from dry, uncovered areas to other regions. This effect is particularly pronounced in areas with loose, weakly-structured soils, deserts, and regions lacking sufficient vegetation cover to mitigate dust spread. In this study, wind speed and wind direction were measured and analyzed to understand their impact on dust dispersion. Wind data were collected and analyzed using the WRPLOT View software, which was utilized to generate wind rose plots. These plots visually represent the distribution of wind speed and direction, aiding in the assessment of how wind contributes to dust transport, elevation, and its subsequent effects on air quality, health, and the environment [40].

## 2.7. MNDWI

The MNDWI is commonly used to identify water bodies in satellite and remote sensing imagery. This index leverages the difference between the Green and Shortwave Infrared (SWIR) bands to effectively distinguish water from other features, such as soil and vegetation. MNDWI is calculated using Equation (1), where Green represents the reflectance intensity in the green spectral band, and SWIR represents the reflectance intensity in the Shortwave Infrared spectral band. Typically, water bodies yield positive MNDWI values, whereas other features like soil and vegetation are associated with negative values. In summary, positive MNDWI values indicate the presence of water bodies, while negative values correspond to non-water features [41]. It is calculated using the following equation (2). where "Green" refers to the green band in the range of 525−602 nm, and "NIR" denotes the near-infrared band with a wavelength of 750−900 nm. A threshold value of

0.3 is commonly used to define water in the NDWI index [38]. In this study, MNDWI was computed for each image in the Sentinel-2 collection, and a median composite was generated over the study area to reduce noise and improve robustness, following the algorithm implemented in Google Earth Engine.

$$\text{MNDWI} = \frac{\text{Green} - \text{SWIR}}{\text{Green} + \text{SWIR}} \tag{1}$$

## 2.8. Water level

Water Level (wetland water depth) is commonly used to monitor hydrological dynamics in wetlands. Water Level represents the actual depth of water in the wetland, measured relative to a reference geoid (EGM2008) and the Jason-2 satellite reference pass [42]. In this study, Water Level data were obtained from the Jason-2 satellite altimetry product TPJOJS.2, which provides smoothed measurements of water surface height. Invalid or missing measurements (e.g., 999.99 or 9999.99) were excluded from the analysis.

## 2.9. Statistical analysis

To provide a clear framework for evaluating the relationships among the study variables, key statistical indices were selected to quantify linear and non-linear associations. These indices offer a rigorous basis for interpreting the dependencies and interactions present in the dataset.

**Coefficient of Determination (R²):** Coefficient of Determination (R²): Measures how well independent variables explain the variability of the dependent variable. Values near 1 indicate strong explanatory power, while lower values suggest other influencing factors [43].

**Pearson Correlation:** This coefficient measures the linear association between two variables, providing insight into their co-variation and direction of influence [44].

**Spearman Rank Correlation:** By ranking values, this non-parametric method captures monotonic relationships, making it robust for non-linear datasets [45].

**Mutual Information (MI):** MI measures shared information between variables based on entropy. Higher MI values indicate stronger dependencies, capturing both linear and non-linear interactions [46].

**K-Means Clustering:** This unsupervised approach partitions data into K clusters based on similarity. The algorithm iteratively assigns data points to centroids and recalculates their positions until convergence. Techniques such as the Elbow Method and Silhouette Score help identify the optimal number of clusters [47].

In this study, satellite imagery and indices from MODIS, including AOD Max and AOD Sum, were used to analyze dust events in Khuzestan Province. AOD Max represents peak aerosol concentrations, while AOD Sum provides an estimate of cumulative dust load over time, enabling identification of high-pollution areas. The HYSPLIT model was applied to track dust trajectories and identify source regions, complemented by MODIS True Color images, trajectory clustering, and WPSCF/WCWT maps. Wind data from meteorological stations were incorporated to evaluate transport pathways. A 24-hour lag correlation analysis was performed between AOD values in Khuzestan and neighboring regions, focusing on Ahvaz, to assess delayed transboundary dust effects. To examine local sources and wetland dynamics, MNDWI from Sentinel-2 and water level measurements from Jason-2 were analyzed monthly for Hoor Al-Azim Wetland. The relationship between wetland area changes and AOD indices was explored to evaluate the influence of wetland shrinkage on dust activity. Finally, the spatial distribution of dust events over 2018–2022 was mapped using AOD and AAI from MODIS and Sentinel-5, providing a detailed overview of dust activity patterns and highlighting correlations with wetland area variations.

## 3. Results and discussion

### 3.1. Validation of satellite data with ground-based measurements

The analysis based on 798 paired AOD–PM$_{2.5}$ observations from 2020 to 2022 revealed a statistically significant linear relationship between satellite-derived AOD and ground-measured particulate matter concentrations, as illustrated in Figs 3 and 4. For PM$_{10}$, the regression model PM$_{10}$ = −32.51 + 623.9 × AOD produced a coefficient of determination

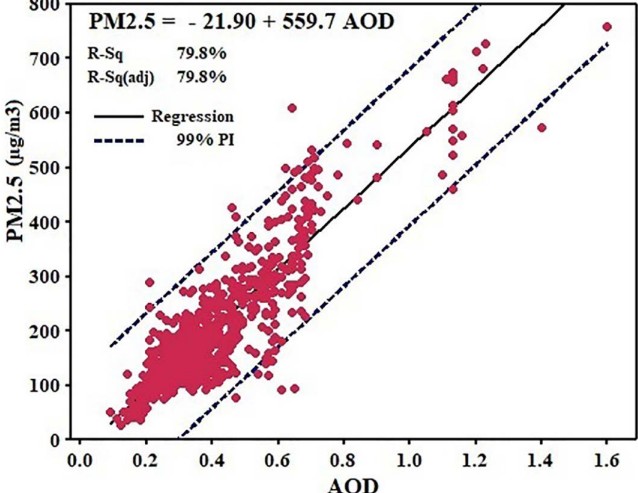

(Derived from MODIS satellite data (NASA EOSDIS) and ground-based observations)

**Fig 3. Regression analysis chart of AOD and PM$_{2.5}$ for the year 2020–2022.** Source: Derived from MODIS satellite data (NASA EOSDIS) and ground-based observations; processed and analyzed by the authors.

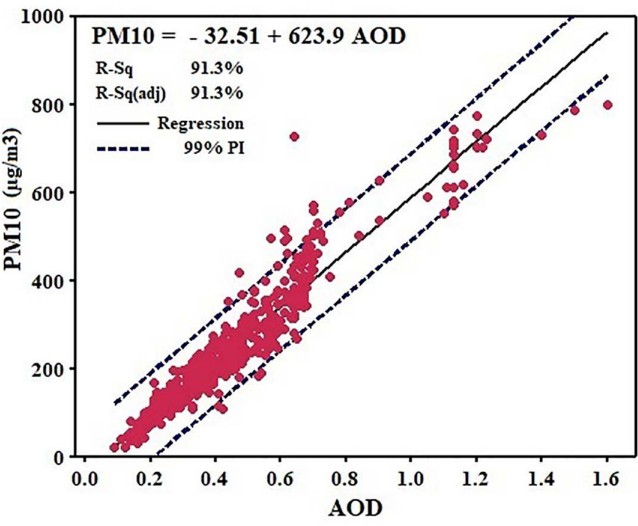

(Derived from MODIS satellite data (NASA EOSDIS) and ground-based observations)

**Fig 4. Regression analysis chart of AOD and PM$_{10}$ for the year 2020–2022.** Source: Derived from MODIS satellite data (NASA EOSDIS) and ground-based observations; processed and analyzed by the authors.

of R² = 0.913 (adjusted R² = 0.913), indicating that approximately 91% of the variation in $PM_{10}$ concentrations can be explained by changes in AOD. The regression slope (623.9) implies that a 0.1-unit increase in AOD corresponds, on average, to a rise of about 62 μg/m³ in $PM_{10}$ levels. For $PM_{2.5}$, the equation $PM_{2.5} = -21.90 + 559.7 \times AOD$ yielded R² = 0.798 (adjusted R² = 0.798), meaning that about 80% of the $PM_{2.5}$ variability is explained by AOD. The slope (559.7) suggests that a 0.1-unit increase in AOD leads to an estimated 56 μg/m³ increase in $PM_{2.5}$ concentration. In both models, the 99% prediction intervals (PIs) encompassed almost all observed data points. Specifically, only 28 out of 798 observations (3.0%) for $PM_{10}$ and 23 out of 798 observations (2.9%) for $PM_{2.5}$ fell outside the 99% prediction limits. Consequently, approximately 97% of all observed values were captured within the models' confidence bounds, indicating high temporal stability and satisfactory predictive performance. The comparison between the two models shows a stronger and more sensitive association between AOD and $PM_{10}$ (R² = 91.3%) than with $PM_{2.5}$ (R² = 79.8%). This suggests that AOD is more representative of course-mode particles, which dominate in arid and dust-prone regions. Within the dataset, $PM_{10}$ concentrations ranged from near 10 to over 800 μg/m³, while $PM_{2.5}$ values varied between 12 μg/m³ and 750 μg/m³, and both regression models maintained satisfactory predictive accuracy across these ranges. Overall, the results confirm that AOD can serve as a reliable predictor of particulate matter, especially $PM_{10}$, in environments affected by dust storms or limited air quality monitoring infrastructure. This highlights the potential of satellite-based AOD observations as an effective complementary approach for air quality assessment, dust monitoring, and environmental management in arid regions.

## 3.2. AOD max

For the annual evaluation of the AOD-Max index, daily MODIS satellite data comprising 360 images per year were integrated to generate composite maps that represent the maximum values of the index throughout each year. The analysis of these maps, presented in Figs 5 and 6, reveals distinct spatiotemporal variations across Khuzestan Province during the study period.

In 2018, the northwestern regions of the province, particularly along the main dust-entry pathways from Iraq and neighboring provinces, recorded the highest AOD-Max values ranging from 3.2 to 3.9, placing them in the hazardous category.

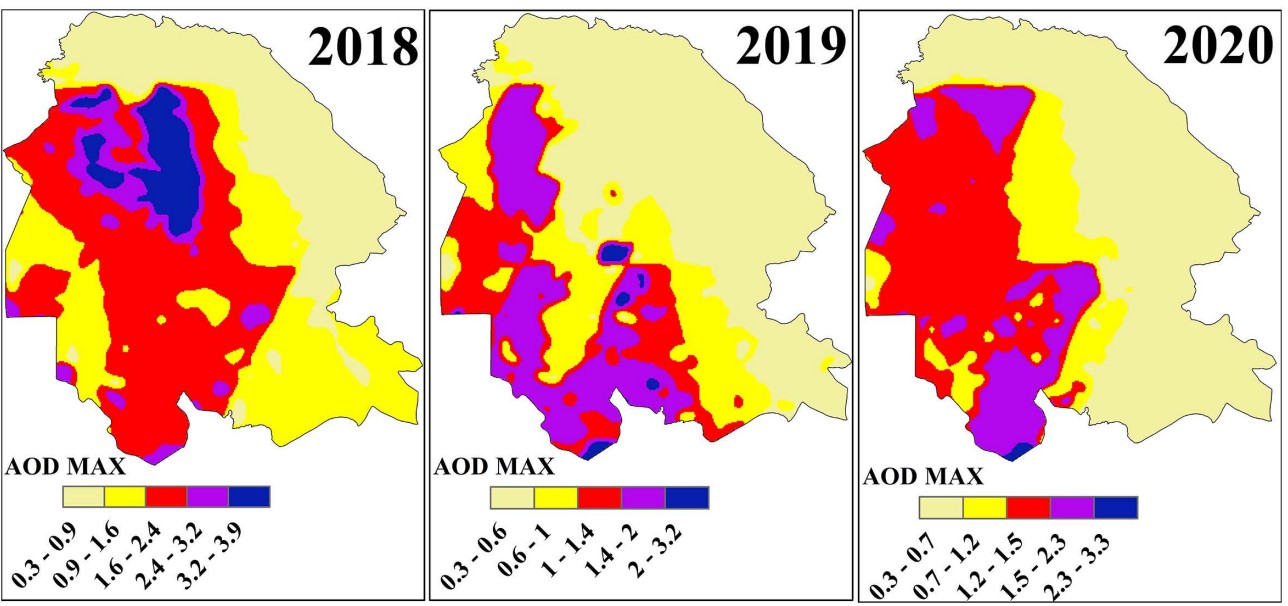

**Fig 5. Spatial Distribution of AOD Max over Khuzestan Province from MODIS Satellite during the period 2017–2020.**

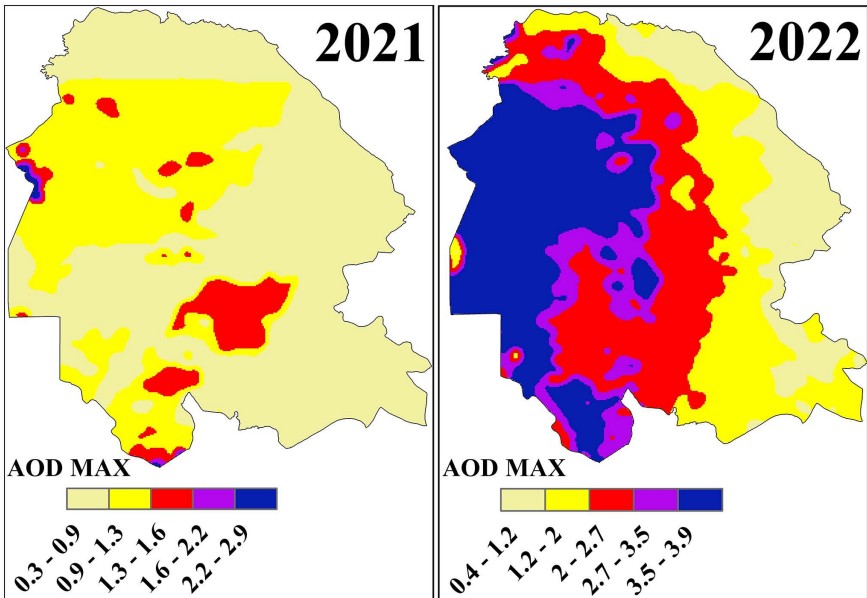

Source: MODIS (NASA EOSDIS). Processed and analyzed by the authors.

**Fig 6.  Spatial Distribution of AOD Max over Khuzestan Province from MODIS Satellite during the period 2021–2022.**

Other western areas reported values between 1.6 and 2.4. A notable pattern was observed around the wetlands of Khuzestan, where AOD-Max values ranged from 0.9 to 2.4, indicating that desiccated zones lost their natural filtration capacity, whereas water-rich areas with preserved wetland functions maintained the lowest dust indices. In contrast, the eastern and northeastern regions situated along the Zagros Mountains, characterized by higher elevation and rangeland cover, exhibited negligible levels of dust event.

In 2019, although the overall AOD-Max values decreased, the highest class ranged between 2.0 and 3.2 and encompassed most of the western, northwestern, southern, southwestern, and central regions of the province, including the city of Ahvaz, which recorded an index of 3.2. A critical observation for this year was related to the Shadegan Wetland, where three distinct levels of AOD-Max were identified: water-covered zones with values of 0.6–1.0 demonstrated strong dust filtration capacity; transitional areas undergoing desiccation with values of 1.0–1.4 reflected reduced filtering efficiency; and permanently dried zones with values of 1.4–2.0 emerged as active dust sources. This gradient underscore the vital role of wetlands in dust mitigation, their progressive loss of function, and their eventual transformation into dust-emitting sources. In 2020, the AOD-Max index increased once again, with the highest class ranging from 2.3 to 3.3. As in previous years, the northwestern, western, and southwestern regions recorded the most hazardous levels, while the spatial extent of polluted areas also expanded compared to 2019, with large portions of western Khuzestan classified as unhealthy, marked by values around 1.5.

In 2021, the reliability of the results was reduced due to noise and technical limitations in satellite imagery, which affected the annual composite map since it was derived from stacking and averaging 360 daily images. In 2022, conditions reached a critical level as severe drought, lack of rainfall, extensive wetland shrinkage, and intensified dust intrusions from neighboring countries culminated in the most extreme dust event of the study period [17]. The highest AOD-Max values, ranging from 3.5 to 3.9, were concentrated in the northwestern and western regions, especially along the major dust transport corridors from Iraq. The spatial extent of dust-affected areas expanded dramatically, sparing only the highland zones, while southern regions recorded unhealthy conditions with values between 2.0 and 2.7. Around the

Shadegan Wetland, values ranged between 2.0–2.7 and 2.7–3.5, highlighting both unhealthy and hazardous conditions and reflecting the dual role of desiccated wetlands as major dust sources.

Areas with elevated AOD-Max values are characterized by extremely poor air quality, leading to significant public health risks such as respiratory and cardiovascular diseases, while environmental consequences include reduced visibility, disruption of agricultural practices, and widespread dust deposition affecting both urban and rural landscapes. The spatial variability captured by the AOD-Max maps provides a valuable basis for policymakers and environmental agencies to identify and prioritize dust mitigation zones, to design region-specific interventions, and to raise public awareness by issuing targeted health advisories and protective measures. Collectively, the results underscore the heterogeneity of dust event in Khuzestan Province and highlight the urgent necessity for targeted and sustained strategies to mitigate the adverse impacts of dust events on both human and environmental systems.

### 3.3. AOD sum

In this study, the annual AOD-Sum index was employed to analyze the intensity and frequency of dust events in Khuzestan Province. This index was derived by integrating the cumulative sum of daily MODIS satellite observations (360 images per year), resulting in a composite that represents the total annual aerosol load associated with dust events. Unlike mean- or point-based metrics, AOD-Sum captures both the recurrence and intensity of dust events over time and space, effectively serving as a proxy for the total number of "dust days" or cumulative dust exposure. The spatial and temporal patterns of AOD-Sum, as shown in Figs 7 and 8, highlight substantial variations and clearly reveal regions experiencing frequent and intense dust events throughout the study period. In 2018, the AOD-Sum values ranged from 0 to 150, with the northwestern, western, southwestern, central, and southern regions of the province experiencing the highest index values (150 days with dust events). Around the Shadegan Wetland, four distinct classes were identified: (i) water-covered areas of the wetland, which recorded no dust events; (ii) moist peripheral zones, which experienced fewer than 10 dusty days; (iii) transitional desiccating zones, with 60–70 dusty days, primarily located in the southwestern, northern, and parts of the southern wetland; and (iv) permanently desiccated zones in the southeastern and eastern parts of

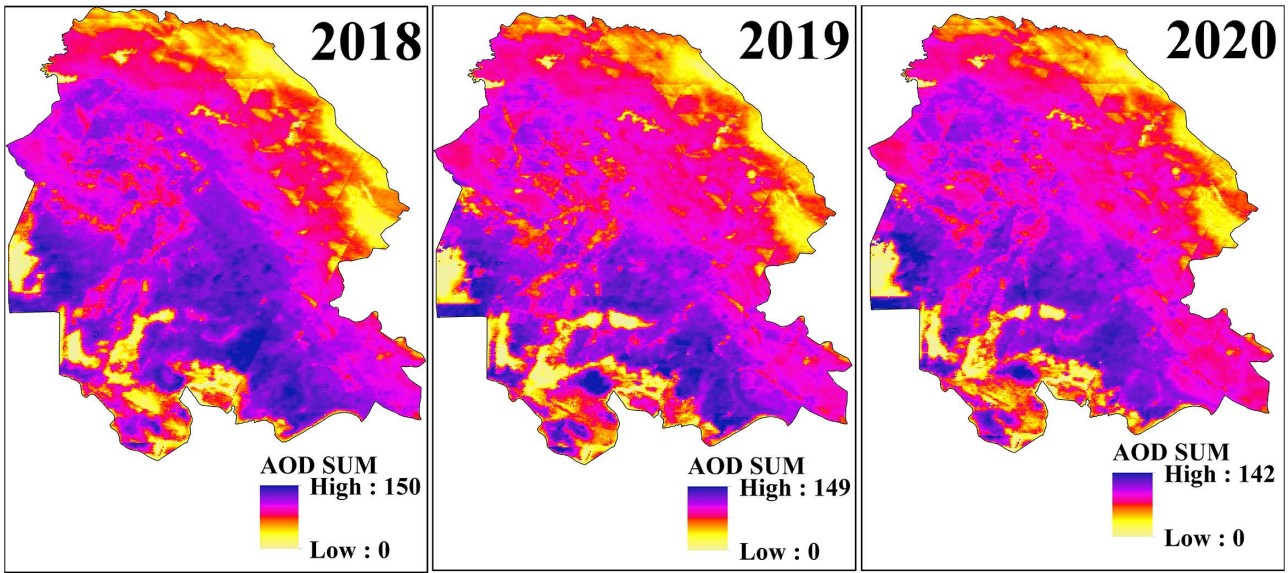

**Fig 7. Spatial Distribution of AOD Sum over Khuzestan Province from MODIS Satellite during the period 2017–2020.**

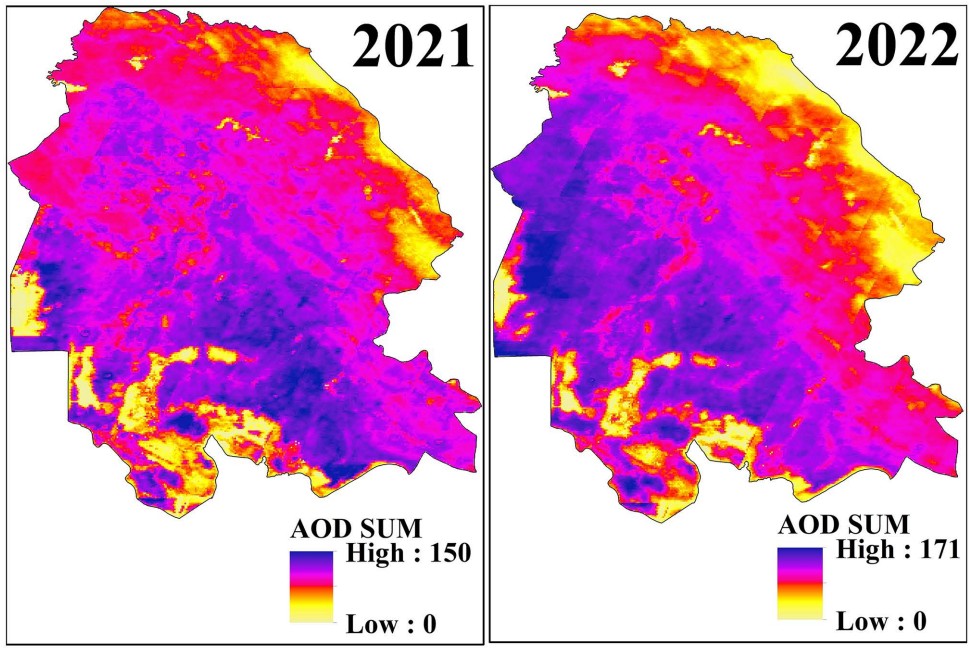

Source: MODIS (NASA EOSDIS). Processed and analyzed by the authors.

**Fig 8. Spatial Distribution of AOD Sum over Khuzestan Province from MODIS Satellite during the period 2021–2022.**

the wetland, which recorded 150 dusty days and fell within the "unhealthy for sensitive groups" to "hazardous" categories of air quality. Similarly, the city of Ahvaz also experienced 150 dusty days in 2018.

In 2019, the AOD-Sum index slightly decreased to 149, with a reduction in the overall extent of affected areas. The highest values were concentrated in the central, western, southern, and southwestern regions of the province. Concurrently, dust-free zones expanded, particularly around wetland areas. In 2020, the decreasing trend continued, with the index dropping to 142, although the spatial distribution pattern remained comparable to previous years.

In 2021, the AOD-Sum index increased again, reaching 150, similar to 2018, though the spatial extent of affected areas was narrower. The central regions and parts of the western and southern provinces recorded the highest values, while dust-free zones—especially around the Shadegan Wetland—expanded further. By contrast, 2022 marked the most critical year of the study, with the AOD-Sum index rising dramatically to 171, corresponding to more than half the year being affected by dust events. The spatial extent of dust-affected areas also expanded significantly, with the major dust-entry corridors into Khuzestan registering the highest index values. The central, western, northwestern, southern, and southwestern regions of the province all recorded more than 170 dusty days.

A key finding of this analysis is the mitigating role of the Zagros Mountains in the north and east of Khuzestan and of the province's wetlands, particularly Shadegan and Hoor al-Azim, in reducing dust event. Mountainous areas act as natural barriers due to their topography, while wetlands contribute through moisture retention and natural filtration. However, the progressive desiccation of wetlands and their transformation into dust sources highlights a severe ecological threat to the province's future.

Overall, the results confirm that Khuzestan Province is severely affected by recurrent dust events, posing a major threat to public health and environmental sustainability. To mitigate dust event, a comprehensive set of management strategies is urgently required. These include reforming cropping patterns and promoting water-efficient crops, implementing integrated watershed management and water storage programs, expanding vegetation cover in arid and desert areas, and

 

enhancing community awareness and participation in dust control. Moreover, the protection and ecological restoration of wetlands must be prioritized, as they play a critical role in reducing dust event intensity and safeguarding environmental stability in the region.

### 3.4. Identifying the sources of dust

Over the past five years, analysis of dust events in Khuzestan province has shown that in 2022, the pollution index reached its highest level in the 2018–2022 period. Given the importance of this issue, the present study, using the HYSPLIT model and MODIS Corrected Reflectance (True Color) satellite images, examines the origin of dust in Khuzestan province on five days in 2022 that had the highest dust index (AQI = 500), as shown in Table 2. The HYSPLIT model simulated dust movement and determined its pathways based on wind and atmospheric properties. A 48-hour Backward trajectory was selected to provide a more comprehensive view of dust dynamics, enabling better assessment of the frequency patterns, altitude changes, and flow evolution over two days. This duration also facilitates comparison between the last 24 hours and the preceding 24 hours, and aligns effectively with satellite observations to trace the sources and movement of dust plumes.

Additionally, MODIS Corrected Reflectance (True Color) satellite images were used to observe and analyze the characteristics of the earth's surface and the identified areas to determine the source of the dust. These images, with high accuracy and the ability to detect changes in the earth's surface, provide valuable information from various parts of the province at the specified time. Accurate analysis of this data and information obtained from the HYSPLIT model and satellite images allows for a more accurate and better analytical identification of the source of dust in Khuzestan province. These results will help environmental authorities and regional planners in adopting appropriate measures to control and reduce dust events.

Figs 9–16 are divided into four sections corresponding to Trajectory Frequencies Map, Trajectory Map, MODIS True Color Map, and AOD MAX, respectively.

Dust event No. 1 originated from southern Turkey, crossing the Mediterranean and traversing Syria and Jordan before converging at the Iraq–Iran border. Trajectory frequencies exceeded 60% along the Iraq–Iran border and peaked over 90% near Khuzestan, with airflow descending from 3,000 m over Syria to below 500 m in Ahvaz. MODIS imagery captured extensive dust coverage, confirming transport from Syrian and Iraqi deserts, while the AQI reached 500, indicating extremely hazardous conditions.

Dust event No. 2 involved airflows forming near the Greece–Turkey border and intensifying over the Syrian deserts, with frequencies rising from <10% over Europe to >30% over Syria and exceeding 90% along the Iraq–Iran border. Vertical profiles show descent from 4,000 m to below 500 m upon reaching Ahvaz. MODIS imagery verified dense dust layers, and the AQI again reached 500, demonstrating the persistent and severe impact of transboundary transport.

Dust event No. 3 was primarily influenced by European airflows that traversed the deserts of Egypt, Jordan, Saudi Arabia, and Iraq before entering Iran. Frequency maps indicate bifurcation into two main branches with maximum frequencies above 90% at the Iran–Iraq border. Air masses descended from 2,000 m to 500 m in Ahvaz, delivering high dust

**Table 2. Selected days for tracking pollution in Khuzestan province with different indices.**

| No. | Date | AQI |
|---|---|---|
| 1 | 29/01/2022 | 500 |
| 2 | 05/03/2022 | 500 |
| 3 | 13/03/2022 | 500 |
| 4 | 03/07/2022 | 500 |
| 5 | 01/08/2022 | 470 |

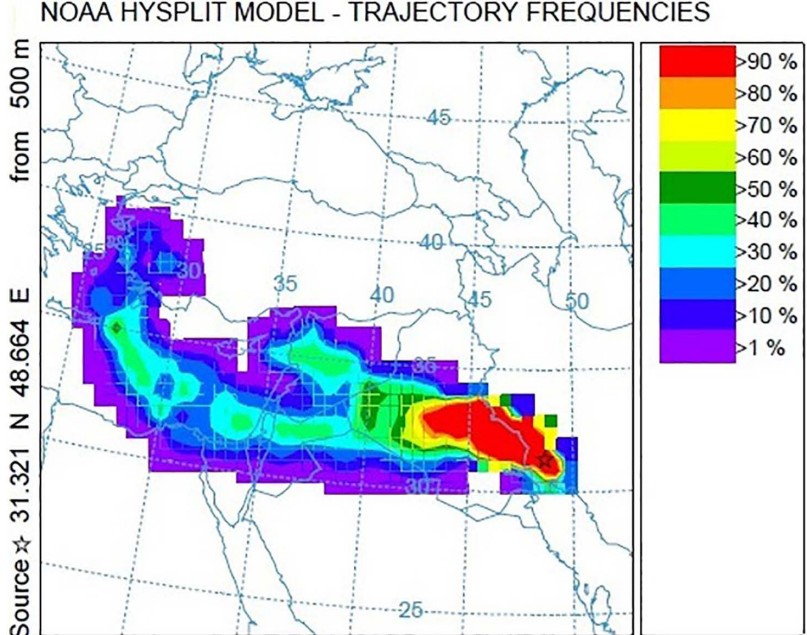

**Fig 9. Tracking of Dust Event using the HYSPLIT Model: 48-hour Trajectory Frequencies (27–29 January 2022).**

concentrations confirmed by MODIS, while the AQI reached 500, highlighting extremely hazardous conditions driven by complex transboundary pathways.

Dust event No. 4 exhibited a merged airflow from Turkey and the Syria–Jordan border, producing high-frequency trajectories (>90%) across central Iraq and descending to <500 m at the Iran–Iraq border. MODIS imagery captured dense dust plumes penetrating Khuzestan and adjacent regions, confirming the interaction of transboundary flows with local atmospheric dynamics.

Dust event No. 5 was distinguished by its predominant origin from internal regions of southern Iran, particularly the Hormozgan province and the Makran coastal areas. The dust plume's frequency increased sharply from approximately 10% over the Persian Gulf to over 90% upon reaching Khuzestan, indicating a rapid intensification during transport. Vertical airflows descended from 2,500 m to below 500 m, resulting in extreme air pollution levels with an AQI of 473 ("hazardous"). Analysis of MODIS True Color imagery revealed the swift accumulation of dense dust layers, highlighting the critical role of local sources, complex topography, and regional meteorological conditions in amplifying the severity and impact of this extreme dust event. This case underscores how southern Iran's internal dust-prone areas can significantly influence air quality in Khuzestan during major events.

Across all five events, trajectory frequency analysis quantitatively demonstrated the persistence and dominance of transboundary dust transport, particularly from the Syrian and Iraqi deserts, while internal sources in southern Iran occasionally amplified regional dust loads. The consistent descent of air masses prior to reaching Khuzestan significantly increased surface dust concentrations, and the spatial correlation between trajectory frequencies and satellite-derived dust plumes confirmed the accuracy of the HYSPLIT simulations. To enhance interpretation, AOD Max imagery was incorporated to validate spatial dust intensity patterns (Figs 12 and 16). The AOD Max maps revealed the highest aerosol concentrations between January 3 and 5 over central and southwestern Iraq and the Saudi border region, extending toward the Iran–Iraq frontier, indicating strong transboundary dust transport. AOD Max values ranged between 0.1 and 3.1 during this period. Similarly, during July 1–3, the AOD Max imagery highlighted elevated dust indices (0.2–5.2) over central and

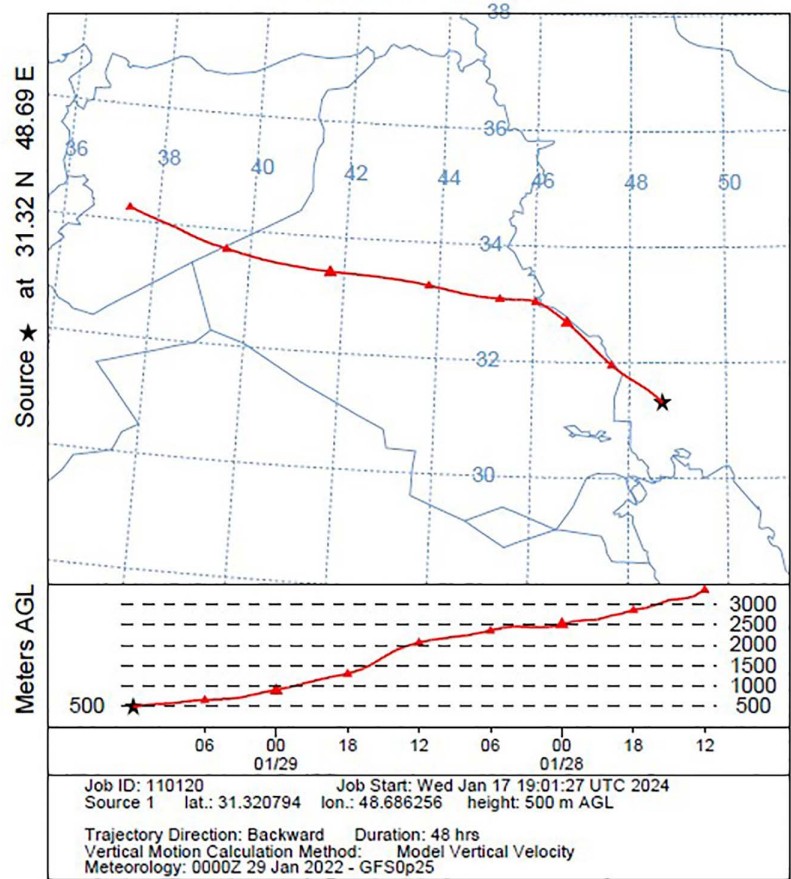

**Fig 10. Tracking of Dust Event using the HYSPLIT Model: 48-hour Backward Trajectory Map (27–29 January 2022).**

northern Iraq and along the Syrian border, with noticeable increases along the Iran–Iraq boundary and the Ahvaz region in Khuzestan Province. These observations not only confirm the precision of AOD Max imagery in identifying high dust concentrations but also substantiate the transboundary transport of dust from Iraq, Syria, and Saudi Arabia into southwestern Iran. Collectively, the integration of trajectory frequency, MODIS validation, and AOD Max analysis provides a comprehensive understanding of the spatial and vertical structure of dust events, reinforcing the reliability of satellite-based and model-based assessments for regional air quality management.

**3.4.1. Clustered dust pathways.** In this study, a total of 220 dust transport trajectories were analyzed for the year 2022, which was selected as a representative year characterized by high levels of dust pollution. These trajectories were generated using the HYSPLIT model driven by GLDAS meteorological data with a spatial resolution of 1° and a 3P configuration, and were computed as backward trajectories. Subsequently, a trajectory clustering procedure was applied to identify the dominant dust transport patterns. The results of this analysis are presented in Fig 17 and Table 3.

The backward trajectory analysis revealed six primary transport pathways with distinct contributions to dust loading over Ahvaz. Pathway 1, originating from Kuwait, accounted for 8.70% of the total trajectories and exhibited a relatively minor influence on the dust burden over the city. Pathway 2, which traverses the western border regions of Iran and

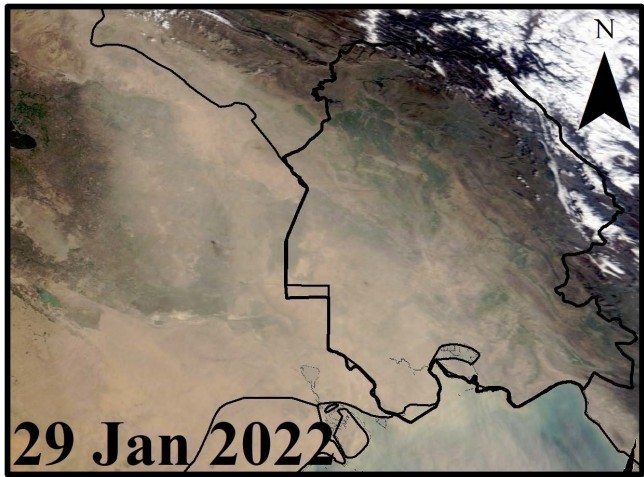

Source: MODIS (NASA EOSDIS). Processed and analyzed by the authors.

**Fig 11. MODIS True Color Image During the Dust Event (29 Jan 2022).**

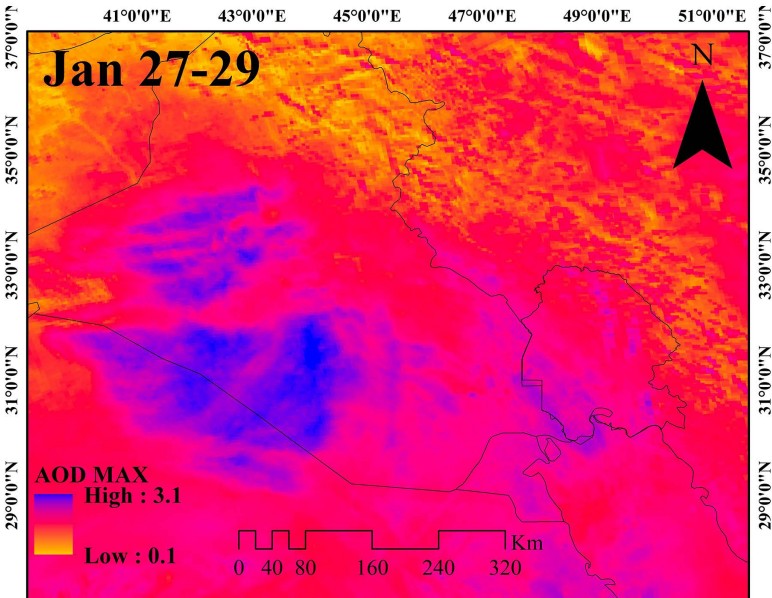

Source: MODIS (NASA EOSDIS). Processed and analyzed by the authors.

**Fig 12. AOD Max distribution during the dust event (27–29 January 2022).**

eastern Iraq and passes over the desiccated areas of the Hoor Al-Azim Wetland, represented the dominant pathway, contributing 43.48% of the total trajectories and exerting the strongest influence on dust pollution in Khuzestan Province and Ahvaz. Pathway 3, encompassing the shared regions of southwestern Kuwait and southeastern Iraq, contributed 13.04% of the total. Pathway 4, originating from southern Iran and internal sources, particularly the provinces of Fars and Bushehr, accounted for 8.70% and showed a limited contribution to dust loading. Pathway 5, extending from the desert regions of Iraq and further influenced by source areas in Syria and Jordan, contributed a substantial 21.74% and played a significant

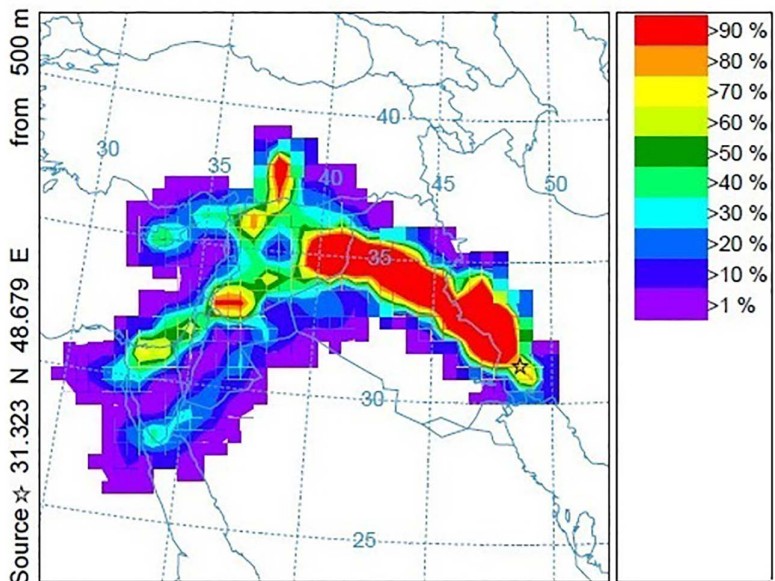

**Fig 13. Tracking of Dust Event using the HYSPLIT Model: 48-hour Trajectory Frequencies (2–3 July 2022).**

role in enhancing the dust influx into Ahvaz. Finally, Pathway 6, associated with air mass flows originating from Europe and Turkey and subsequently passing over Syria and Iraq before reaching Ahvaz, accounted for only 4.35%, representing the least influential transport route.

The integrated assessment of these pathways indicates that, with the exception of Pathways 1 and 4, all remaining transport classes traverse the desert regions of Iraq, collectively accounting for 82.6% of the total dust transport routes. This pattern underscores the strategic and dominant role of Iraqi desert landscapes as primary source and transfer zones for dust affecting Khuzestan Province and the city of Ahvaz. Moreover, the high proportion of air mass flows passing through the western border regions of Iran and northeastern Iraq (exceeding 40%) highlights the potential for targeted management strategies, transboundary cooperation, and source-control policies in these areas to substantially mitigate the dust burden impacting western Iran, particularly Ahvaz, and to enhance the effectiveness of environmental and public health risk management.

**3.4.2 Dust emission hotspots: WPSCF & WCWT Analysis.** The analysis of the WPSCF and WCWT maps presented in Figs 18 and 19 indicates that the main dust hotspots are located along a northwest–southeast belt in Iraq, aligning with the prevailing regional wind direction (northern Shamal winds) from Iraq and Syria toward Khuzestan. According to the WPSCF map, where AOD values range from 0 to 1, areas with values approaching 1 were identified as primary dust sources. Central, eastern, northern, and northeastern Iraq particularly regions surrounding the dried Hoor al-Azim wetland, the Mesopotamian basin, and abandoned agricultural lands around Baghdad with AOD values between 0.6 and 0.8, play the most significant role in dust particle generation. Additionally, the Syria–northwestern Iraq border, exhibiting high AOD indices, is recognized as a key transboundary source that directs dust particles along a southeastward path. Areas with moderate AOD values (0.4–0.6), including southern Iraq, Kuwait, and the border regions between Iraq and Khuzestan, act as secondary sources. These areas complete the transport pathways of dust plumes and contribute to increasing the concentration of suspended particles as they reach Khuzestan. The results indicate

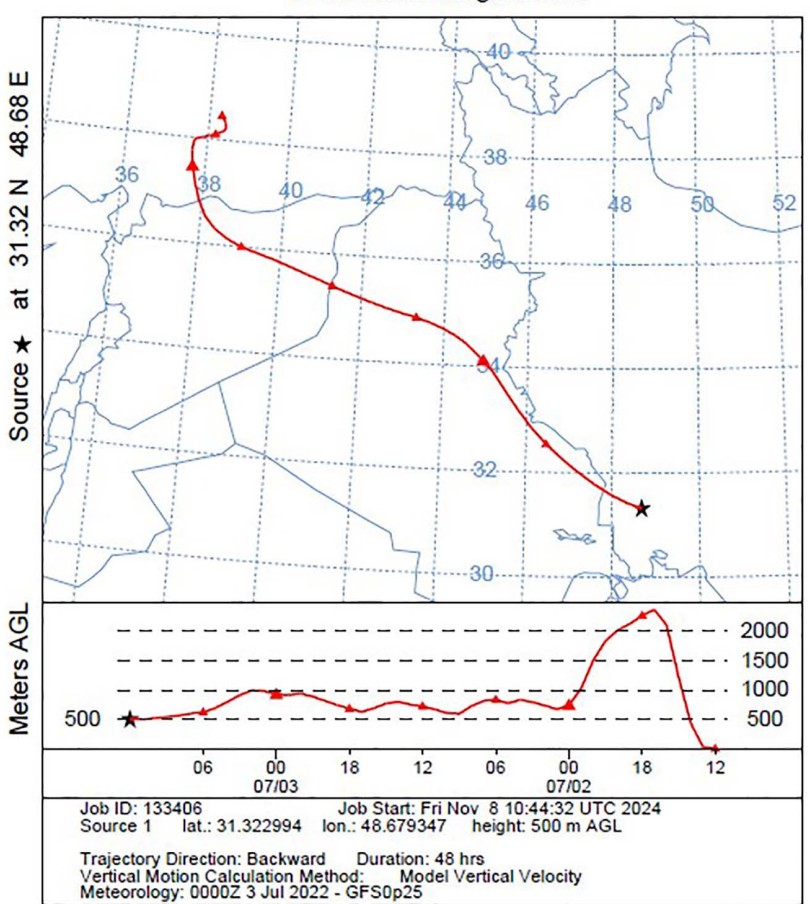

**Fig 14. Tracking of Dust Event using the HYSPLIT Model: 48-hour Backward Trajectory Map (2–3 July 2022).**

that during the studied days, particle concentrations in these regions were exceptionally high, with backward trajectories passing through these same points.

The WCWT index analysis further revealed that critical cores with AOD values greater than 1 are predominantly concentrated in two key regions: the central and southern Mesopotamian basin and the northwestern border strip of Khuzestan. In the Mesopotamian basin, the lowlands between the Tigris and Euphrates rivers particularly dried wetlands such as Hoor al-Hammar and the western parts of Hoor al-Azim have become major dust sources due to drought and reduced soil moisture. The northwestern border strip of Khuzestan, exhibiting high index values, indicates that dust particles generated in these cores enter the province with minimal deposition and maximum kinetic energy, intensifying local dust concentrations.

The elongated, linear arrangement of colored pixels from northwest to southeast reflects the dominance of stable synoptic flows, especially the northern Shamal winds. Areas with moderate AOD values (0.6–0.9) in the deserts of eastern Syria and Ninawa Province act as "amplifiers"; the passage of air masses through these regions gradually loads dust, reaching saturation points in central Iraq. In contrast, pixel discontinuities in remote northwestern areas suggest that contributions from distant deserts to severe events in Khuzestan are limited, as coarse particles tend to deposit over longer transport paths.

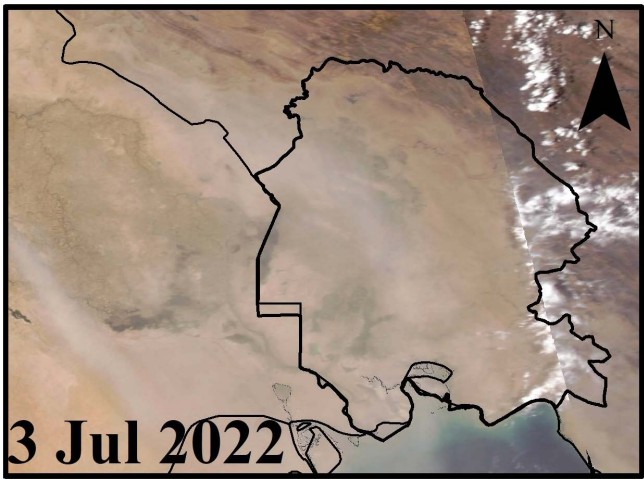

Source: MODIS (NASA EOSDIS). Processed and analyzed by the authors.

**Fig 15. MODIS True Color Image During the Dust Event (3 July 2022).**

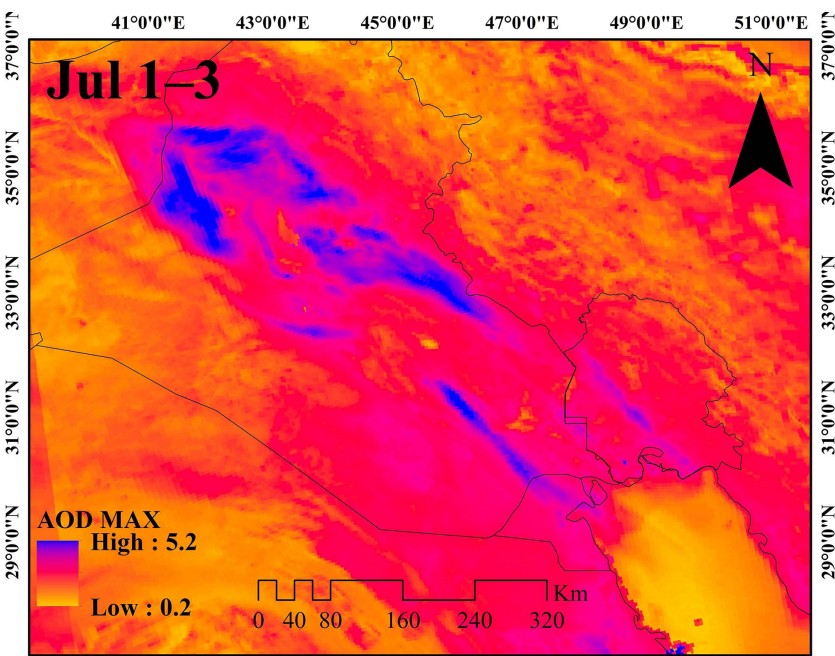

Source: MODIS (NASA EOSDIS). Processed and analyzed by the authors.

**Fig 16. AOD Max distribution during the dust event (1–3 July 2022).**

High AOD values near the borders of Khuzestan (0.9 to above 1) indicate that incoming air masses not only carry long-range dust but also interact with dense dust plumes generated in local cores, such as southern and eastern Ahvaz, leading to sudden increases in pollution indices at ground stations.

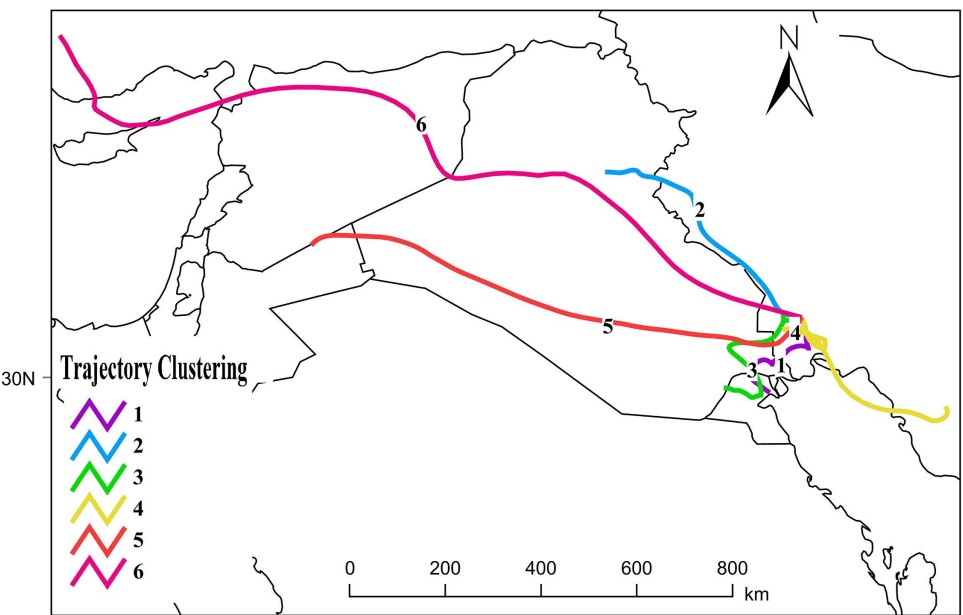

**Fig 17. Backward Trajectory Clusters for Dust Transport.**

**Table 3. Percentage share of dust trajectory clusters.**

| Cluster | Ratio |
| --- | --- |
| 1 | 8.70% |
| 2 | 43.48% |
| 3 | 13.04% |
| 4 | 8.70% |
| 5 | 21.74% |
| 6 | 4.35% |

Overall, this analysis demonstrates that the dust crisis in Khuzestan operates within a transboundary system: distant sources in Syria and northern Iraq generate the initial cores of dust plumes, while the dried lands of southern Iraq and the Khuzestan border strip act as amplifying reservoirs, producing the highest AOD values at the closest geographical proximity to the province. These results clearly highlight the simultaneous role of transboundary and local sources in the occurrence of severe dust events and can serve as a basis for management policies aimed at mitigating the environmental and health impacts of this phenomenon.

**3.4.3 Transboundary dust effects on Khuzestan.** Based on HYSPLIT model analyses, it was found that dust plumes originating from the western borders of Khuzestan Province follow trajectories through Syria, Iraq, and Saudi Arabia, reaching their highest intensity and frequency upon entering Iraq. These dust events typically arrive in Khuzestan within 24 hours after peaking in Iraq. Accordingly, to assess the relationship between Iraq-originated dust and air quality in the provincial center, i.e., Ahvaz, a 24-hour lag was applied. The data were then analyzed using selected statistical indices, as presented in Table 4.

The results indicate a strong association between transboundary dust from Iraq and AOD levels in Ahvaz, reflecting temporal coherence with aerosol transport patterns. Linear regression analysis shows $R^2$ scores of 0.85 (2018) and 0.75

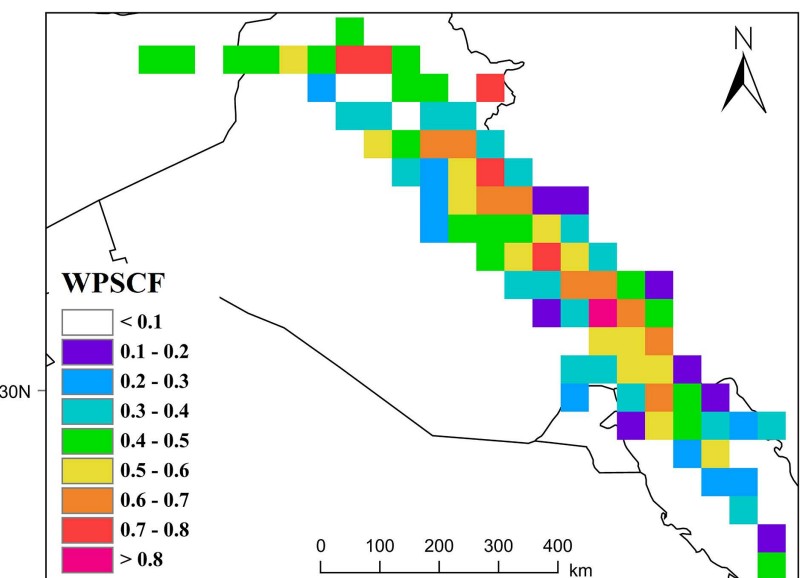

**Fig 18.  Source Contribution to Khuzestan Based on the WPSCF Model.**

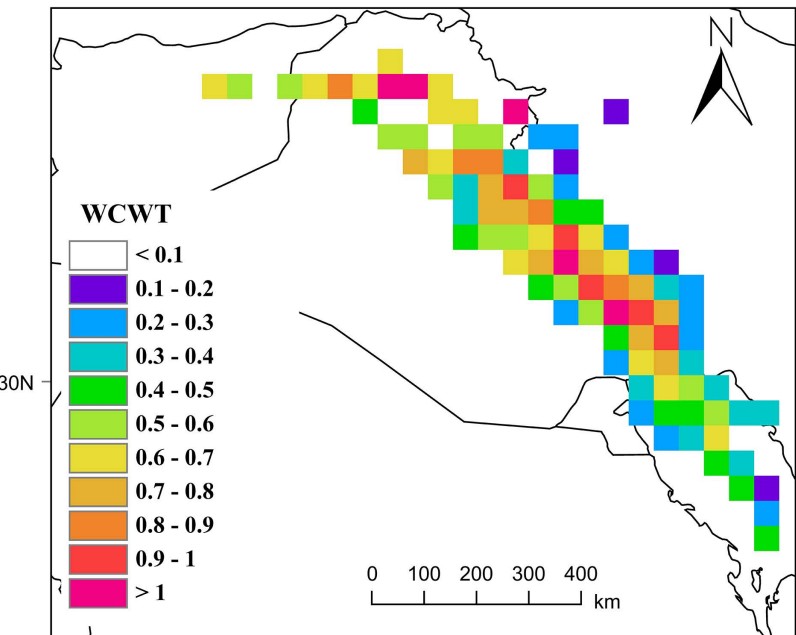

**Fig 19.  Dust Transport Pathways to Khuzestan Based on the WCWT Model.**

(2022), suggesting that a substantial portion of the variance in AOD-Ahvaz corresponds with AOD-Iraq. Consistently low p-values (0.001) in both years confirm the statistical significance of these relationships. Correlation analyses further support this association. Pearson coefficients (0.92 and 0.86) indicate high linear correlation, Spearman correlations (0.90 and 0.88) show robust monotonic relationships despite minor interannual fluctuations, and Mutual Information (0.92 and

**Table 4. Interannual Analysis of AOD-Iraq and AOD-Ahvaz Relationships.**

| Statistical Analysis | 2018 | 2022 |
|---|---|---|
| R² Score | 0.85 | 0.75 |
| P-value | 0.001 | 0.001 |
| Pearson Correlation | 0.92 | 0.86 |
| Spearman Correlation | 0.90 | 0.88 |
| Mutual Information | 0.92 | 0.80 |

0.80) reveals shared variability that includes nonlinear dependencies. Collectively, these analyses provide strong statistical evidence of a close temporal and quantitative association between dust levels in Iraq and aerosol loading in Ahvaz and the wider Khuzestan region. These results, supported by trajectory clustering, PSCF, and CWT analyses, indicate a clear association between regional dust transport and aerosol levels in Ahvaz, emphasizing the relevance of transboundary contributions in understanding and interpreting air quality patterns.

K-Means clustering analysis of the scatterplot between AOD in Iraq and AOD in Ahvaz identified three distinct pollution regimes with varying transboundary sensitivities, as shown in Fig 20. Cluster 0 (red), comprising ~65% of the data, spans AOD Iraq 0.0–1.8 and AOD Ahvaz 0.0–0.9, with a centroid at (0.7, 0.4), representing low-to-moderate pollution dominated by local sources. The local transfer gradient is ~0.35, meaning a 1-unit increase in AOD Iraq leads to a 0.35-unit increase in Ahvaz, highlighting the limited influence of Iraq under normal conditions.

Cluster 1 (green), ~30% of the data, spans AOD Iraq 1.2–2.5 and AOD Ahvaz 0.9–1.6, centroid at (1.8, 1.2), representing moderate pollution with regional synchrony. Transfer sensitivity rises to ~0.70, indicating significant contribution from medium-intensity Iraqi dust events transported by prevailing winds.

Cluster 2 (blue), ~5% of the data, spans AOD Iraq 2.5–5.5 and AOD Ahvaz 1.5–2.6, centroid at (3.8, 2.2), representing severe transboundary pollution events. The transfer gradient increases to ~0.85, with over 85% of Ahvaz pollution attributable to Iraqi sources.

The progressive increase in transfer sensitivity from 0.35 to 0.85 demonstrates that Iraq dominates Ahvaz pollution under moderate-to-severe conditions, while local sources control baseline levels. This three-tiered classification enables accurate prediction of Ahvaz pollution from Iraqi AOD and provides a scientific basis for targeted local and transboundary mitigation measures.

The HYSPLIT model is a Lagrangian particle tracking model used to trace the movement of particles in the atmosphere. The results of this model show that most dust events in Khuzestan province originate from the west and northwest, including border areas such as parts of Iraq, Turkey, Jordan, Syria, and Saudi Arabia. Several factors contribute to dust events in Khuzestan, including the region's dry and semi-arid climate, the drying of wetlands, and regional air currents.

Khuzestan's dry and semi-arid climate has led to very poor vegetation cover in the province. Vegetation plays a crucial role in stabilizing soil and preventing wind erosion. As a result, the lack of vegetation in Khuzestan has increased dust emissions. Wetlands also play an important role in absorbing and depositing dust, but the drying of wetlands in Khuzestan has reduced this capacity, further contributing to the rise in dust levels.

Air currents are another critical factor in dust transport. In Khuzestan, air currents blow from the west and northwest toward the east and south, carrying dust from neighboring regions. To better understand the impact of these wind currents on dust in the province, wind rose diagrams were plotted for various stations, including Abadan, Ahvaz, Bostan, Mahshahr, Shadegan, and Gotvand, using real data on wind direction and speed. A wind rose is a diagram that shows the wind direction and speed at a specific location over a given time period.

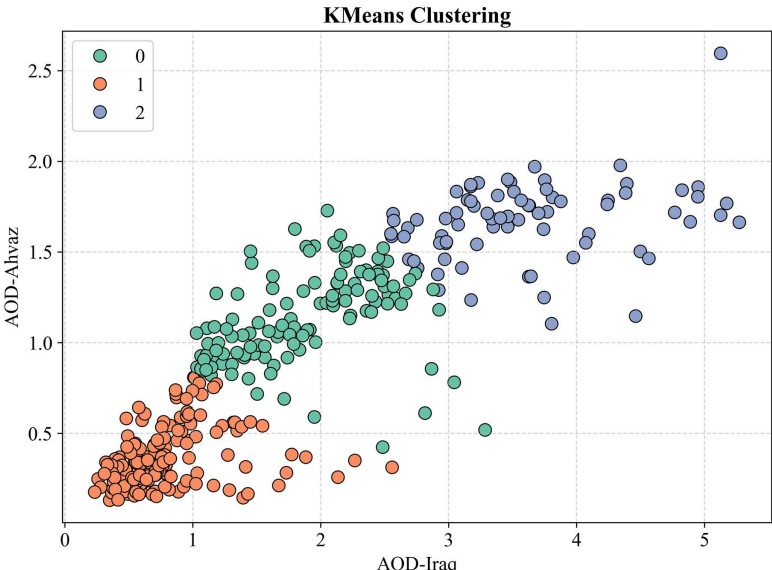

**Fig 20. K-means clustering of AOD in Iraq and Ahvaz, 2022.**

### 3.6. Wind analysis

Following the examination of dust paths and currents, annual wind roses were analyzed to identify the predominant wind patterns and their contributions to dust dispersion in Khuzestan Province. The analysis used WRplot 7 software to categorize winds into sixteen directions and six speed classes. Winds with speeds exceeding 1 mph (5.0 meters per second) were included in the analysis, while those below this threshold were classified as calm. Notably, dust events in the region often occur with winds exceeding 13 mph (5.8 meters per second), though Khuzestan's climate means that even lower wind speeds can contribute to dust events (Lotfinasabasl et al., 2021).

The annual wind roses provide insights into the frequency and direction of winds throughout the year, highlighting how prevailing wind patterns may influence dust distribution and intensity. The results, shown in Fig 21, reveal that over eighty-seven percent of the winds in Khuzestan Province are classified as calm, underscoring the importance of analyzing these patterns to understand their role in dust event formation and dispersion.

**Abadan Station:** The results for the 2022 wind rose of Abadan Station show that winds blow from the northwest more frequently than from other directions. Since Abadan Station is the closest station to the border, its wind rose reflects the currents coming from Iraq. Most of these winds have speeds exceeding 2 meters per second and can reach up to 11 meters per second, causing dust events.

**Ahvaz Station:** The wind rose for Ahvaz Station shows that most winds blow from the northeast and northwest, with speeds ranging from 2 to 6 meters per second. The speed of most of the winds that blow in different areas of Ahvaz station is more than 5 m/s.

**Gotvand Station:** An analysis of Gotvand Station, a northern station, reveals that most winds blow from the southwest and west.

**Shadegan Station:** The prevailing winds at Shadegan Station are from the northwest and south originating from the Persian Gulf. Given the wind speed and the dry areas of Shadegan Wetland, these winds cause dust dispersion in Shadegan County. Additionally, winds from the northwest and west transport dust from neighboring countries into Khuzestan

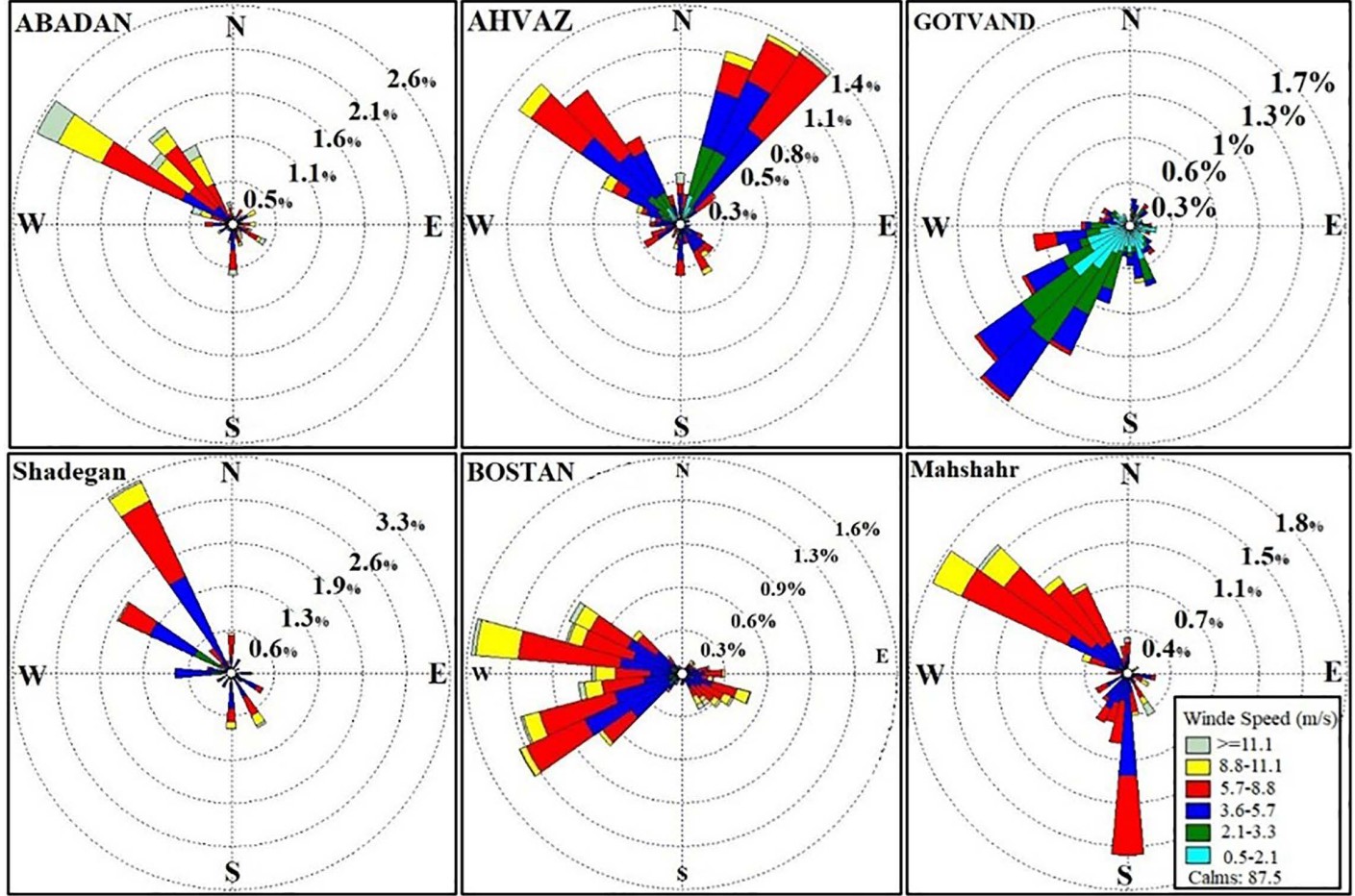

**Fig 21. Wind rose diagram (2022) of meteorological stations of Khuzestan province.**

Province. Upon passing through the dry areas of Shadegan Wetland, they further increase dust levels and carry it to the southern and southeastern parts of the province.

**Bostan Station:** The winds at Bostan Station are mainly from the west, northwest, and southeast, indicating transboundary currents.

**Mahshahr Station:** The wind rose for Mahshahr Station also shows that winds predominantly blow from the northwest and south (from the Persian Gulf) of the station, reaching speeds of over 5 m/s.

Based on the results of the wind roses, it can be concluded that the prevailing winds in Khuzestan Province are from the northwest and west. These winds carry dust from neighboring countries like Iraq and Syria into Khuzestan Province, causing air pollution. Additionally, the passage of wind and dust currents over the dry areas of the Wetlands increases the dust index, and this dust is transported to the central regions of the province. Moreover, winds from the south and southeast can transport dust from the Gulf countries into Khuzestan Province. These winds cause dust events in the dry areas of Shadegan Wetland.

### 3.7. Assessment of Hoor Al-Azim Wetland Shrinkage

The extensive wetlands of Khuzestan Province, particularly the Shadegan and Hoor al-Azim Wetlands, have historically acted as natural filters by trapping and depositing suspended particles, thereby reducing atmospheric dust levels and

mitigating the intensity and frequency of local dust events. Under conditions of severe hydrological stress, however, this filtration capacity weakens, increasing the vulnerability of the system to internal dust generation. Based on recent assessments, 2022 was identified as the driest year for these wetlands within the 2018–2022 period [39]. Considering that more than 70% of dust transport trajectories pass through Hoor al-Azim Wetland, this study specifically examined the hydrological condition of the wetland (i.e., its water availability or dryness) and its relationship with dust events during 2022.

The variations in wetland surface water extent (derived from the MNDWI index) and the water level of the Hoor al-Azim Wetland over a five-year period (2018 to the end of 2022), as well as the relationship between these variables, are illustrated in Figs 22 and 23. During 2018, both indicators exhibit relatively low and stable values, with the surface area generally remaining below 900 km² and water levels fluctuating between approximately 1.7 and 2.9 m, reflecting a moderately constrained hydrological regime. A pronounced expansion phase is observed throughout 2019 and early 2020, characterized by a substantial increase in wetland surface area, reaching a peak of nearly 2,000 km² in April 2019, concurrently with elevated water levels exceeding 5 m. This synchronous response indicates a strong positive relationship between water availability and the spatial extent of surface water, confirming the sensitivity of the MNDWI-derived area to variations

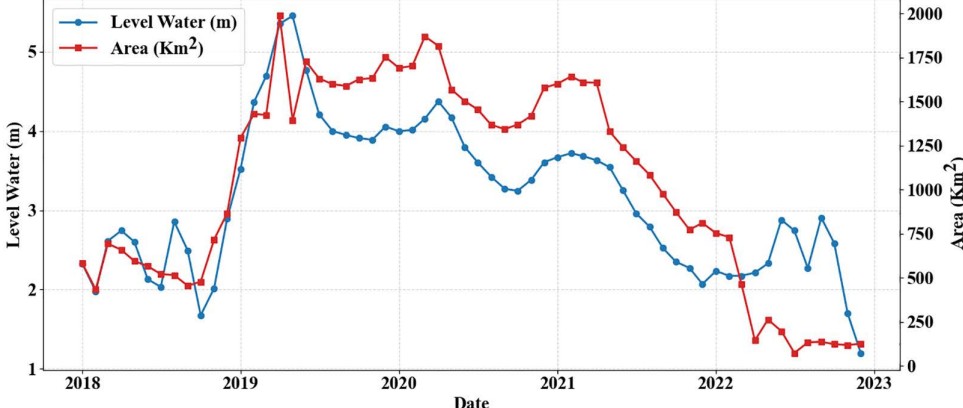

**Fig 22. Temporal dynamics of Area (Km²) in the Hoor al-Azim Wetland (2018–2022).**

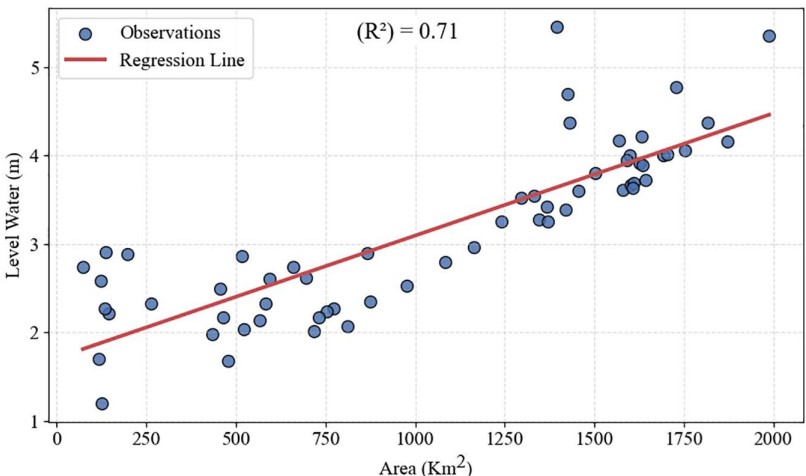

**Fig 23. Regression-based relationship between Area (Km²) and water level in the Hoor al-Azim Wetland (2018–2022).**

in wetland inundation depth. From mid-2020 onward, both variables display a gradual declining trend. The wetland surface area contracts steadily from values above 1,500 km² to below 1,000 km² by late 2021, while water levels decrease from approximately 4.0 m to around 2.1–2.5 m. This phase reflects a transition toward hydrological stress and partial desiccation of the wetland system. The most critical conditions are observed in 2022, during which the wetland experiences an abrupt and severe contraction. Surface area values derived from MNDWI drop dramatically, reaching a minimum of approximately 73 km² in July 2022, accompanied by a marked decline in water levels to nearly 1.2–2.9 m. This decoupling at very low water levels suggests extensive exposure of the wetland bed and fragmentation of surface water patches, indicating extreme drying conditions.

Overall, the relationship between variations in wetland surface area derived from MNDWI and changes in water depth (water level) is statistically significant, with a coefficient of determination of $R^2 = 0.71$, indicating a strong and meaningful linkage between spatial extent and hydrological conditions. The observed fluctuations are predominantly seasonal in nature; however, the five-year record reveals a pronounced long-term declining trend in both surface water area and water depth. This sustained reduction indicates progressive hydrological degradation of the wetland system, culminating in severe drying conditions in 2022, during which the Hoor al-Azim Wetland experienced a critical phase of water scarcity and extensive contraction of surface water coverage.

The spatiotemporal patterns of the MNDWI index for the twelve months of 2022 in the Hoor al-Azim Wetland are presented in Figs 24 and 25. The results indicate that the wetland was predominantly subjected to persistent dry conditions and a substantial reduction in surface water extent throughout the year. In January, MNDWI values ranged from 0.42 to −0.29, reflecting moderate to high dryness across large portions of the wetland, with surface water mainly confined to limited areas in the eastern, northwestern, and parts of the southeastern sectors, while the remaining regions exhibited characteristics of dry land. In February, the index range (0.43 to −0.28) showed no marked change compared to the previous month, indicating the persistence of stable but hydrologically constrained conditions. In March, a decline in MNDWI values to 0.35 to −0.25 was accompanied by a noticeable retreat of surface water in the eastern and southeastern sectors, suggesting an intensification of drying processes relative to earlier months. In April, although the index range remained broadly similar, the spatial distribution of surface water continued to be limited to scattered patches in the northwestern and eastern parts of the wetland, while extensive areas of the central and southern regions remained dry.

In May, despite a relative increase in MNDWI values to 0.38 to −0.27, the surface water extent further contracted, and the southeastern sector largely transitioned to dry conditions, leaving only restricted areas in the eastern and northwestern portions of the wetland with detectable surface water. This trend intensified during June and July. In July, with MNDWI values decreasing to 0.33 to −0.25, the wetland reached its annual minimum in surface water extent, and surface water was primarily limited to small, fragmented patches in the eastern sector and a minor area in the northwestern part, indicating a critical phase of wetland desiccation. In August, a slight and localized recovery of surface water was observed. In September, a relative increase in MNDWI values to 0.37 to −0.28 corresponded to a modest expansion of surface water in parts of the eastern and northwestern sectors. However, in October, November, and December, the index values remained within a similar range, and the spatial pattern indicated the stabilization of dry conditions across much of the wetland. During this period, the western, central, northern, and southern regions were predominantly characterized by persistent dryness, with surface water confined to limited areas in the eastern, northwestern, and parts of the southeastern sectors.

Overall, the results for 2022 demonstrate widespread and sustained desiccation of the Hoor al-Azim Wetland, which not only substantially reduced its hydrological and ecological capacity but also significantly weakened its natural function in trapping and depositing suspended particles. Under these conditions, the exposed and dried wetland surfaces become increasingly susceptible to acting as active sources of dust, while the wetland's ability to attenuate and filter externally transported particulate matter is markedly diminished.

To examine the relationship between the reduction of the Hoor Al-Azim Wetland area and its impact on dust emissions in Khuzestan, a statistical analysis and K-Means clustering were performed on the wetland surface area and the

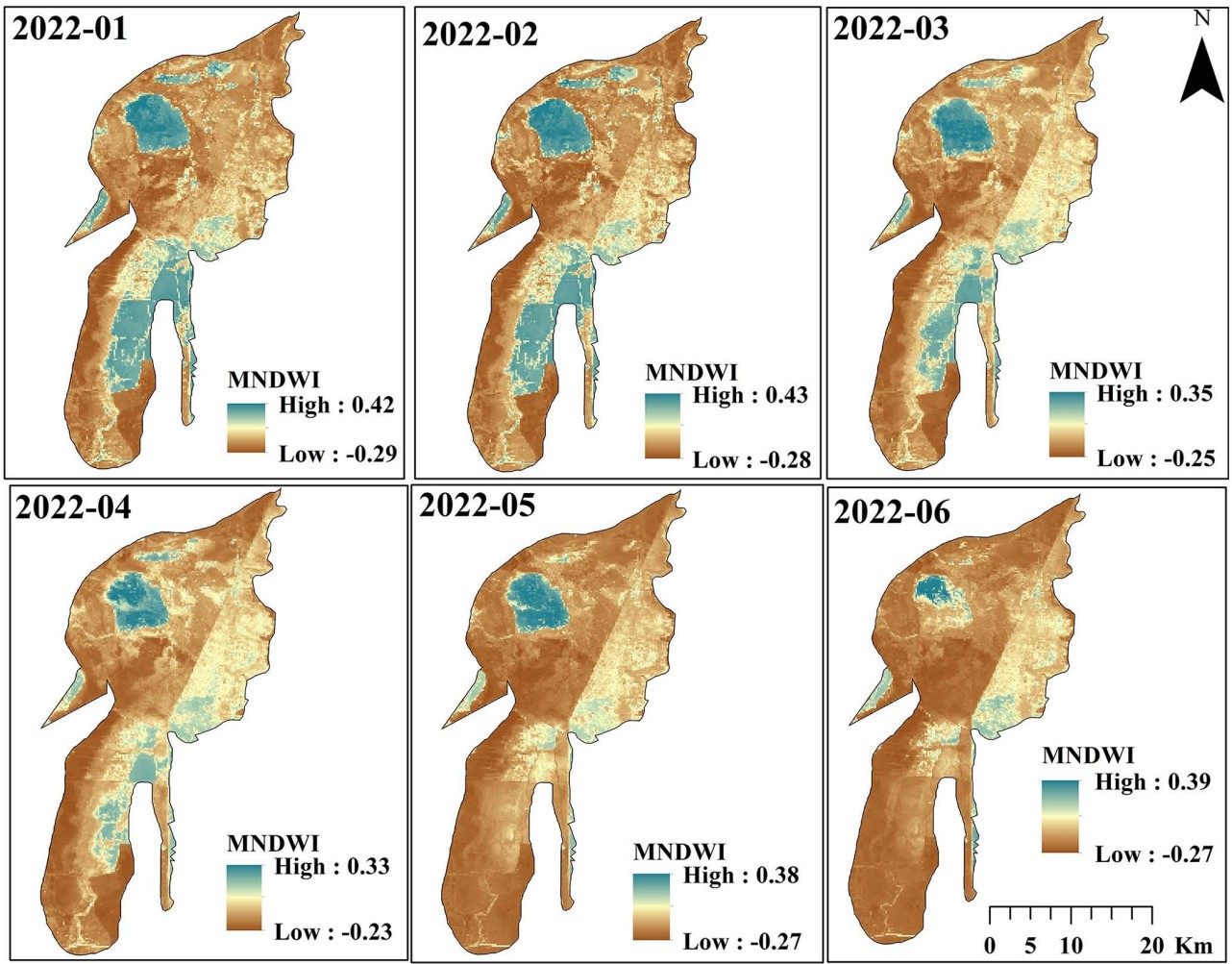

Data source: Sentinel-2 (Copernicus Open Access Hub). Data were processed, analyzed, and visualized by the authors.

Fig 24. Monthly variations of the water surface area of the Hoor al-Azim Wetland based on the MNDWI for the first six months of 2022 (January–June).

AOD surrounding the wetland, as shown in Fig 26. The R² was 0.78, indicating a strong relationship in which variations in wetland area explain a significant portion of the changes in AOD. Fig 26 further illustrates three distinct clusters representing the wetland's conditions, ranging from fully healthy to critical. The first cluster, indicated in purple at the lower right of the plot, represents an ideal and hydrologically healthy state, characterized by a large wetland area (approximately 1100–2000 km²) and very low AOD values (mostly below 0.4). These points correspond to wet periods, such as in 2019 and early 2020, when the extensive wetland acted as a natural filter, suppressing the suspension of particulate matter and maintaining air quality. The second cluster, shown in yellow at the center of the plot, represents a transitional and alert condition, with a moderate wetland area (400–1000 km²) and medium AOD values (0.4–1.0). This cluster indicates water stress; as the wetland area decreases, marginal dust sources become active and AOD begins to rise. Such conditions typically occur at the end of summer or during dry years, like 2018, signaling the onset of a potential crisis. The third cluster, depicted in green/blue at the upper left, represents a critical state and an active dust source, characterized by a

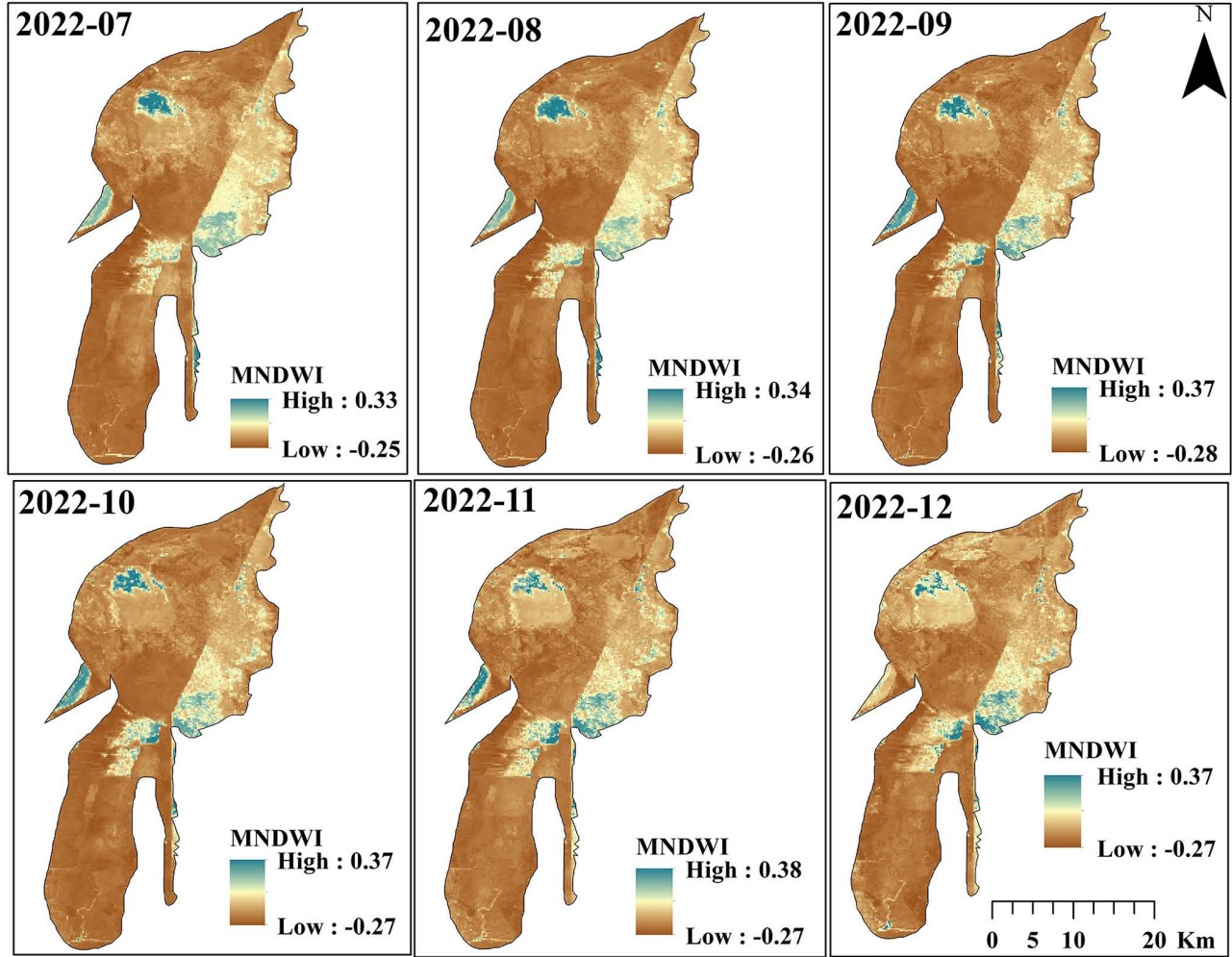

Data source: Sentinel-2 (Copernicus Open Access Hub). Data were processed, analyzed, and visualized by the authors.

**Fig 25. Monthly variations of the water surface area of the Hoor al-Azim Wetland based on the MNDWI for the last six months of 2022 (July–December).**

very small wetland area (less than 250 km²) and very high AOD values (above 1.0, up to 1.7). These points reflect the most critical conditions, as observed in 2022, when the wetland was nearly dry, turning its bed into one of the most active internal sources of dust. A clear inverse relationship between wetland area and AOD is evident; as the area decreases, dust concentrations increase exponentially.

The cluster centroids, marked by red stars, indicate the average behavior of each cluster. The centroid of the critical cluster suggests that when the wetland area approaches approximately 150 km², AOD is expected to reach a hazardous level of 1.4, whereas the centroid of the healthy cluster indicates that maintaining clean air (AOD around 0.25) requires a wetland area above 1500 km². The staged reduction of wetland area is particularly noteworthy: the first reduction stage, from 1500 to 700 km², leads to a relative AOD increase of about 0.4 units, while the second stage, from 700 to 150 km², results in a sharp AOD rise of approximately 0.7 units, highlighting the exponential growth of dust concentration under critical conditions. The clustering analysis, as presented in Fig 13, clearly identifies a "red line" for wetland area at around 500 km², below which the wetland's capacity to suppress dust emissions drastically decreases and the system enters a

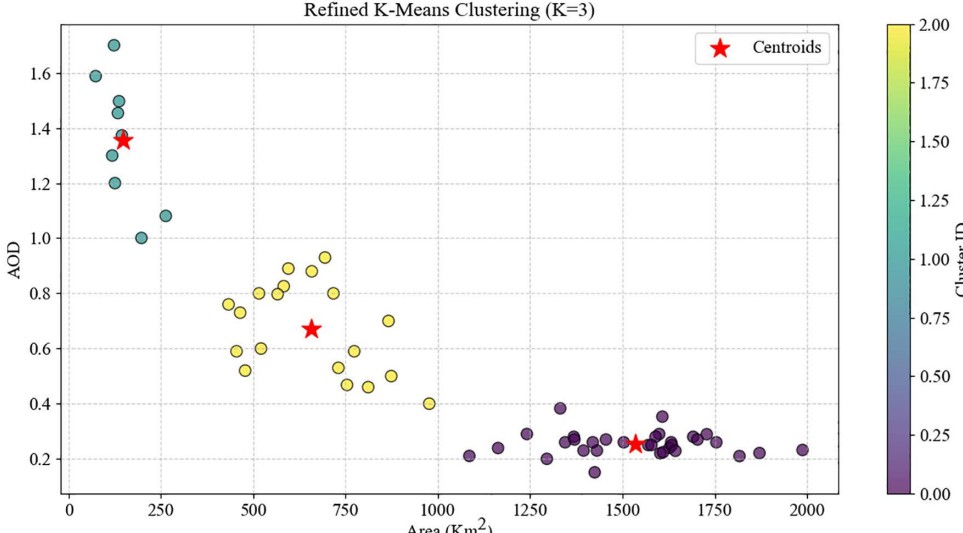

**Fig 26. K-Means clustering of Hoor Al-Azim Wetland based on surface area and surrounding AOD.**

critical phase (AOD > 1). This information can be instrumental for dust crisis management in Khuzestan, enabling the identification of critical wetland areas. Overall, the combined cluster and regression analysis not only demonstrates the direct linkage between wetland hydrology and air quality but also establishes the minimum wetland area required to prevent substantial dust increases, providing a practical framework for environmental monitoring and management.

### 3.8. Analysis of dust event maps

To more precisely examine the role of wind patterns and the reduction of wetland area in exacerbating dust event in Khuzestan Province, annual data of AOD from MODIS and AAI from Sentinel-5 were analyzed for the period 2018–2022, as shown in Figs 27 and 28. Based on AOD and AAI indices, in 2018, AOD values ranged from 0.1 to 2.1, with the highest concentrations observed in the western, central, and southern regions of Khuzestan, which correspond to the main dust transport pathways. In particular, areas surrounding the Wetland, especially its eastern and southern margins, experienced the highest AOD values. The AAI index during the same year ranged from −0.7 to 0.8, with a spatial pattern largely consistent with AOD, indicating the presence of light-absorbing aerosols in these regions.

In 2019, AOD exhibited a relative decrease, and the spatial extent of dust event was reduced compared to 2018; AOD values ranged between 0.1 and 2, affecting the western, central, and southern parts of the province, albeit with lower intensity. The AAI index also decreased, ranging from −1.1 to 0.3, and its spatial coverage contracted. This reduction in dust event coincided with the Wetland reaching its maximum recorded area, highlighting the wetland's significant role in mitigating dust emissions.

In 2020, the trend shifted. Although AOD values remained between 0.1 and 2, the spatial extent of dust event expanded relative to 2019, impacting larger areas, particularly the central, southern, and parts of the western provinces. The eastern and southeastern margins of the wetland also exhibited a notable increase in AOD. The AAI index similarly increased, ranging from −1.3 to 0.5, with its spatial distribution mirroring that of AOD, reflecting higher concentrations of light-absorbing aerosols and intensified local dust event.

In 2021, AOD values continued to vary between 0.1 and 2, with spatial coverage further expanding compared to 2019 and 2020; extensive areas of the central, southern, western, and southwestern provinces approached AOD values of 2.

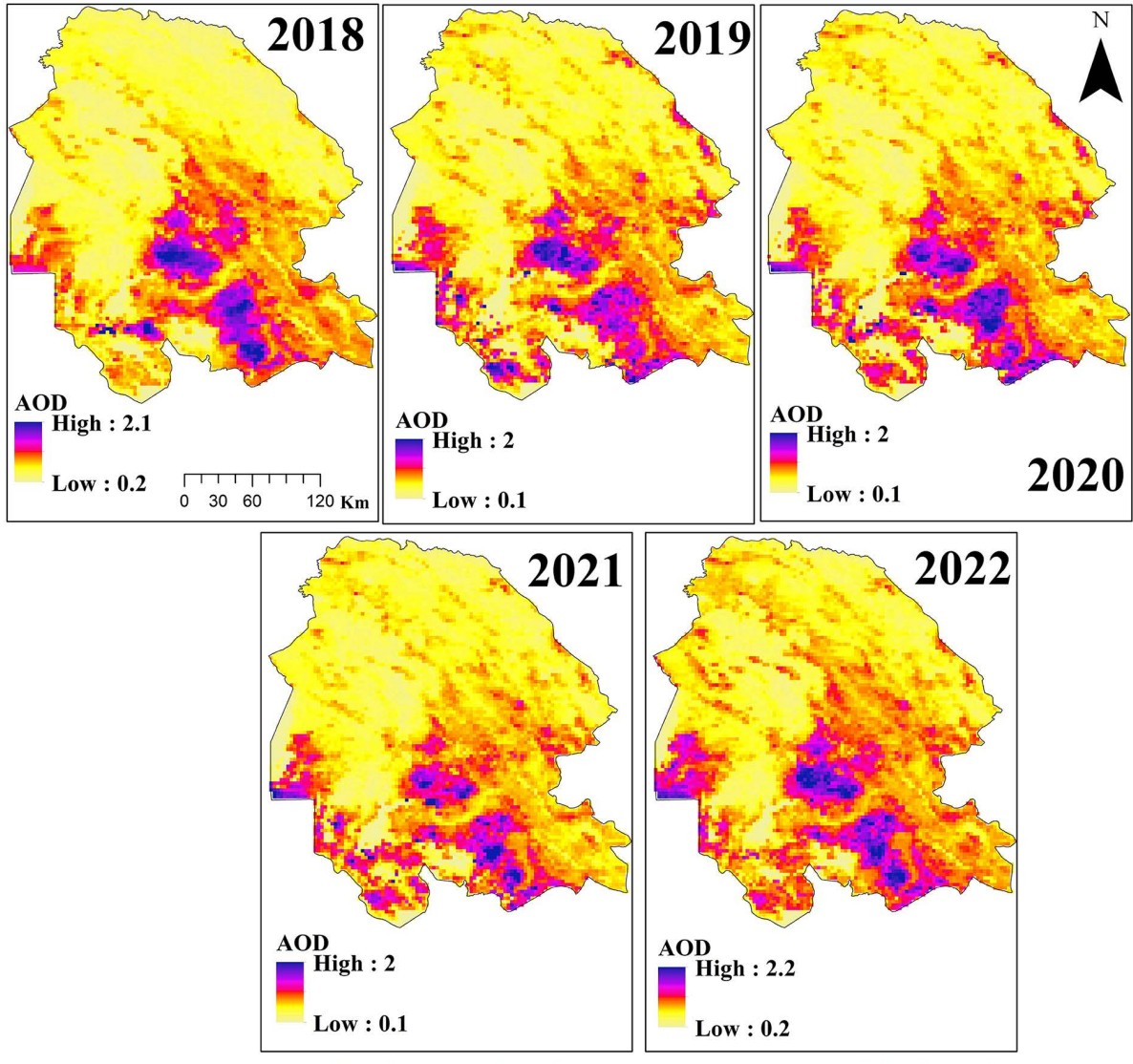

Source: MODIS (NASA EOSDIS). Processed and analyzed by the authors.

**Fig 27. Distribution of AOD MODIS for dust monitoring in Khuzestan Province during 2018-2022.**

The southern and eastern parts of Wetland also exhibited substantial increases in AOD. The AAI index ranged from −0.7 to 0.7, with a spatial pattern closely following that of AOD, indicating a continued intensification of dust event and the wetland's role in modulating or amplifying it.

The year 2022 marked the peak of dust events during the study period. AOD reached its maximum, ranging from 0.2 to 2.2, with the central, southern, and western regions experiencing the highest values. In addition, the southeastern and southern parts of the wetland recorded elevated AOD levels. The AAI index also exhibited a significant increase, ranging from −0.1 to 1.3, indicating critical dust event conditions and high concentrations of light-absorbing aerosols. The spatial distribution of AAI closely mirrored that of AOD and coincided with the most pronounced reduction in wetland area observed during the study period. This strong association between wetland shrinkage and the increased intensity and spatial extent of dust events underscores the critical role of the wetland as a natural regulator of regional dust emissions.

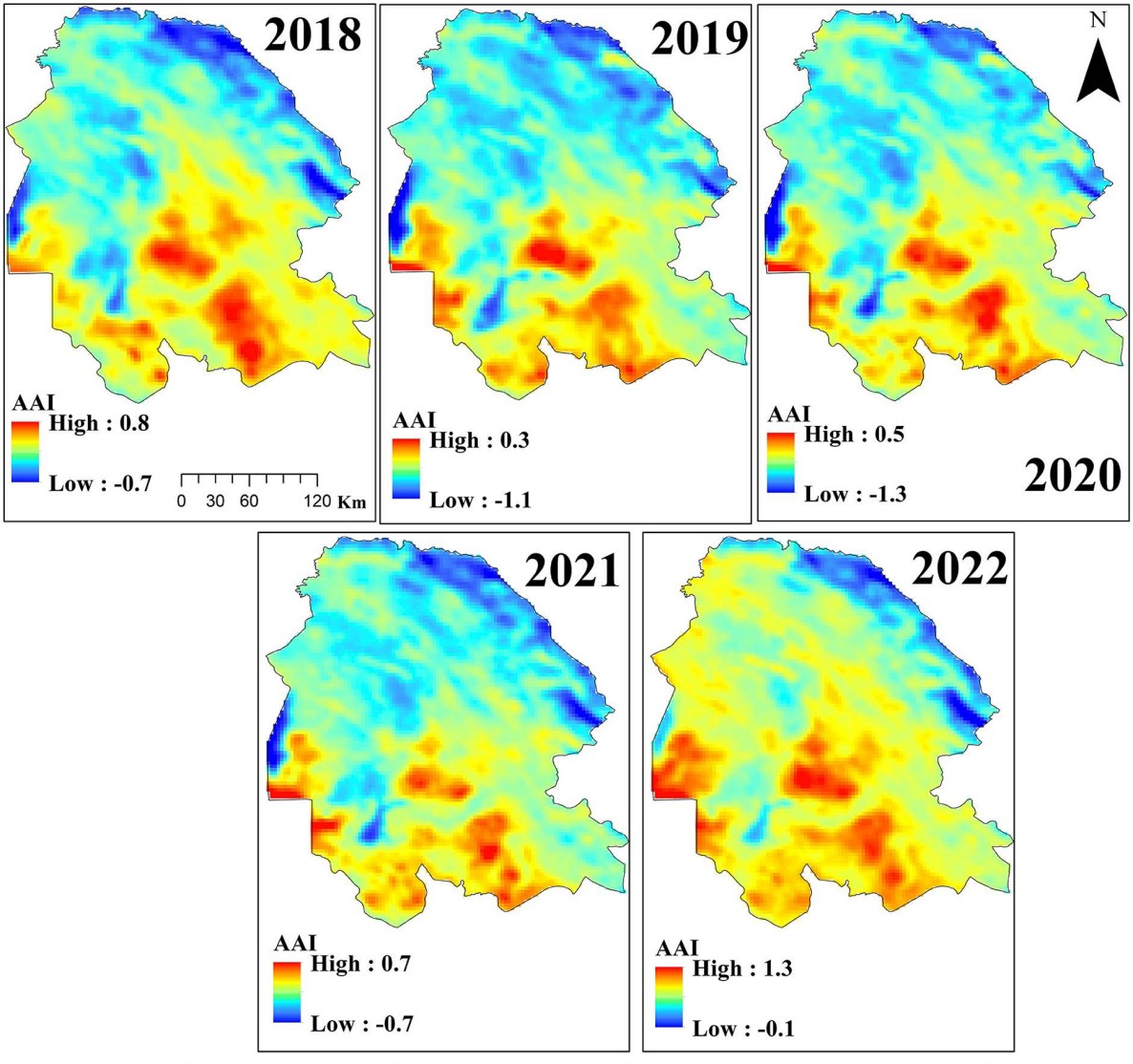

Data source: Sentinel-5P (TROPOMI), Copernicus Open Access Hub. Data were processed, analyzed, and visualized by the authors.

**Fig 28. Distribution of AAI SENTINEL 5 dust monitoring in Khuzestan Province during 2018-2022.**

Figs 27 and 28 further highlight the highest pollution levels in the southeastern, eastern, and southern directions of the wetland. Based on wind rose analyses, prevailing winds entrain dust particles from the dried wetland beds and transport them to other regions of the province. The highest levels of both AOD and AAI during the study period were observed in 2018 and 2022.

The analysis of MODIS AOD and Sentinel-5 AAI data effectively elucidates the spatial and temporal distribution of dust event in Khuzestan Province. Wind patterns, the drying of wetlands, drought effects, desertification, and human activities emerge as key factors in shaping this environmental challenge. Addressing dust event requires an integrated approach that includes land-use management, dust mitigation strategies, and regional cooperation to control transboundary dust transport.

## 4. Discussion

Dust events are significant phenomena caused by airflows and winds that affect air quality in various regions worldwide. Specifically, desert areas of Iraq, Syria, Jordan, Turkey, and Saudi Arabia actively contribute to dust events, serving as major sources of sand and dust in the Middle East. Both local and regional factors influence the formation of these storms. Regional climate changes are a primary driver, mainly due to environmental alterations such as desertification, land degradation, and wetland drying. Moreover, a substantial reduction in annual rainfall and prolonged droughts are key local contributors. When this dust enters Khuzestan province, it leads to significant air pollution. Climatic conditions in the border areas of Khuzestan and the drying of wetlands further elevate dust indices.

This study utilized MODIS satellite data and the AOD-Max index to identify areas with the highest dust levels, while the AOD-Sum index was applied to quantify total AOD for Khuzestan province during 2018–2022. Dust pathways and sources were analyzed using the HYSPLIT model, complemented by cluster maps to assess the percentage of predominant dust entry routes and WPSCF & WCWT analysis to evaluate the contribution of source regions. MODIS True Color images were used for visual validation, and wind roses from various stations highlighted the prevailing wind directions. Additionally, the relationship between the decreasing area of Hoor al-Azim Wetland (derived from Sentinel-2 MNDWI and water level data) and AOD was examined. A 24-hour lag correlation was applied to evaluate the delayed impact of external dust sources from neighboring regions, such as Iraq, on air quality in Khuzestan, particularly regarding PM concentrations. Dust distribution maps were generated using Sentinel-5 AAI and MODIS AOD indices.

### 4.1. Comparison with previous studies

MalAmiri et al. [48] evaluated the socio-economic impacts of dust storms in Khuzestan Province, focusing on human health, agriculture (Estamran dates), and migration. Total damages were estimated at around USD 39 million, with major losses from respiratory and cardiovascular diseases and agricultural damage, particularly in Shadegan. In comparison, our study focuses on the environmental and physical aspects of dust events. We analyzed five major events, mapped high AOD areas using Sentinel-5 and MODIS, assessed wind patterns, included internal sources such as the Hoor Al-Azim Wetland, and evaluated the contribution of each dust entry route across five classes using cluster mapping, along with WPSCF & WCWT analyses. This approach provides a more comprehensive understanding of dust dynamics and sources, complementing socio-economic analyses like MalAmiri et al. [48].

Shayeghannoor and Ahmadi [49], they analyzed dust events in Khuzestan over 65 years using hourly, monthly, and annual data from 19 synoptic stations. The Mann–Kendall test and Sen's slope method were used to assess trends, and minimum and maximum dust days were examined. While this study provides valuable insights into temporal trends and peak dust hours, our study expands on it by analyzing five major dust events, using maximum AOD to identify high aerosol regions, incorporating multiple satellite-derived indices (Sentinel-5 and MODIS), assessing wind patterns, evaluating wetland influence, and applying 24-hour lag correlations to study the relationship between surrounding dust and Ahvaz air quality.

Dargahian et al. [50] used MODIS and HYSPLIT (2003–2017) to identify major external dust sources affecting Khuzestan, finding eastern Iraq, the Iraq-Syria border, southern Iraq, and northern/eastern Arabia as dominant contributors, with Iraq responsible for 68.8% of dust emissions. In contrast, our study extends this approach by also considering internal sources, particularly the Hoor Al-Azim Wetland. We analyzed five major dust events, identified five main dust entry pathways into the province and quantified the contribution percentage of each route. Furthermore, using WPSCF & WCWT analyses, we evaluated the potential contribution of each source to pollutant concentrations, while also assessing the role of Hoor Al-Azim Wetland reduction in increasing dust indices. Maximum and averaged AOD indices from Sentinel-5 and MODIS, wind patterns, and 24-hour lag correlations were applied to provide a more comprehensive and integrated understanding of dust dynamics in the region.

Mohammadi et al. [51] This study explored the drivers of dust storms in Khuzestan Province, linking increased frequency to human activities (e.g., dam construction, oil extraction, agricultural changes) and climate variables such as

rising air and land surface temperatures and reduced water bodies (1985–2019). The analysis showed that temperature increases explained 95% of dust variability, and stressed the need for regional cooperation and policy adjustments to mitigate dust hazards. In contrast, our study emphasizes the real-time dynamics of dust events rather than long-term climate and human activity trends. By analyzing five major dust events, mapping high AOD areas with Sentinel-5 and MODIS, examining wind patterns, including internal sources like Hoor al-Azim wetland, and applying 24-hour lag correlations, we provide a detailed understanding of both internal and external contributors to dust in Khuzestan. Unlike Mohammadi et al. [51] our approach links environmental measurements directly to air quality impacts in Ahvaz, offering actionable insights for local monitoring and prediction.

Tajiki et al. [52], this study examined the effect of dust sources on airborne fungi using HYSPLIT and local wind data, focusing on dust from Arvand Free Zone and soils from Hoor al-Azim and Shadegan wetlands. They found that Hoor al-Azim was the main contributor to airborne fungi. In contrast, our study examined the Hoor al-Azim wetland in detail, considering different land cover classes to assess how wetland drying contributes to increased dust emissions. Additionally, we conducted a broader analysis of dust sources entering Khuzestan Province, identifying their origins and pathways. Therefore, our study provides a more comprehensive and spatially extensive assessment compared to Tajiki et al. [52].

Borna et al. [53], this study identified the origin, transport, and deposition of dust in western Iran (2000–2014) using MODIS satellite data and HYSPLIT simulations, highlighting both domestic and international sources, with major contributions from Syria and Iraq. Our study shares a similar approach in using HYSPLIT for source analysis, but extends it significantly. In addition to HYSPLIT, we employed maximum and averaged AOD indices to identify the most dust-prone areas in Khuzestan. A 24-hour lag correlation was applied to quantify the influence of dust from neighboring countries on Ahvaz air quality. Wind rose analyses were conducted, and internal dust sources, including the Hoor al-Azim wetland, were evaluated. Overall, our study provides a more extensive and detailed regional assessment compared to Borna et al. [53].

Khalidy et al. [54], this research simulated a severe dust event in June 2016 using HYSPLIT for 33 potential sources, quantifying contributions from neighboring regions and Hoor Al-Azim wetland. Compared to this, our study integrates multiple dust events, uses maximum AOD to identify high-concentration areas, monitors dust continuously, and applies 24-hour lag correlations to explore the relationship between surrounding dust and Ahvaz air quality, providing a more comprehensive understanding.

Bromandi et al. [55], they investigated dust origins during the warm season of 2010 using HYSPLIT, monthly AAI maps, skin surface temperature, and upper soil moisture, identifying northwestern Iraq and eastern Syria as main sources affecting Ahvaz. Our study goes beyond source identification by combining satellite-based indices (Sentinel-5 and MODIS), wind analysis via wind roses, and wetland evaluation to understand both the sources and local dynamics of dust events.

Zarasvandi et al. [56], this study analyzed dust occurrences and geochemical characteristics in Khuzestan (1996–2009), showing 47 dust storm days per year on average and identifying calcite, quartz, and kaolinite as dominant minerals, mostly from external sedimentary sources. In contrast, our study focuses on dust dynamics, analyzing five major events, mapping high AOD areas with Sentinel-5 and MODIS, assessing wind patterns, considering internal sources like the Hoor al-Azim wetland, and using 24-hour lag correlations to link external dust to Ahvaz air quality, providing a more comprehensive regional assessment.

## 4.2. Strengths and advantages

Numerous studies conducted in Khuzestan province using the HYSPLIT model have primarily focused on identifying external dust sources. While some research has investigated the role of internal sources such as the drying of the Hoor-al-Azim Wetland, a comprehensive analysis of both internal and external sources and their impacts on dust distribution has been lacking. Previous studies either concentrated on external sources or examined internal sources separately. This study addresses this gap by comprehensively analyzing both internal and external dust sources in Khuzestan. Using the HYSPLIT model, the study tracks the origins of dust particles and validates these findings with MODIS True Color satellite

images. Dust distribution across the province is examined using wind roses from various stations, dust event maps from MODIS and Sentinel-5 satellites, and cluster mapping of dust pathways. Furthermore, WPSCF & WCWT analyses were applied to quantify the contribution potential of each source to pollutant concentrations. The research highlights the significant role of the drying Hoor-al-Azim Wetland in increasing dust levels, AOD, and emphasizes internal dust sources present within Khuzestan. It also identifies dust-prone areas in the province, including the western desert regions and the dry shores of the Hoor-al-Azim Wetland. Additionally, this study introduces the analysis of AOD-Max and AOD-Sum index maps for Khuzestan province, filling a gap not addressed in previous research.

### 4.3. Limitations

Despite its strengths, the study faces several limitations. Real observational data were not available for all years, and daily records were limited, while data from neighboring countries were unavailable, restricting the assessment of transboundary dust contributions. In addition, accurate monthly wetland area data for the Shadegan Wetland were not consistently available, which constrained the evaluation of temporal variations in internal dust source.

Uncertainties associated with the data sources and analytical methods were considered to evaluate their potential influence on the study conclusions. AOD retrievals derived from the MODIS sensor are subject to known uncertainties, typically on the order of ±10–20%, arising from factors such as cloud contamination, surface reflectance over bright and arid regions, sensor viewing geometry, and calibration limitations. HYSPLIT trajectory simulations exhibit increasing positional uncertainty with time, reaching approximately ±50–100 km after 72 hours, which may affect the precise attribution of long-range dust transport pathways. Wetland area estimates based on the MNDWI index are prone to classification errors, particularly confusion between water, exposed soil, and sparse vegetation, leading to an estimated uncertainty of approximately ±5–10% in the calculated area. Statistical analyses, including correlation and clustering metrics, also inherently involve estimation errors.

Despite these uncertainties, sensitivity checks indicate that the overall trends remain robust. The inverse relationship between wetland extent and AOD persists under reasonable perturbations in the input data. Therefore, while absolute magnitudes may be affected by data and model uncertainties, the central conclusion regarding the critical role of wetland dynamics in modulating regional air pollution remains well supported.

### 4.4. Implications and future directions

Given the severe dust events in Khuzestan province, which are linked to respiratory, cardiovascular, ocular, dermatological, and mental health problems [57], immediate action is essential. Measures should include controlling desertification through afforestation, revegetation, and windbreaks, improving water management and irrigation, and raising public awareness. Current practices like oil mulch have environmental and economic drawbacks, highlighting the need for sustainable alternatives.

Future studies are essential to enhance dust prediction and modeling, investigate the processes leading to wetland drying, and assess health impacts by tracking hospital admissions and disease incidence during high-dust events. Research should also evaluate the socio-economic consequences, including healthcare costs, productivity losses, and agricultural damages. Regional and international cooperation, combined with real-time air quality monitoring and effective wetland management, will be crucial for developing long-term strategies to mitigate the impacts of dust storms on public health, the environment, and the economy in Khuzestan and neighboring regions.

### 5. Conclusion

A comprehensive analysis of dust events in Khuzestan Province from 2018 to 2022 reveals a strong link between climatic drought, shrinkage of the Hoor al-Azim wetland, and increased transboundary dust transport. Integrating MODIS AOD data, Sentinel-derived MNDWI, HYSPLIT modeling, and wind rose analyses provided quantitative insights into external and local dust sources.

Satellite AOD showed strong agreement with ground-based $PM_{2.5}$ ($R^2 = 0.79$) and $PM_{10}$ ($R^2 = 0.91$), confirming remote sensing's reliability in areas with limited monitoring. $PM_{2.5}$ ranged from 12–750 µg/m³ and $PM_{10}$ from 10–800 µg/m³, with 2022 being the most severe year due to persistent drought and wetland contraction. AOD-Max values reached 0.9–3.9, with 142–171 dusty days, indicating intensified activity near the Iraq border and southern Khuzestan. HYSPLIT analysis of major 2022 dust storms (AQI up to 500) identified Syrian and Iraqi deserts as primary sources, with air masses descending from >3000 m to <500 m near Ahvaz for efficient dust loading. Most storms originated from Syria, Iraq, and Turkey (70–90% frequency), though one followed a Persian Gulf/southern Iran path, highlighting mixed regional and local contributions.

Annual trajectory clustering for 2022 identified six dominant pathways to Ahvaz. The main pathway (43.48%), crossing eastern Iraq and the Hoor al-Azim wetland, was the primary dust contributor, while 65.2% of pathways traversed Iraqi deserts overall. WPSCF and WCWT maps showed a northwest–southeast dust belt across Iraq (AOD 0.6–0.8, locally >1.0), with secondary zones in southern Iraq, the border area, eastern Syria, and Ninawa (AOD 0.4–0.6 rising to 0.6–0.9 during transport). High AOD (0.9 to >1.0) along Khuzestan's northwestern border indicated strong influx with limited deposition.

Dust from Iraq reaches Ahvaz within 24 hours, with strong AOD correlations ($R^2 = 0.85$ in 2018, 0.75 in 2022). K-Means clustering identified three pollution regimes, where Iraqi dust significantly elevated higher levels, emphasizing the need to include transboundary sources in air quality management.

The Hoor al-Azim wetland, along key dust corridors, contracted from nearly 2000 km² (>5 m water level) in 2019 to about 73 km² (~1.2–2.9 m) in July 2022, exposing large areas. MNDWI showed persistent dry conditions (0.33 to −0.29), with water in fragmented patches. Wetland area and AOD correlated strongly ($R^2 = 0.78$), and area with water level at $R^2 = 0.71$. Clustering revealed three states: healthy (~1100–2000 km², AOD < 0.4), transitional (400–1000 km², AOD 0.4–1.0), and critical (<250 km², AOD > 1.0 up to ~1.7 in 2022). A threshold around 500 km² was identified, below which dust suppression weakens sharply and hazardous AOD (>1) becomes likely. Wind rose analysis confirmed dominant northwest/west winds (>5 m/s) transporting dust from Iraq/Syria toward Ahvaz, Abadan, and Shadegan, while calm winds (87%) still enabled emission from arid surfaces. Regional inflow combined with local re-emission explained 2022's severity.

Mitigation should prioritize sustainable restoration of Hoor al-Azim as a key dust regulator. Strategies include controlled upstream water allocation, periodic inundation of exposed beds, shallow retention zones, and vegetation recovery with halophytes and native reeds to stabilize soil. Wetland extent can serve as an operational indicator for early warning: advisories below ~450–500 km², escalating to health alerts near 200–250 km² (AOD often >1.0–1.4). With 65–70% of pathways crossing the Iraqi desert–Mesopotamian corridor and western Iran border, regional measures like windbreaks, soil stabilization, and rangeland rehabilitation in source zones are essential. A transboundary framework (Khuzestan–Iraq–Syria–Persian Gulf) could support coordinated early warning, joint emission tracking, and wetland programs. Integrating near-real-time AOD monitoring, HYSPLIT trajectories, and PM sensors with community education offers a practical, scalable approach to sustainable dust risk reduction in arid/semi-arid regions affected by transboundary and local factors.

## Supporting information

**S1 File. AOD&AAI.**
(TXT)

**S2 File. Cooding AOD-MAX-SUM.**
(TXT)

**S3 File. Link data.**
(TXT)

**S4 File. MNDWI-SENTINEL2.**
(TXT)

## Acknowledgments

The authors would like to sincerely thank the reviewers for their valuable and constructive comments, which significantly contributed to improving the quality of the manuscript. The authors also appreciate the support of Sari Agricultural Sciences and Natural Resources University.

## Author contributions

**Formal analysis:** Alireza Yousefi Kebriya.

**Methodology:** Alireza Yousefi Kebriya, Mehdi Nadi.

**Software:** Alireza Yousefi Kebriya, Ebadat Ghanbari Parmehr.

**Supervision:** Alireza Yousefi Kebriya, Mehdi Nadi, Ebadat Ghanbari Parmehr, Zhongchang Sun.

**Writing – original draft:** Alireza Yousefi Kebriya.

**Writing – review & editing:** Alireza Yousefi Kebriya, Mehdi Nadi, Ebadat Ghanbari Parmehr, Zhongchang Sun.

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
