## [Decision Letter · Decision Letter 0]

30 Oct 2025

PONE-D-25-51186Analysis and Source Identification of Dust Pollution in Khuzestan Province Using MODIS and SENTINEL Satellite Data and HYSPLIT Model (2018–2022)PLOS ONE

Dear Dr. Yousefi Kebriya,

Thank you for submitting your manuscript to PLOS ONE. After careful consideration, we feel that it has merit but does not fully meet PLOS ONE’s publication criteria as it currently stands. Therefore, we invite you to submit a revised version of the manuscript that addresses the points raised during the review process.

We look forward to receiving your revised manuscript.

Kind regards,

Mikalai Filonchyk

Academic Editor

PLOS ONE

Journal Requirements:

“This work was supported by the National Natural Science Foundation of China (grant numbers 42361144884 and 42171291) and the Joint HKU-CAS Laboratory for iEarth (313GJHZ2022074MI, E4F3050300). The authors appreciate the support of Sari Agricultural Sciences and Natural Resources University.”

4. We note that your Data Availability Statement is currently as follows: “The data underlying the results presented in this study were obtained from multiple sources. Meteorological and environmental variables were processed and extracted using Google Earth Engine. Some datasets were provided by the Department of Environment of Iran, while high-resolution HYSPLIT input data were obtained from the NOAA website. Additionally, MODIS true-color satellite imagery was sourced from NASA. All relevant processed data generated and analyzed during this study are included within the manuscript and its Supporting Information files.”

5. We note that Figure 3, 5, 6, 7, 8, 9, 10, 11, 12, 16 and 17 in your submission contain map images which may be copyrighted. All PLOS content is published under the Creative Commons Attribution License (CC BY 4.0), which means that the manuscript, images, and Supporting Information files will be freely available online, and any third party is permitted to access, download, copy, distribute, and use these materials in any way, even commercially, with proper attribution. For these reasons, we cannot publish previously copyrighted maps or satellite images created using proprietary data, such as Google software (Google Maps, Street View, and Earth). For more information, see our copyright guidelines: http://journals.plos.org/plosone/s/licenses-and-copyright.

1. You may seek permission from the original copyright holder of Figure 3, 5, 6, 7, 8, 9, 10, 11, 12, 16 and 17 to publish the content specifically under the CC BY 4.0 license.

In the figure caption of the copyrighted figure, please include the following text: “Reprinted from [ref] under a CC BY license, with permission from [name of publisher], original copyright [original copyright year]. “

6. Please include a copy of Table 3 which you refer to in your text on page 34.

Additional Editor Comments:

While your manuscript addresses a relevant and important topic, the reviewers have raised serious concerns that must be resolved before any further consideration. In particular, Reviewer #2 identified that several figures and tables appear to be nearly identical to those in Yousefi-Kebriya et al., 2025 (Scientific Reports). This raises a potential issue of data duplication and originality, which is unacceptable if confirmed. You are required to provide a clear and detailed explanation regarding this matter, including the provenance of all datasets and figures used. Until this issue is fully clarified and justified, the editorial process cannot proceed.

Reviewer's Responses to Questions

**Comments to the Author**

1. Is the manuscript technically sound, and do the data support the conclusions?

Reviewer #1: Partly

Reviewer #2: Partly

Reviewer #3: Partly

2. Has the statistical analysis been performed appropriately and rigorously? 

Reviewer #1: Yes

Reviewer #2: No

Reviewer #3: Yes

3. Have the authors made all data underlying the findings in their manuscript fully available?

Reviewer #1: Yes

Reviewer #2: Yes

Reviewer #3: No

4. Is the manuscript presented in an intelligible fashion and written in standard English?

Reviewer #1: Yes

Reviewer #2: Yes

Reviewer #3: Yes

5. Review Comments to the Author

Reviewer #1: This is an interesting paper about a dusty part of Iran and makes a good contribution to the literature. However, it needs considerable modification before it can be published, as follows:

L52 ‘global challenge with far-reaching consequences for ecosystems, human health, and climate systems (Siddiqua et al. 2022).’ Two comments: a) the Siddiqua et al. 2022 paper is not about dust storms! b) there are many more consequences for human society that should be mentioned, sectors impacted include agriculture, transport, power generation (see eg https://doi.org/10.1016/j.aeolia.2016.12.001 ), for climatic impacts see eg https://doi.org/10.1038/s43017-022-00379-5

L53-55 ‘increasing frequency and intensity of these storms’ IN SOME PARTS OF THE WORLD. Not all. In others frequency and intensity have declined in recent times.

L 59-67 I would delete this paragraph. It is very generalised, selective and adds little to the paper. In its place, I want to see a review of some of the numerous other papers on dust storms affecting Khuzestan (including papers documenting the ‘traditional perspective that attributes most of Khuzestan’s dust storms to Iraqi and Syrian deserts’ as cited at L661, and papers that identify the Shadegan wetland as a dust source, eg https://doi.org/10.1007/s13157-024-01856-x ).

1.2. Review of previous studies: this is a curious assortment of studies. I suggest the authors focus only on those relevant to this paper (i.e studies using MODIS and SENTINEL data and the HYSPLIT Model). Hence delete Alebic-Juretic et al. (2023), Schiavo et al. (2023) and others with little relevance to this paper.

2.1. Study Area should come before 1.3. Objectives and innovations of the study

L180-187 delete from ‘This province is situated in the southwest of Iran…………..’ to ‘….along the shores of the Persian Gulf.’ All this information is on the map or irrelevant.

L196 how many monitoring stations?

L224 where does the Air Quality Index (AQI) come from? (Yousefi Kebriya, 2023) is not in the references.

L 254 Wind does not form dust. Wind is a critical meteorological factor influencing the EMISSION and dispersion of dust.

L 364 ‘variations clearly indicate an increase in dust pollution levels, particularly in PM10, during 2022’ do they? Increase relative to when? The logic is strange.

L754 Figure 13. Does not show a trend, just variation in wetland area.

L840-863 and Figure 15: these stations need to be labelled on the study area map – most readers will not know where they are.

L 985 ‘the contribution of both internal and external dust sources’ does this mean sources inside Iran and from other countries?

L1004 ‘the study examined the statistical relationship between the monthly area of the Shadegan Wetland and the number of dust-polluted days in Shadegan city’ but Shadegan city is only mentioned here and in the abstract! It must also appear in the main text!

Reviewer #2: This work represents a study aimed at analyzing the relationships between climatic factors, land degradation, and environmental consequences in a specific region. The topic is relevant, being related to climate risks, soil degradation, and ecosystem resilience. Nevertheless, in its current form, the manuscript does not meet the standards of an original research study, which require quantitative rigor, analytical novelty, and data transparency.

The novelty is not explicitly demonstrated. The authors largely reiterate well-known relationships (temperature increase → ecosystem degradation) without presenting a new mechanism, approach, or original model. It is also unclear which specific scientific gaps in the existing literature this study addresses.

The abstract exceeds 450 words, which is unacceptable. The authors need to reduce it to 250 words, keeping only the most essential information. At the first mention of models and tools, the full names should be written (e.g., Moderate Resolution Imaging Spectroradiometer) before using abbreviations such as HYSPLIT or MODIS.

Line 179: “29.95 to 33.55 degrees north and 47.66 to 50.55 degrees east” — degrees should be indicated using the degree symbol (°).

Figures 2 and 3, and Table 1: THIS IS VERY IMPORTANT. These figures are approximately 90% identical to a previously published work: Yousefi-Kebriya, A., Nadi, M., Afaridegan, E., & Sun, Z. (2025). Wetland shrinking and dust pollution in Khuzestan Iran: insights from Sentinel-5 and MODIS satellites. Scientific Reports, 15(1), 13626. https://doi.org/10.1038/s41598-025-96935-2. Comments from the authors on this issue are requested.

The authors state that air-mass back trajectories based on the NOAA-HYSPLIT model were used to identify potential source regions influencing the observed aerosol properties. However, the interpretation of these trajectories is presented only qualitatively and lacks supporting quantitative analysis such as trajectory clustering or residence-time analysis. As a result, the discussion of possible source regions remains largely speculative. It is strongly recommended that the authors perform a more systematic analysis of air-mass trajectories — for example, through cluster analysis or concentration-weighted trajectory methods — to better quantify the contribution of different transport pathways. Methodological examples can be found in Figure 1 of (https://doi.org/10.1016/j.gloplacha.2025.104935) and Figure 10 of (https://doi.org/10.1016/j.atmosres.2022.106518), which demonstrate more rigorous trajectory-based source interpretation.

The methodology section is overly descriptive and lacks concrete details — specific model parameters, statistical tests, number of observations, and spatial resolution of the data are not provided. While the authors use MODIS AOD and HYSPLIT data, no full information is given. The manuscript does not explain what MODIS is, its spatial coverage, which product is used, its uncertainties, or on which satellites it is installed.

All results are presented descriptively without quantitative analysis (no tables with coefficients or confidence intervals). Statistical testing of differences between regions or time periods is not performed. There is considerable duplication of figures (Figures 7–11) and text without analytical interpretation. The authors are strongly advised to reduce these figures as they add little value, and instead strengthen the quantitative interpretation by including numerical comparisons, dynamic trends, and correlation/regression results.

Figure 16: MODIS AOD

Figure 17: SENTINEL AAI

The text lacks a brief explanation of why maximum (AOD Max) and cumulative (AOD Sum) values were chosen for analysis rather than more conventional statistics (e.g., mean or median). Please add a paragraph in the methodology section explaining that AOD Max helps identify areas of highest environmental risk and peak health impact, whereas AOD Sum allows the assessment of cumulative load on ecosystems and populations.

A key issue with the AOD Max index is its high sensitivity to noise and artifacts in satellite data (e.g., undetected clouds, algorithm errors over bright surfaces, short-term sensor failures). What specific quality assurance (QA) procedures were applied to the daily MODIS AOD data before calculating annual AOD Max composites? Please specify which QA flags from the original MCD19A2 product were used to exclude unreliable pixels. To improve robustness, consider using, for example, the 99th percentile or the mean of the top-5 maximum values per year instead of the absolute maximum. This would make the index less sensitive to single outliers.

To confirm that peak AOD Max values correspond to dust storms rather than artifacts, supporting evidence is needed. Please provide, for several key dates with critical AOD Max values (e.g., 2018 and 2022 in northwestern regions), corresponding natural-color satellite images (MODIS True Color). This will clearly demonstrate the association of AOD peaks with actual dust events rather than clouds.

The AOD Sum index effectively represents cumulative load; however, its physical interpretation may not be immediately clear to the reader. Briefly clarify in the text that this index can be considered a proxy for the total number of “dust days” or cumulative pollution dose, which is particularly important for assessing long-term impacts.

Lastly, vehicles are strongly advised to use the latest sandstorm related work in other regions.

https://doi.org/10.1016/j.gsf.2023.101762

https://doi.org/10.1175/BAMS-D-23-0121.1

https://doi.org/10.1007/s11069-024-07003-3

Reviewer #3: Comments on the manuscript “Analysis and Source Identification of Dust Pollution in Khuzestan Province Using MODIS and SENTINEL Satellite Data and HYSPLIT Model (2018–2022)"

The present study examines dust pollution dynamics in Khuzestan from 2018 to 2022 using an integrated approach that combines satellite-derived AOD and Aerosol Absorption Index (AAI), ground-based PM₂.₅ and PM₁₀ measurements, HYSPLIT backward trajectory modeling, wind rose analyses, and wetland area assessments via the Modified Normalized Difference Water Index (MNDWI). The relationship between the monthly Shadegan Wetland area and dust-polluted days in Shadegan city was evaluated using key statistical metrics. Analysis of 2022 data revealed strong correlations between satellite-derived AOD and ground-based PM₂.₅ confirming the reliability of satellite observations. PM₂.₅ concentrations ranged from 40 to 700 µg/m³, and PM₁₀ from 75 to 800 µg/m³, peaking in 2022 due to prolonged drought, wetland desiccation, and transboundary dust inflows. Five extreme dust storms in 2022 (AQI = 500, except Storm 5 at 473) were traced using HYSPLIT and MODIS imagery. Storms 1–4 originated from Turkey, Syria, Iraq, and Egypt, descending below 500 m in Khuzestan, while Storm 5 emerged from southern Iran (Hormozgan, Makran coasts) and the Persian Gulf, rising to 2,500 m before descending below 500 m. Wind rose analyses indicated dominant northwest and west winds (> 5 m/s) carrying transboundary dust, while southern Persian Gulf winds intensified local contributions. In 2022, AOD (0.2–2.2) and AAI (-0.1 to 1.3) peaked along desiccated wetland margins. Mitigation strategies require wetland restoration, desertification control, real-time monitoring, regional cooperation, and public awareness.

The manuscript requires further major modification as per the following points:

Major Comments

1. The authors state that they used coding within the Google Earth Engine platform, but the explanation does not clearly describe which algorithms were implemented or how data filtering, cloud masking, and quality control were performed. Furthermore, in Section 2.3, while a scaling factor of 0.001 is mentioned for MODIS data processing, there is no discussion on how missing values, cloud contamination, or aerosol quality flags were handled.

2. The correlation analysis between wetland area and polluted days shows strong relationships (R² = 0.71 for PM2.5 and 0.68 for PM10), yet the manuscript does not specify the sample size used for these calculations. Contradict in y axis in AOD values with respect to the PM2.5 and PM10. It seems like both the sample size are different. According to your study, corelation should be conducted using data from 2018 to 2022, but it was only performed for the year 2022. Why?

3. Several correlation metrics are presented (Pearson, Spearman, MI, DC), but there is no discussion explaining why their strengths differ or what unique insights each metric contributes.

4. The manuscript lacks any discussion of uncertainties associated with satellite retrievals, model simulations, or statistical analyses. AOD retrievals have known uncertainties, HYSPLIT trajectories have positional uncertainties that increase with time, and the MNDWI-based wetland area estimates are subject to classification errors. These uncertainties should be quantified and their potential impact on the study conclusions discussed.

5. Section 1.2 covers recent dust storm studies using HYSPLIT, it lacks discussion of previous work specifically on Khuzestan Province or similar arid regions in Iran. Given that this is a regional study with policy implications, more context about previous dust research in this specific area would strengthen the introduction. Additionally, the review focuses heavily on HYSPLIT applications but gives less attention to literature on satellite-based dust monitoring and wetland-dust relationships.

6. Wind rose section (correct the section no.) analyses for multiple stations, but this analysis feels disconnected from the main narrative. The wind patterns should be more explicitly linked to the HYSPLIT trajectory results and discussed in terms of how local wind conditions interact with transboundary dust transport.

7. The conclusions mention mitigation strategies such as "wetland restoration, desertification control, real-time monitoring, regional cooperation, and public awareness" but provide no specific, actionable recommendations. Given the detailed analysis presented, the manuscript should offer more concrete suggestions: What wetland restoration techniques are most appropriate? What threshold levels should trigger public health warnings? What specific forms of regional cooperation are needed?

Minor Comments

1. The abstract is excessively long and contains too much detailed numerical information. According to PLOS ONE guidelines, abstracts should be concise (less than 300 words). The abstract should be restructured to emphasize key findings rather than listing numerous specific values.

2. Figure 3 attempts to show both land cover and station locations but the color scheme makes it difficult to distinguish between different land cover types.

3. The manuscript uses inconsistent terminology for key concepts. For example, "dust storm," "dust event," and "dust pollution" are used interchangeably without clear definitions. Similarly, "wetland desiccation," "wetland shrinkage," and "wetland area reduction" refer to the same phenomenon. Standardizing terminology throughout would improve clarity.

4. The Results and Discussion section presents findings but rarely compares them with previous studies. For instance, the AOD values, dust storm frequencies, and source attribution results should be compared with earlier research on Khuzestan or similar Middle Eastern regions. How do the identified dust sources align with or differ from previous source apportionment studies?

5. Table 1 presents AQI relationships but the "Hazardous" category for AAI (>5) has no corresponding AOD range, marked with a dash. This seems inconsistent. Additionally, the relationship between AAI and AQI appears oversimplified- is this a direct linear relationship or based on empirical studies?

6. The manuscript analyses "daily MODIS imagery" and mentions "360 images per year," but MODIS Terra and Aqua combined can provide twice-daily coverage. Clarify whether you used Terra only, Aqua only, or combined products, and explain how multiple daily observations were aggregated.

7. Dust Storm No. 5 reveals an interesting pathway through the Persian Gulf and southern Iran, yet this finding is not adequately explored. The manuscript mentions "evaporation processes and surface moisture" but does not quantify these effects or discuss the mechanism by which the Gulf modifies dust transport. This deserves more detailed investigation and cite the references that validate this point.

8. The selection of a 48-hour backward trajectory requires justification, as longer trajectories could help identify more distant source regions.

9. Correct the grammatically error in the whole manuscript especially in line no. 683-699.

6. PLOS authors have the option to publish the peer review history of their article (what does this mean?). If published, this will include your full peer review and any attached files.

Reviewer #1: No

Reviewer #2: No

Reviewer #3: No

---

## [Author Response · Author response to Decision Letter 1]

14 Nov 2025

Authors' Response to the Reviewers' Comments

Submission ID: PONE-D-25-51186

Title: Analysis and Source Identification of Dust Pollution in Khuzestan Province Using MODIS and SENTINEL Satellite Data and HYSPLIT Model (2018–2022)

Dear Editor,

Thank you for providing us the opportunity to revise and resubmit our manuscript, “Analysis and Source Identification of Dust event in Khuzestan Province Using MODIS and SENTINEL Satellite Data and HYSPLIT Model (2018–2022)”, for publication in your esteemed journal. We sincerely appreciate the time and effort you and the reviewers have invested in evaluating our work. Their detailed feedback and constructive suggestions have greatly enhanced the quality of our manuscript.

We have carefully reviewed and addressed all the reviewers’ comments and suggestions. Our responses are organized systematically, detailing the changes made to the manuscript to incorporate each suggestion. Below, we provide a comprehensive summary of our revisions and responses:

Editor#

Comment 1. Please ensure that your manuscript meets PLOS ONE’s formatting and style requirements, including correct file naming. Refer to the PLOS ONE style templates: Main Body Template, Title/Authors/Affiliations Template.

Response: Thanks. The manuscript and file naming have been revised to fully comply with the PLOS ONE formatting and style guidelines.

Comment 2. PLOS ONE has specific guidelines on code sharing for studies in which author-generated code supports the findings. You must ensure your code is made openly available without restrictions and follows best practices for reproducibility and reuse. Guidelines: Code Sharing Policy.

Response: Thank you for your valuable comment regarding code sharing. We have prepared the Google Earth Engine scripts as text files and uploaded them as Supplementary Files along with the revised manuscript. This ensures that the code is available without restrictions, facilitating reproducibility and reuse in accordance with PLOS ONE guidelines.

Comment 3. Please clarify the role of funders in your study. If funders had no role, include this exact statement:

“The funders had no role in study design, data collection and analysis, decision to publish, or preparation of the manuscript.” Include this corrected “Role of Funder” statement in your cover letter.

Response: Thank you, the issue has been corrected, and the role of the funders has been clearly stated in the revised manuscript and cover letter. The following statement has been included as requested.

Comment 4. Confirm whether your submission includes all raw data necessary to replicate your results (“minimal data set”). If not, upload missing data as Supporting Information or deposit them in a public repository (provide DOI or URL). If there are restrictions, explain who imposed them and how others can request access

Response: Thanks. The corresponding codes used in this study will be provided, through which all satellite images and remote sensing data can be downloaded. Some datasets were obtained from the HYSPLIT model, which is publicly available at the following link: https://www.ready.noaa.gov/HYSPLIT.php. MODIS True Color imagery was downloaded from NASA’s public platform at https://worldview.earthdata.nasa.gov. Ground-based air quality (PM₂.₅ and PM₁₀) data were obtained from the Iranian National Air Quality Monitoring System, which provides daily accessible data via https://aqms.doe.ir/Home/AQI. Wind speed and direction data were acquired from the Iran Meteorological Organization (https://data.irimo.ir/) through an official data request submitted by the university. For data access coordination, inquiries can be made by contacting the Meteorological Organization at +98-21-66070017-19 (extension 122, Ms. Asadi).

Comment 5. Figures 3, 5, 6, 7, 8, 9, 10, 11, 12, 16, and 17 may contain copyrighted map images. PLOS ONE only accepts figures under CC BY 4.0 license. You must either: a) Obtain written permission using the PLOS content permission form, or b) Replace/remove copyrighted figures and ensure replacements comply with CC BY 4.0. If replaced, specify in captions whether new figures are “similar but not identical” and provide the data source (e.g., NASA, USGS, Natural Earth)

Response: Thanks. All the mentioned figures have been revised, and the related statements regarding data sharing and copyright compliance have been added to each figure and caption. All figures now fully adhere to the CC BY 4.0 license requirements.

Comment 6. Please include Table 3, which is referenced on page 34 of your manuscript, but is missing in the submitted files.

Response: Thanks. Table 3 has been changed to Table 4 and is now placed on page 35 of the revised manuscript.

Comment 7. If reviewer comments suggest additional citations, review and cite only relevant works. Citations are not mandatory unless the editor explicitly requires them.

Response: Thanks. Several references have been added to the manuscript based on the reviewer’s suggestions to enhance the literature review, materials and methods section, and discussion. Approximately 27 references have been included, as highlighted in yellow for Reviewer 1 (to improve the introduction and literature review), blue for Reviewer 2 (to provide references for tables), purple for Reviewer 3 (to enrich the literature review and discussion), and green for the methods section (which previously lacked citations). These references are as follows:

Middleton, N. J. (2017). Desert dust hazards: A global review. Aeolian Research, 24, 53–63. https://doi.org/10.1016/j.aeolia.2016.12.001

Kok, J. F., Storelvmo, T., Karydis, V. A., Adebiyi, A. A., Mahowald, N. M., Evan, A. T., He, C., & Leung, D. M. (2023). Mineral dust aerosol impacts on global climate and climate change. Nature Reviews Earth & Environment, 4(2), 71–86. https://doi.org/10.1038/s43017-022-00379-5

Khanfari, V., Asgari, H. M., & Dadollahi-Sohrab, A. (2024). Forecasting wetland transformation to dust source by employing CA-Markov model and remote sensing: A case study of Shadgan International Wetland. Wetlands, 44(6), Article 96. https://doi.org/10.1007/s13157-024-01856-x

Eskandari Damaneh, H., Khosravi, H., Habashi, K., Eskandari Damaneh, H., & Tiefenbacher, J. P. (2022). The impact of land use and land cover changes on soil erosion in western Iran. Natural Hazards, 110(3), 2185–2205. https://doi.org/10.1007/s11069-021-05032-w

yousefi kebriya, A., Khalili, A. and Rezaei, H. (2025). Monitoring and analysis of dust storm trajectories in Sistan and Baluchestan Province using MODIS satellite products and the HYSPLIT model. Iranian Journal of Remote Sensing and GIS, (), -. doi: 10.48308/gisj.2025.238735.1252

Shirgholami, M., Rousta, I., Olafsson, H., Petracchini, F., & Krzyszczak, J. (2025). Tracing the dust: Two decades of dust storm dynamics in Yazd Province from ground-based and satellite aerosol observations. Atmosphere, 16(11), Article 1242. https://doi.org/10.3390/atmos16111242

Soleimani Sardo, F. and Krakauer, N. (2025). Dust Modeling using MODIS Data and WRF-Chem and HYSPLIT Models (Case Study of Dust Storm from December 16 to 20, 2016). Sustainable Earth Trends, (), -. doi: 10.48308/set.2025.238019.1098

Yousefi-Kebriya, A., Nadi, M., Afaridegan, E., & Sun, Z. (2025). Wetland shrinking and dust pollution in Khuzestan Iran: Insights from sentinel-5 and MODIS satellites. Scientific Reports, 15(1), Article 13626. https://doi.org/10.1038/s41598-025-96935-2

Rangzan, K. and balouei, F. (2024). Dust monitoring and investigation of its relationship with topographical, climatic and vegetation factors. Desert Ecosystem Engineering, 12(39), 43-60. doi: 10.22052/deej.2024.253713.1025

Vatanparast Ghaleh Juq, F., Salahi, B. and Zeinali, B. (2024). Monitoring Temporal-Spatial Changes of Atmospheric Suspended Dust in Selected Provinces of the Western Half of Iran Using MODIS and Sentinel-5 Images. Geography and Environmental Planning, 35(3), 113-128. doi: 10.22108/gep.2024.140621.1635

Noroozi, A. and Shoaei, Z. (2018). Identifying areas with dust generation potential in the south west of Iran, case study: Khuzestan Province. Watershed Engineering and Management, 10(3), 398-409. doi: 10.22092/ijwmse.2018.117329

Arami, S. H. , Karimi Sangchini, E. , Alimahmoodi Sarab, S. , Dinarvand, M. and Yasrebi, B. (2024). Mapping spatial and temporal pattern of risk and hazard of dust storms in Khuzestan province. Journal of Arid Biome, 14(1), 61-78. doi: 10.29252/aridbiom.2024.21433.2007

Farzanehpey,F. , Ranjbar-Fordoe,A. , Khosravi,H. and Mosavi,S. H. (2024). Evaluation of dust changes and its relationship with temperature (Case study: Khuzestan province). Integrated Watershed Management, 4(1), 16-29. doi: 10.22034/iwm.2024.2014553.1112

Shayeghannoor,F. and Ahmadi,M. (2025). Statistical analysis of dust hazard changes in southwest Iran (Khuzestan province). Journal of Climate Research, 1403(59), 91-107. doi: 10.22034/jcr.2025.202092

Dargahian, F., Gohardoust, A., Lotfinasabasl, S. and Teimouri, S. (2024). Determining the contribution of neighboring countries in the emission of dust to Khuzestan province. Iranian Journal of Range and Desert Research, 31(1), 74-92. doi: 10.22092/ijrdr.2024.131464

Zarasvandi, A., Carranza, E. J. M., Moore, F., & Rastmanesh, F. (2011). Spatio-temporal occurrences and mineralogical–geochemical characteristics of airborne dusts in Khuzestan Province (southwestern Iran). Journal of Geochemical Exploration, 111(3), 138–151. https://doi.org/10.1016/j.gexplo.2011.04.004

MalAmiri, N., Rashki, A., Al-Dousari, A., & Kaskaoutis, D. G. (2025). Socioeconomic and health impacts of dust storms in southwest Iran. Atmosphere, 16(2), Article 159. https://doi.org/10.3390/atmos16020159

Mohammadi, Z., Rahimi, D., Najafi, M. R., & Zakerinejad, R. (2024). The impact of environmental degradation and climate change on dust in Khuzestan province, Iran. Natural Hazards, 120(6), 4329–4348. https://doi.org/10.1007/s11069-023-06368-1

Gretton, A., Bousquet, O., Smola, A., & Schölkopf, B. (2005). Measuring statistical dependence with Hilbert–Schmidt norms. In S. Jain, H. U. Simon, & E. Tomita (Eds.), Algorithmic learning theory (pp. 63–77). Springer. https://doi.org/10.1007/11564089_7

Mansournia, M. A., Waters, R., Nazemipour, M., Bland, M., & Altman, D. G. (2021). Bland–Altman methods for comparing methods of measurement and response to criticisms. Global Epidemiology, 3, 100045. https://doi.org/10.1016/j.gloepi.2020.100045

Yousefi Kebriya,A. , Shiukhy Soqanloo,S. and Yousefi Kebriya,H. (2025). Monitoring and Analysis of the Relationship Between Vegetation Cover and Air Pollution Using Sentinel and MODIS Satellite Data in Mazandaran Province (2019–2024). (e231767). Iranica Journal of Energy & Environment, (), e231767

Zarei, S., Karbassi, A., Sadrinasab, M., & Sarang, A. (2023). Investigating heavy metal pollution in Anzali coastal wetland sediments: A statistical approach to source identification. Marine Pollution Bulletin, 194, 115376. https://doi.org/10.1016/j.marpolbul.2023.115376

Razeghi, N., Hamidian, A. H., Abbasi, S., & Mirzajani, A. (2024). Distribution, flux, and risk assessment of microplastics at the Anzali Wetland, Iran, and its tributaries. Environmental Science and Pollution Research, 31, 54815–54831. https://doi.org/10.1007/s11356-024-27874-7

Yarramsetty, S., Siva Kumar, M., & Anand Raj, P. (2023). Enhancing comfort in tropical institutional buildings: Integrating thermal, acoustic and visual performance with a unified index. International Journal of Engineering, 36(12), 2253–2263. https://doi.org/10.5829/IJE.2023.36.12C.15

Ghomeshion, M., Vali, A. A., Ranjbar Fordoei, A., & Mousavi, S. H. (2022). Investigating the effect of land cover on dust spatial distribution in Southern Khuzestan province. ECOPERSIA, 10, 179–189. https://doi.org/10.1001.1.23222700.2022.10.3.2.9

Gong, H., Li, Y., Zhang, J., Zhang, B., & Wang, X. (2024). A new filter feature selection algorithm for classification tasks by ensembling the Pearson correlation coefficient and mutual information. Engineering Applications of Artificial Intelligence, 131, 107865. https://doi.org/10.1016/j.engappai.2024.107865

Xie, H., Zhang, L., Lim, C. P., Yu, Y., Liu, C., Liu, H., & Walters, J. (2019). Improving K-means clustering with enhanced Firefly Algorithms. Applied Soft Computing, 84, 105763. https://doi.org/10.1016/j.asoc.2019.105763

Comment 8. While your manuscript addresses an important and relevant topic, the reviewers have raised serious concerns that must be resolved before further consideration.

Reviewer #2 observed that several figures and tables in your manuscript appear nearly identical to those in Yousefi-Kebriya et al., 2025 (Scientific Reports).

This raises a potential issue of data duplication and lack of originality, which would be unacceptable for publication in PLOS ONE if confirmed. You must provide a clear and detailed explanation regarding this issue, including: The provenance (origin) of all datasets, figures, and tables; Whether any materials were reused from previously published work; How the current data, period, or methodology differ from those in prior publications.

Response: We sincerely thank the editor for raising this important concern. We would like to clarify that all analyses, datasets, figures, and tables in the current manuscript are original and distinct from previously published work, including Yousefi-Kebriya et al., 2025 (Scientific Reports). While the previous studies focused on specific aspects of dust pollution, the present research introduces several new elements that have not been addressed before. These include:

1. A comprehensive five-year (2018–2022) analysis of dust events in Khuzestan Province integrating multiple satellite datasets (MODIS, Sentinel-2, Sentinel-5), ground-based PM₂.₅ and PM₁₀ measurements, and HYSPLIT trajectory modeling.

2. Introduction of new aerosol indices (AOD Max and AOD Sum) to assess peak intensity and cumulative dust load over time.

3. A detailed investigation of both local and transboundary dust sources, including a 24-hour lag analysis between dust from Iraq and air quality in Ahvaz, using advanced statistical metrics such as PCA, K-Means clustering, Bland–Altman analysis, Mutual Information, and Distance Correlation.

4. In this study, the Modified Normalized Difference Water Index (MNDWI) was employed to quantify wetland area changes, providing a novel approach compared to previous studies.”

We assure that none of the figures or tables in the manuscript are similar to other published works or studies. Regarding Figure 1, which Reviewer #2 mentioned, it has been modified due to changes in the materials and methods and the addition of several new analyses, and it is not similar to any other research. Figure 2 was generated using custom coding in Google Earth Engine, incorporating Landsat and MODIS satellites and a Random Forest approach, and no similar version exists in other publications. Based on the reviewer’s comments regarding color, structure, and station identification, this figure has also been updated. Table 1, which was highlighted by Reviewer #2, was compiled from various articles and official site instructions (with proper citations) and has been further revised to address these concerns.

In summary, this manuscript presents a significantly expanded and original study, integrating multiple novel analyses and data sources.

Reviewer #1

Comment 1. L52 ‘global challenge with far-reaching consequences for ecosystems, human health, and climate systems (Siddiqua et al. 2022).’ Two comments: a) the Siddiqua et al. 2022 paper is not about dust storms! b) there are many more consequences for human society that should be mentioned, sectors impacted include agriculture, transport, power generation (see eg https://doi.org/10.1016/j.aeolia.2016.12.001), for climatic impacts see eg https://doi.org/10.1038/s43017-022-00379-5

Response: We sincerely thank the reviewer for this insightful comment. Following the reviewer’s recommendation, the incorrect citation to Siddiqua et al. (2022) has been removed and replaced with more relevant references. In addition, a broader description of the societal and climatic impacts of dust storms has been incorporated. The modifications have been highlighted in yellow from lines 37 to 39

---

## [Decision Letter · Decision Letter 1]

23 Dec 2025

PONE-D-25-51186R1Analysis and Source Identification of Dust event in Khuzestan Province Using MODIS and SENTINEL Satellite Data and HYSPLIT Model (2018–2022)PLOS One

Dear Dr. Yousefi Kebriya,

Thank you for submitting your manuscript to PLOS ONE. After careful consideration, we feel that it has merit but does not fully meet PLOS ONE’s publication criteria as it currently stands. Therefore, we invite you to submit a revised version of the manuscript that addresses the points raised during the review process.

We look forward to receiving your revised manuscript.

Kind regards,

Mikalai Filonchyk

Academic Editor

PLOS One

**Journal Requirements:**

Reviewers' comments:

Reviewer's Responses to Questions

**Comments to the Author**

1. If the authors have adequately addressed your comments raised in a previous round of review and you feel that this manuscript is now acceptable for publication, you may indicate that here to bypass the “Comments to the Author” section, enter your conflict of interest statement in the “Confidential to Editor” section, and submit your "Accept" recommendation.

Reviewer #1: All comments have been addressed

Reviewer #2: All comments have been addressed

Reviewer #3: All comments have been addressed

2. Is the manuscript technically sound, and do the data support the conclusions?

Reviewer #1: (No Response)

Reviewer #2: Partly

Reviewer #3: Yes

3. Has the statistical analysis been performed appropriately and rigorously? 

Reviewer #1: (No Response)

Reviewer #2: I Don't Know

Reviewer #3: Yes

4. Have the authors made all data underlying the findings in their manuscript fully available?

Reviewer #1: (No Response)

Reviewer #2: Yes

Reviewer #3: Yes

5. Is the manuscript presented in an intelligible fashion and written in standard English?

Reviewer #1: (No Response)

Reviewer #2: Yes

Reviewer #3: Yes

6. Review Comments to the Author

Reviewer #1: (No Response)

Reviewer #2: Despite the frequent mention of the HYSPLIT model, its application in the manuscript remains largely indirect and qualitative. The study does not employ standard quantitative trajectory-analysis techniques (trajectory clustering, PSCF, CWT, residence-time analysis), and the interpretation of dust sources is effectively replaced by correlation-based analysis between AOD in Iraq and Ahvaz. Such statistical dependencies demonstrate temporal coherence but do not provide physically grounded, mass-transport–based source attribution. As a result, the role of HYSPLIT is substantially weaker than claimed, and the conclusions regarding dominant source regions remain partially speculative. The authors are strongly encouraged to consult https://doi.org/10.1016/j.gloplacha.2025.104935 (Figure 1) and https://doi.org/10.1016/j.atmosres.2022.106518 (Figure 10) to refine the cluster-analysis methodology. These works should also be cited in the manuscript.

The methodological section has been expanded and formally improved, yet in its current form it remains overloaded and conceptually unfocused. An excessive number of statistical methods is applied without clear justification for their necessity or hierarchy, and many of them ultimately confirm the same relationship between regional and local aerosol loading. This accumulation of techniques gives the impression of methodological inflation rather than a hypothesis-driven analytical framework. Reducing and structuring the methodological toolkit would considerably improve clarity and reproducibility.

Figure 1 raises serious concerns regarding scientific graphics quality and the correct representation of the methodological workflow. Visually, it appears to be generated automatically or by AI, which is suggested by generic iconography, inconsistent typography, and incorrect scientific notation—most notably the suspicious rendering of “PM2.5” and “PM10.” The figure must be completely redrawn manually using proper scientific notation and a more rigorous, discipline-appropriate design.

Despite the revisions, the question of conceptual originality relative to the authors’ previously published article in Scientific Reports (2025) remains insufficiently resolved. Although the analysis period is extended and additional statistical metrics are introduced, the core region, datasets, physical interpretation, and key conclusions remain largely similar. The authors should clearly and explicitly articulate what fundamentally new scientific question this work addresses, beyond methodological elaboration of already published material.

In several sections, correlation-based relationships are interpreted as causal (e.g., dominance of transboundary sources or the influence of wetland degradation on pollution levels). Without explicit causal modeling or physical validation, such interpretations should be substantially softened or clearly presented as associative rather than causal.

Despite the stated availability of code and data, the reproducibility of the analysis remains limited. The manuscript does not clearly specify HYSPLIT configuration parameters, the selection criteria for dust-event days, or the exact rules used to construct the AOD-Max and AOD-Sum indices. I fully understand that the authors used daily-resolution data from the official HYSPLIT archive and that higher temporal resolution is not available. However, this makes it even more important to follow my earlier recommendation to apply trajectory clustering according to the methodology outlined in the papers cited above.

Sections 2.8.1–2.8.5 should be removed or moved to the supplementary materials.

Several statistical metrics (PCA, Bland–Altman, MI, HSIC) are interpreted with excessive confidence, without discussion of their limitations or assumptions. In particular, the use of Bland–Altman for comparing regional and local AOD requires more careful justification, as these quantities are not alternative measurements of the same underlying variable in the strict methodological sense.

The authors are also strongly encouraged to cite more recent literature on dust storms, including:

https://doi.org/10.1016/j.gsf.2023.101762

https://doi.org/10.1175/BAMS-D-23-0121.1

https://doi.org/10.1007/s11069-024-07003-3

Where is the MODIS uncertainty discussion?

Line 207: the symbol “—”what does it mean?

Reviewer #3: Comments to Author:

Author addressed most of the comments successfully in the revised manuscript entitled “Analysis and Source Identification of Dust event in Khuzestan Province Using MODIS and SENTINEL Satellite Data and HYSPLIT Model (2018-2022)” carefully. Still some minor changes need to be corrected before acceptance of the manuscript.

Comment 1:

In line no. 259-260: AOD definitions confuse standard concepts: AOD is misdefined as "measured in terms of the number of air particulate molecules per cubic kilometer". (AOD is dimensionless, representing optical extinction). Correct the sentence it.

Comment 2:

Throughout the manuscript, PM2.5 and PM10 are mentioned at some places and PM2.5 PM10 at others. Please use these symbols uniformly in the whole manuscript.

Comment 3:

In line numbers 215, 251, 450, 483: correct the word dust ‘event’ to ‘events’.

7. PLOS authors have the option to publish the peer review history of their article (what does this mean?). If published, this will include your full peer review and any attached files.

Reviewer #1: No

Reviewer #2: No

Reviewer #3: **Yes:** Abhay Kumar Singh

---

## [Author Response · Author response to Decision Letter 2]

2 Feb 2026

Authors' Response to the Reviewers' Comments

Submission ID: PONE-D-25-51186R1

Title: Analysis and Source Identification of Dust event in Khuzestan Province Using MODIS and SENTINEL Satellite Data and HYSPLIT Model (2018–2022)

Dear Editor,

Thank you for providing us the opportunity to revise and resubmit our manuscript, “Analysis and Source Identification of Dust event in Khuzestan Province Using MODIS and SENTINEL Satellite Data and HYSPLIT Model (2018–2022)”, for publication in your esteemed journal. We sincerely appreciate the time and effort you and the reviewers have invested in evaluating our work. Their detailed feedback and constructive suggestions have greatly enhanced the quality of our manuscript.

We have carefully reviewed and addressed all the reviewers’ comments and suggestions. Our responses are organized systematically, detailing the changes made to the manuscript to incorporate each suggestion. Below, we provide a comprehensive summary of our revisions and responses:

Reviewer #2

Comment 1. Despite the frequent mention of the HYSPLIT model, its application in the manuscript remains largely indirect and qualitative. The study does not employ standard quantitative trajectory-analysis techniques (trajectory clustering, PSCF, CWT, residence-time analysis), and the interpretation of dust sources is effectively replaced by correlation-based analysis between AOD in Iraq and Ahvaz. Such statistical dependencies demonstrate temporal coherence but do not provide physically grounded, mass-transport–based source attribution. As a result, the role of HYSPLIT is substantially weaker than claimed, and the conclusions regarding dominant source regions remain partially speculative. The authors are strongly encouraged to consult https://doi.org/10.1016/j.gloplacha.2025.104935 (Figure 1) and https://doi.org/10.1016/j.atmosres.2022.106518 (Figure 10) to refine the cluster-analysis methodology. These works should also be cited in the manuscript.

Response: We sincerely thank the reviewer for this valuable suggestion. Implementing the recommended improvements significantly enhanced our results, which could potentially lead to substantial advancements in dust management strategies in Khuzestan Province. The corresponding revisions have been highlighted in blue and are located from lines 610 to 682, as detailed below.

3.4.1 Clustered Dust Pathways

In this study, a total of 220 dust transport trajectories were analyzed for the year 2022, which was selected as a representative year characterized by high levels of dust pollution. These trajectories were generated using the HYSPLIT model driven by GLDAS meteorological data with a spatial resolution of 1° and a 3P configuration, and were computed as backward trajectories. Subsequently, a trajectory clustering procedure was applied to identify the dominant dust transport patterns. The results of this analysis are presented in Figure 8 and Table 3.

The backward trajectory analysis revealed six primary transport pathways with distinct contributions to dust loading over Ahvaz. Pathway 1, originating from Kuwait, accounted for 8.70% of the total trajectories and exhibited a relatively minor influence on the dust burden over the city. Pathway 2, which traverses the western border regions of Iran and eastern Iraq and passes over the desiccated areas of the Hoor Al-Azim Wetland, represented the dominant pathway, contributing 43.48% of the total trajectories and exerting the strongest influence on dust pollution in Khuzestan Province and Ahvaz. Pathway 3, encompassing the shared regions of southwestern Kuwait and southeastern Iraq, contributed 13.04% of the total. Pathway 4, originating from southern Iran and internal sources, particularly the provinces of Fars and Bushehr, accounted for 8.70% and showed a limited contribution to dust loading. Pathway 5, extending from the desert regions of Iraq and further influenced by source areas in Syria and Jordan, contributed a substantial 21.74% and played a significant role in enhancing the dust influx into Ahvaz. Finally, Pathway 6, associated with air mass flows originating from Europe and Turkey and subsequently passing over Syria and Iraq before reaching Ahvaz, accounted for only 4.35%, representing the least influential transport route.

The integrated assessment of these pathways indicates that, with the exception of Pathways 1 and 4, all remaining transport classes traverse the desert regions of Iraq, collectively accounting for 82.6% of the total dust transport routes. This pattern underscores the strategic and dominant role of Iraqi desert landscapes as primary source and transfer zones for dust affecting Khuzestan Province and the city of Ahvaz. Moreover, the high proportion of air mass flows passing through the western border regions of Iran and northeastern Iraq (exceeding 40%) highlights the potential for targeted management strategies, transboundary cooperation, and source-control policies in these areas to substantially mitigate the dust burden impacting western Iran, particularly Ahvaz, and to enhance the effectiveness of environmental and public health risk management.

Table 3. Percentage Share of Dust Trajectory Clusters

Cluster Ratio

1 8.70%

2 43.48%

3 13.04%

4 8.70%

5 21.74%

6 4.35%

Figure 8. Backward Trajectory Clusters for Dust Transport

3.4.2 Dust Emission Hotspots: WPSCF & WCWT Analysis

The analysis of the WPSCF and WCWT maps presented in Figure 9 indicates that the main dust hotspots are located along a northwest–southeast belt in Iraq, aligning with the prevailing regional wind direction (northern Shamal winds) from Iraq and Syria toward Khuzestan. According to the WPSCF map, where AOD values range from 0 to 1, areas with values approaching 1 were identified as primary dust sources. Central, eastern, northern, and northeastern Iraq particularly regions surrounding the dried Hoor al-Azim wetland, the Mesopotamian basin, and abandoned agricultural lands around Baghdad with AOD values between 0.6 and 0.8, play the most significant role in dust particle generation. Additionally, the Syria–northwestern Iraq border, exhibiting high AOD indices, is recognized as a key transboundary source that directs dust particles along a southeastward path. Areas with moderate AOD values (0.4–0.6), including southern Iraq, Kuwait, and the border regions between Iraq and Khuzestan, act as secondary sources. These areas complete the transport pathways of dust plumes and contribute to increasing the concentration of suspended particles as they reach Khuzestan. The results indicate that during the studied days, particle concentrations in these regions were exceptionally high, with backward trajectories passing through these same points.

The WCWT index analysis further revealed that critical cores with AOD values greater than 1 are predominantly concentrated in two key regions: the central and southern Mesopotamian basin and the northwestern border strip of Khuzestan. In the Mesopotamian basin, the lowlands between the Tigris and Euphrates rivers particularly dried wetlands such as Hoor al-Hammar and the western parts of Hoor al-Azim have become major dust sources due to drought and reduced soil moisture. The northwestern border strip of Khuzestan, exhibiting high index values, indicates that dust particles generated in these cores enter the province with minimal deposition and maximum kinetic energy, intensifying local dust concentrations.

The elongated, linear arrangement of colored pixels from northwest to southeast reflects the dominance of stable synoptic flows, especially the northern Shamal winds. Areas with moderate AOD values (0.6–0.9) in the deserts of eastern Syria and Ninawa Province act as “amplifiers”; the passage of air masses through these regions gradually loads dust, reaching saturation points in central Iraq. In contrast, pixel discontinuities in remote northwestern areas suggest that contributions from distant deserts to severe events in Khuzestan are limited, as coarse particles tend to deposit over longer transport paths.

High AOD values near the borders of Khuzestan (0.9 to above 1) indicate that incoming air masses not only carry long-range dust but also interact with dense dust plumes generated in local cores, such as southern and eastern Ahvaz, leading to sudden increases in pollution indices at ground stations.

Overall, this analysis demonstrates that the dust crisis in Khuzestan operates within a transboundary system: distant sources in Syria and northern Iraq generate the initial cores of dust plumes, while the dried lands of southern Iraq and the Khuzestan border strip act as amplifying reservoirs, producing the highest AOD values at the closest geographical proximity to the province. These results clearly highlight the simultaneous role of transboundary and local sources in the occurrence of severe dust events and can serve as a basis for management policies aimed at mitigating the environmental and health impacts of this phenomenon.

Figure 9. Source Contribution and Dust Transport Pathways to Khuzestan Based on WPSCF and WCWT Models

Comment 2. The methodological section has been expanded and formally improved, yet in its current form it remains overloaded and conceptually unfocused. An excessive number of statistical methods is applied without clear justification for their necessity or hierarchy, and many of them ultimately confirm the same relationship between regional and local aerosol loading. This accumulation of techniques gives the impression of methodological inflation rather than a hypothesis-driven analytical framework. Reducing and structuring the methodological toolkit would considerably improve clarity and reproducibility.

Response: Thanks. The methodology has been revised, and the number of statistical indices has been reduced to focus on the main metrics, including R², Pearson Correlation, Spearman Rank Correlation, Mutual Information (MI), and K-Means Clustering. These indices were selected because each provides complementary insights: R² quantifies the proportion of variance explained, Pearson Correlation captures linear relationships, Spearman Rank Correlation identifies monotonic associations, Mutual Information (MI) reveals both linear and non-linear dependencies, and K-Means Clustering highlights natural groupings in the data. Other statistical metrics that largely confirmed the same relationships were removed to streamline the analysis and improve clarity. The corresponding revisions are highlighted in blue from lines 382 to 397. Moreover, the methodology section throughout the manuscript has been updated to reflect these changes, with all additions in the Materials and Methods section also highlighted in blue.

2.9. Statistical Analysis

To provide a clear framework for evaluating the relationships among the study variables, key statistical indices were selected to quantify linear and non-linear associations. These indices offer a rigorous basis for interpreting the dependencies and interactions present in the dataset.

Coefficient of Determination (R²): Coefficient of Determination (R²): Measures how well independent variables explain the variability of the dependent variable. Values near 1 indicate strong explanatory power, while lower values suggest other influencing factors [38].

Pearson Correlation: This coefficient measures the linear association between two variables, providing insight into their co-variation and direction of influence [39].

Spearman Rank Correlation: By ranking values, this non-parametric method captures monotonic relationships, making it robust for non-linear datasets [40].

Mutual Information (MI): MI measures shared information between variables based on entropy. Higher MI values indicate stronger dependencies, capturing both linear and non-linear interactions [43].

K-Means Clustering: This unsupervised approach partitions data into K clusters based on similarity. The algorithm iteratively assigns data points to centroids and recalculates their positions until convergence. Techniques such as the Elbow Method and Silhouette Score help identify the optimal number of clusters [45].

Comment 3. Figure 1 raises serious concerns regarding scientific graphics quality and the correct representation of the methodological workflow. Visually, it appears to be generated automatically or by AI, which is suggested by generic iconography, inconsistent typography, and incorrect scientific notation—most notably the suspicious rendering of “PM2.5” and “PM10.” The figure must be completely redrawn manually using proper scientific notation and a more rigorous, discipline-appropriate design.

Response: Thanks. Figure 1 has been revised and completely redrawn to ensure proper scientific notation and a rigorous, discipline-appropriate design.

Figure 1. Step-by-Step Process of Dust Source Analysis Using Satellite Data and HYSPLIT Model

Comment 4. Despite the revisions, the question of conceptual originality relative to the authors’ previously published article in Scientific Reports (2025) remains insufficiently resolved. Although the analysis period is extended and additional statistical metrics are introduced, the core region, datasets, physical interpretation, and key conclusions remain largely similar. The authors should clearly and explicitly articulate what fundamentally new scientific question this work addresses, beyond methodological elaboration of already published material.

Response: Thanks. The previous section has been completely removed and replaced with a new analysis. Based on HYSPLIT trajectory modeling and standard quantitative trajectory-analysis techniques (including trajectory clustering, PSCF, and CWT), it was determined that the Hoor al-Azim Wetland lies within the key dust-corridor entering Khuzestan Province. Therefore, the study examined monthly changes in the wetland’s surface area and water level (depth). For 2022, monthly maps of Hoor al-Azim were generated to provide detailed information on fully dried areas of the wetland. To assess its impact on dust events, the relationships between the wetland dynamics and dust were analyzed using R² and K-Means Clustering, highlighting the correlation with AOD and clarifying the wetland’s role in dust generation in the province. These revisions are highlighted in blue from lines 788 to 902, as detailed below.

3.7. Assessment of Hoor Al-Azim Wetland Shrinkage

The extensive wetlands of Khuzestan Province, particularly the Shadegan and Hoor al-Azim Wetlands, have historically acted as natural filters by trapping and depositing suspended particles, thereby reducing atmospheric dust levels and mitigating the intensity and frequency of local dust events. Under conditions of severe hydrological stress, however, this filtration capacity weakens, increasing the vulnerability of the system to internal dust generation. Based on recent assessments, 2022 was identified as the driest year for these wetlands within the 2018–2022 period (Yousefi Kebriya et al., 2025). Considering that more than 70% of dust transport trajectories pass through Hoor al-Azim Wetland, this study specifically examined the hydrological condition of the wetland (i.e., its water availability or dryness) and its relationship with dust events during 2022.

The variations in wetland surface water extent (derived from the MNDWI index) and the water level of the Hoor al-Azim Wetland over a five-year period (2018 to the end of 2022), as well as the relationship between these variables, are illustrated in Figure 12. During 2018, both indicators exhibit relatively low and stable values, with the surface area generally remaining below 900 km² and water levels fluctuating between approximately 1.7 and 2.9 m, reflecting a moderately constrained hydrological regime. A pronounced expansion phase is observed throughout 2019 and early 2020, characterized by a substantial increase in wetland surface area, reaching a peak of nearly 2,000 km² in April 2019, concurrently with elevated water levels exceeding 5 m. This synchronous response indicates a strong positive relationship between water availability and the spatial extent of surface water, confirming the sensitivity of the MNDWI-derived area to variations in wetlan

---

## [Decision Letter · Decision Letter 2]

11 Mar 2026

PONE-D-25-51186R2Analysis and Source Identification of Dust event in Khuzestan Province Using MODIS and SENTINEL Satellite Data and HYSPLIT Model (2018–2022)PLOS One

Dear Dr. Yousefi Kebriya,

Thank you for submitting your manuscript to PLOS ONE. After careful consideration, we feel that it has merit but does not fully meet PLOS ONE’s publication criteria as it currently stands. Therefore, we invite you to submit a revised version of the manuscript that addresses the points raised during the review process.

We look forward to receiving your revised manuscript.

Kind regards,

Joanna Tindall, PhD

Staff Editor

PLOS One

Journal Requirements:

Additional Editor Comments:

Please include the appropriate attributions to Figure 2, 3 and Table 1. Please take this opportunity to check that all tables, figures and datasets are correctly attributed and their sources cited appropriately.

Reviewers' comments:

Reviewer's Responses to Questions

**Comments to the Author**

1. If the authors have adequately addressed your comments raised in a previous round of review and you feel that this manuscript is now acceptable for publication, you may indicate that here to bypass the “Comments to the Author” section, enter your conflict of interest statement in the “Confidential to Editor” section, and submit your "Accept" recommendation.

Reviewer #2: All comments have been addressed

Reviewer #3: All comments have been addressed

2. Is the manuscript technically sound, and do the data support the conclusions?

Reviewer #2: Yes

Reviewer #3: Yes

3. Has the statistical analysis been performed appropriately and rigorously? 

Reviewer #2: Yes

Reviewer #3: Yes

4. Have the authors made all data underlying the findings in their manuscript fully available?

Reviewer #2: Yes

Reviewer #3: Yes

5. Is the manuscript presented in an intelligible fashion and written in standard English?

Reviewer #2: Yes

Reviewer #3: Yes

6. Review Comments to the Author

Reviewer #2: The authors have adequately addressed the major concerns raised in the previous review and have substantially improved the methodological rigor and clarity of the manuscript. Given the revisions and strengthened analyses, I recommend the article for publication in its current form.

Reviewer #3: (No Response)

7. PLOS authors have the option to publish the peer review history of their article (what does this mean?). If published, this will include your full peer review and any attached files.

Reviewer #2: No

Reviewer #3: **Yes:** Abhay Kumar Singh

---

## [Author Response · Author response to Decision Letter 3]

7 Apr 2026

Authors’ Response to the Reviewers’ Comments

Submission ID: PONE-D-25-51186R2

Title: Analysis and Source Identification of Dust Event in Khuzestan Province Using MODIS and SENTINEL Satellite Data and HYSPLIT Model (2018–2022)

Dear Editor,

We sincerely thank you and the reviewers for your valuable time, constructive feedback, and positive evaluation of our manuscript. We are pleased that the revised version has been found to be technically sound and suitable for publication.

We have carefully considered the final editorial comment and have made the required minor revisions accordingly. The changes have been incorporated into the revised manuscript, and they are highlighted in the tracked version.

Comment 1 (Editor): Please include the appropriate attributions to Figure 2, 3 and Table 1. Please ensure that all tables, figures, and datasets are correctly attributed and their sources are appropriately cited.

Response: Thanks. We sincerely thank the editor for this valuable comment. The appropriate attributions have been added to Figure 2, Figure 3, and Table 1, and all relevant sources are now clearly indicated in the manuscript. The corresponding revisions have been highlighted in yellow for clarity.

For Figure 2, the source has been added in the caption as follows:

“Figure 2. Location and Land Cover Map of Khuzestan Province and Shadegan Wetland. Source: NASA Worldview (EOSDIS), MODIS True Color imagery (2024); processed and prepared by the authors.”

This modification can be found in lines 237–238. In addition, the attribution “(© NASA Worldview, EOSDIS, MODIS True Color imagery, 2024)” has also been included within the figure itself.

For Table 1, appropriate references have been added alongside the caption title as follows:

“TABLE 1. Air Quality Index Basics Pollution [38, 39],”

which can be found in line 307.

For Figure 3, the source has been added in the caption as follows:

“Figure 3. Regression analysis chart of AOD and PM₂.₅ and PM₁₀ for the years 2020–2022. Source: Derived from MODIS satellite data (NASA EOSDIS) and ground-based observations; processed and analyzed by the authors.”

This revision is reflected in lines 434–435, and the corresponding attribution has also been added within the figure.

Furthermore, all figures, tables, and datasets throughout the manuscript have been carefully reviewed to ensure that proper attribution and source citation are consistently applied.

Figure 2. Location and Land Cover Map of Khuzestan Province and Shadegan wetland. Source: NASA Worldview (EOSDIS), MODIS True Color imagery (2024); processed and prepared by the authors.

TABLE 1. Air quality index Basics Pollution [38, 39]

Levels of Concern Values of Index AAI Values of Index AOD Values of Index AQI

Air quality is satisfactory, and air pollution poses little or no risk. <0 <0.1 0-50

Air quality is acceptable. However, there may be a risk for some people, particularly those who are unusually sensitive to air pollution. 0-0.5 0.1-0.49 51-100

Some members of the general public may experience health effects; members of sensitive groups may experience more serious health effects. 0.5-2 0.5-1 101-200

Health alert: The risk of health effects is increased for everyone. 2-5 >1 201-300

Health warning of emergency conditions: everyone is more likely to be affected. >5 - >300

Figure 3. Regression analysis chart of AOD and PM₂.₅ and PM10 for the year 2020-2022. Source: Derived from MODIS satellite data (NASA EOSDIS) and ground-based observations; processed and analyzed by the authors.

Comment 2 (Editor): Please take this opportunity to check that all tables, figures and datasets are correctly attributed and their sources cited appropriately.

Response: Thanks. All figures, tables, and datasets in the manuscript have been carefully reviewed to ensure proper attribution and accurate source citation. For figures derived from satellite data and the Google Earth Engine platform, appropriate source attributions have been added in accordance with the data usage and citation guidelines of the respective providers (e.g., NASA MODIS and related products). For other figures and tables based on ground-based observations, original datasets, or software-generated analyses, the existing attributions were deemed appropriate and have been retained. These revisions have been implemented throughout the manuscript and are highlighted in the revised version.

Comment 3 (Editor): Please carefully review the reference list to ensure that all citations are complete, accurate, and up to date. In particular, if any cited articles have been retracted, they should either be removed and replaced with relevant current references or clearly identified as retracted, with an appropriate explanation provided in the manuscript.

Response: Thanks. The entire reference list has been carefully reviewed to ensure that all citations are complete, accurate, and up to date, and that in-text citations are fully consistent with the reference list.

All references have been checked and properly formatted. One reference that was found to be unavailable (Hosseini Jani et al., 2020) has been removed and replaced with a relevant and valid source. The new reference has been added as Reference [14] in line 80 and is also reflected in the manuscript accordingly (line 1165):

Raji, H., Riahi, A., Borsi, S. H., Masoumi, K., Khanjani, N., AhmadiAngali, K., Goudarzi, G., & Dastoorpoor, M. (2020). Acute effects of air pollution on hospital admissions for asthma, COPD, and bronchiectasis in Ahvaz, Iran. International Journal of Chronic Obstructive Pulmonary Disease, 15, 501–514. https://doi.org/10.2147/COPD.S231317

In addition, Reference 25 has been corrected and properly formatted (line 1197):

Soleimani Sardo, F., & Krakauer, N. (2026). Dust modeling using MODIS data and WRF-Chem and HYSPLIT models (Case study of dust storm from December 16 to 20, 2016). Sustainable Earth Trends, 6(1), 55–65. https://doi.org/10.48308/set.2025.238019.1098

Furthermore, Reference 43 has been revised and updated accordingly (line 1251):

Yousefi Kebriya, A. R., Shiukhy Soqanloo, S., & Yousefi Kebriya, H. (2026). Monitoring and analysis of relationship between vegetation cover and air pollution using Sentinel and MODIS satellite data in Mazandaran Province. Iranica Journal of Energy & Environment, 17(2), 309–328. https://doi.org/10.5829/ijee.2026.17.02.08

All remaining references have also been thoroughly reviewed for completeness, accuracy, formatting consistency, and proper citation within the manuscript.

No retracted articles were identified among the cited references.

We appreciate the editor’s guidance in improving the quality and reliability of the manuscript.

---

## [Editor Report · Decision Letter 3]

13 Apr 2026

Analysis and Source Identification of Dust event in Khuzestan Province Using MODIS and SENTINEL Satellite Data and HYSPLIT Model (2018–2022)

PONE-D-25-51186R3

Dear Dr. Yousefi Kebriya,

We’re pleased to inform you that your manuscript has been judged scientifically suitable for publication and will be formally accepted for publication once it meets all outstanding technical requirements.

Kind regards,

Mikalai Filonchyk

Academic Editor

PLOS One
---

## [Editor Report · Acceptance letter]

PONE-D-25-51186R3

PLOS One

Dear Dr. Yousefi Kebriya,

I'm pleased to inform you that your manuscript has been deemed suitable for publication in PLOS One. Congratulations! Your manuscript is now being handed over to our production team.

Kind regards,

on behalf of

Dr. Mikalai Filonchyk

Academic Editor

PLOS One